# Neuro-Symbolic AI for Analytical Solutions of Differential Equations

## Abstract

Analytical solutions to differential equations offer exact insight but are rarely available because discovering them requires expert intuition or exhaustive search in large combinatorial spaces. We introduce SIGS, a neuro-symbolic framework that automates this process. SIGS uses a formal grammar to generate only syntactically and physically valid building blocks, embeds these expressions into a continuous latent space, and then searches this space to assemble, score, and refine candidate closed-form solutions by minimizing a physics-based residual. This design unifies symbolic reasoning with numerical optimization; the grammar constrains candidate solution blocks to be proper by construction, while the latent search makes exploration tractable and data-free. Across a range of differential equations, SIGS recovers exact solutions when they exist and finds highly accurate approximations otherwise, outperforming tree-based symbolic methods, traditional solvers, and neural PDE baselines in accuracy and wall-clock efficiency. These results are a step forward, integrating symbolic structure with modern ML to discover interpretable, closed-form solutions at scale.

## 1 Introduction

The understanding of physical processes has been a long-standing effort for scientists and engineers. A key step in this endeavor is to translate physical insights (laws) into precise mathematical relationships that capture the underlying phenomena. These relationships are then tested through experiments that either validate the proposed hypothesis or suggest refinements. Among such mathematical formulations, differential equations (DEs) are especially ubiquitous across disciplines, as they describe how physical quantities evolve over time and space. Analytical solutions, closed-form expressions satisfying governing equations and boundary/initial conditions, not only validate theory against experiment but also reveal intrinsic properties such as stability, periodicity, and symmetries. Classical analytical methods are inherently compositional: they assemble solutions from elementary building blocks such as eigenfunctions, basis expansions, or Green's functions.

Unlike the inverse problem of discovery the governing equations given measurements of the solution, which has been widely considered by adapting symbolic regression Petersen et al. (2019b); Landajuela et al. (2022); Petersen et al. (2021); Yu et al. (2025); Kamienny et al. (2022); Biggio et al. (2021); Vastl et al. (2022) to this setting, the forward problem of discovering analytical solutions to DEs, considered here, is less explored. In this context, proposed approaches include genetic programming and its variants (Tsoulos & Lagaris, 2006; Seaton et al., 2010; Kamali et al., 2015; Boudouaoui et al., 2020). Lately, symbolic approaches have been enriched with machine learning components to overcome this combinatorial complexity. Lample & Charton (2019) train neural networks on sequence representations of trees in order to solve simple explicit ODEs. Wei et al. (2024) propose SSDE, a methodology that employs a recurrent neural network to generate symbolic candidates, guided by a reinforcement learning policy constrained by the governing equations and conditions. As a baseline, they considered the accuracy of fitting symbolic solutions to functions obtained by physics-informed neural networks, relying on deep symbolic regression (Petersen et al., 2019a). Cao et al. (2024) use transfer learning to lift genetic programming results from

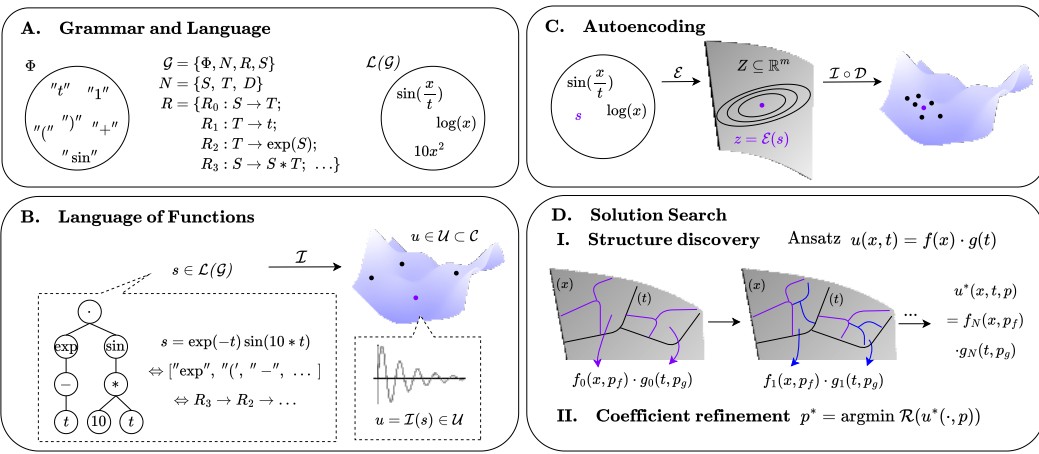

Figure 1: Overview over the proposed Symbolic Iterative Grammar Solver (SIGS). A. Terminal symbols $\Phi$ and rules $R$, together with non-terminals $N$ and starting symbol $S$, form the grammar $\mathcal{G}$ which generates the mathematical expressions in the library $\mathcal{L}(\mathcal{G})$. B. Each expression $w \in \mathcal{L}(\mathcal{G})$ is identified with a function $u$ in the finite set of candidate functions $\mathcal{U}$. C. The encoder $\mathcal{E}$ and decoder $\mathcal{D}$ of the Grammar Variational Autoencoder (GVAE, (Kusner et al., 2017)) embed the finite $\mathcal{L}(\mathcal{G})$ into the continuous latent space $Z$. D. Given a differential equation and system conditions, a structure search is performed over $z \in Z$ using iterative clustering, followed by a separate optimizations of the constants in the final structure, optimizing for lowest residual $\mathcal{R}$ of the corresponding candidate function $u = \mathcal{I} \circ \mathcal{D}(z) \in \mathcal{U}$.

one-dimensional problems to higher dimensions (HD-TLGP). Unlike symbolic regression, where the primitives of differential operators are chosen from a dictionary of fundamental operations such as curl or divergence, there exists no principled way to systematically choose components in solution discovery to combine and get mathematically proper and physically plausible solutions. As a result, solution discovery methods have tended towards two extremes: (i) unconstrained search, which faces combinatorial explosion, sensitivity to initialization, and lack of principled incorporation of domain knowledge; or (ii) narrow pretraining, which biases discovery toward limited problem classes and hinders generalization. A principled middle ground is missing.

This raises the key question: Can we design a framework that generalizes across PDEs while systematically constraining the search to mathematically admissible, physically meaningful solutions?

We answer affirmatively with the Symbolic Iterative Grammar Solver (SIGS). At its core, SIGS casts solution discovery as a hierarchical, grammar-guided composition of analytic atoms (eigenfunctions and related sub-expressions). The hierarchy operates at two levels. At the top level, an Ansatz specifies the structural form of candidate solutions: e.g. the Ansatz $f(x) \times g(t)$ restricts the search to a product of spatial, temporal, or combined terms. At the lower level, each placeholder function in the Ansatz is instantiated with concrete atoms drawn from a grammar; e.g. $f(x)$ is replaced by $sin(x)$. Using formal grammars (Hopcroft & Ullman, 1979), elementary functions act as terminals while operations such as addition or exponentiation act as production rules. This formalism generalizes classical construction techniques, providing a principled way to generate only admissible expressions and to systematically explore the solution space defined by the PDE.

To overcome the combinatorial complexity of assembling such expressions, SIGS embeds grammar-generated candidates into a continuous latent manifold using a Grammar Variational Autoencoder Kusner et al. (2017). We further impose a novel topological regularization within the GVAE, ensuring that latent neighborhoods map smoothly to valid expressions and that clusters of candidates form convex regions. This embedding transforms discrete tree search into quasi-continuous optimization: instead of enumerating oper-

ators and integers, we navigate the latent manifold, progressively refining the search around promising regions. The final constants are then optimized with gradient descent, yielding exact or approximate analytical solutions.

Our key contributions are as follows.

- A grammar-based framework (SIGS) that efficiently balances computational complexity with generality by composing solution units through hierarchical Ansatz+atom combinations, modeled through formal grammars.
- The Topological Grammar VAE (TGVAE), which encodes admissible solutions onto a smooth latent manifold for efficient search.
- An efficient and task-agnostic approach that employs compositionality of solutions to solve a broad selection of PDEs, without the need for numerical data.
- State-of-the-art performance on recent benchmarks, including recovery of exact solutions and symbolic approximations of PDEs lacking closed-form solutions.

## 2 Method

Problem setup   We consider the generic form of a time-dependent partial differential equation (PDE) as (Molinaro et al., 2024),

$$
\begin{aligned}
\partial_t u + \mathbb{D}(u) &= \mathbf{f}, & \forall (\mathbf{x}, t) \in \Omega \times [0, T], \\
u(\mathbf{x}, 0) &= u_0(\mathbf{x}), & \forall \mathbf{x} \in \Omega, \\
\mathbb{B}[u](\mathbf{x}, t) &= g, & \forall (\mathbf{x}, t) \in \partial\Omega \times [0, T],
\end{aligned}
\tag{1}
$$

where $\Omega \subset \mathbb{R}^d$ is the spatial domain, $u \in \mathcal{U} \subseteq \mathcal{C}(\Omega \times (0, T))$ is the space-time continuous solution, $\mathbf{f} \in \mathcal{U}$ is a forcing term, $u_0 \in H^s(\Omega)$ an initial condition, $\mathbb{B}[u](\mathbf{x}, t)$ denotes the boundary conditions, and $\partial\Omega$ is the boundary of the domain. The differential operator can include higher-order derivatives, $\mathbb{D}(u) = \mathbb{D}(\xi, u, \partial_{tt}u, \nabla_{\mathbf{x}}u, \nabla_{\mathbf{x}}^2 u, ...)$, where $\xi \in \mathbb{R}^{d_\xi}$ are PDE parameters. We remark that Equation 1 represents a very general form of differential equations as the solution $u = u(x, t)$ is a function of both space and time. By setting $u = u(t)$, we recover general ODEs, while setting $u = u(x)$ enables us to recover time-independent PDEs from the same overall formulation. Henceforth, we use PDEs of the form of Equation 1, as the objects for which we discover analytical solutions. We call the collection of $\mathbf{f}, \mathbb{B}[u]$, and $u_0$ the system conditions that need to be specified in order to solve a given PDE. We define the symbolic form of a PDE as:

$$
\mathcal{S}(u) = \partial_t u + \mathbb{D}(u) - \mathbf{f}, \quad \forall u \in \mathcal{U}.
$$

We formulate solving PDEs as an iterative computational process, where given a domain discretization, a set of boundary and initial conditions, and the symbolic form of the PDE

$$
(\Omega, \mathbb{B}[u], u_0, S) \xrightarrow{\mathcal{D}(z)} u^i.
$$

The method searches for a parameterization $z$ of $u_z \in \mathcal{U}$ that minimizes the loss,

$$
\mathcal{R}(u) = \|\mathcal{S}(u)\|^2 + \|u(0, x) - u_0^i\|^2 + \|\mathbb{B}[u] - g\|^2,
\tag{2}
$$

where we generally use equal weighting between the residual terms. We restate our goal as finding an analytical expression $u^*$ that minimizes the residual $\mathcal{R}(u)$, yielding an analytical solution in case $\mathcal{R}(u^*) = 0$ and an analytical approximate solution if $0 < \mathcal{R}(u^*) << 1$.

Grammar Construction.   Analytic expressions are commonly represented as trees, with internal node labels denoting unary or binary expressions (e.g. "sin", "+") and leaves denoting constants or variables. However, care must be taken when generating such trees to avoid exponential complexity and the generation of syntactically wrong expressions Virgolin & Pissis (2022); Kissas et al. (2024). To alleviate this issue, we consider a Context-Free Grammar (CFG) (Chomsky, 1956; Hopcroft & Ullman, 1979) as a principled way to generate exactly the classes of atoms included in an Ansatz. A CFG is defined as $\mathcal{G} = \{\Phi, N, R, S\}$, where $\Phi$ is the set of terminal symbols, $N$ is the set of non-terminal symbols and $\Phi \cap N = \emptyset$,

$R$ is a finite set of production rules and $S \in N$ is the starting symbol. Each rule $r \in R$ is a map $\alpha \to \beta$, where $\alpha \in N$, and $\beta \in (\Phi \cup N)^*$ (see Fig. 1A). A language $\mathcal{L}(\mathcal{G})$ is defined as the set of all possible terminal strings that can be derived by applying the production rules of the grammar starting from $S$, or all possible ways that the nodes of a derivation tree can be connected starting from $S$ as $\mathcal{L}(\mathcal{G}) = \{w \in \Phi^* \mid S \to^* w\}$, where $\to^*$ implies $T \geq 0$ applications of rules in $R$. Each expression is equivalently represented by the string $w$ (as a sequence of symbols), by the list of rules applied to generate $w$ from $S$, and by a derivation tree that represents the syntactic structure of string $w \in \mathcal{L}(\mathcal{G})$ according to grammar $\mathcal{G}$. We define an interpretation map $\mathcal{I} : \mathcal{L}(\mathcal{G}) \to \mathcal{U}$, which assigns to each syntactic expression $w \in \mathcal{L}(\mathcal{G})$ semantic meaning in terms of a function $u_w : D \to \mathbb{R}$. The set of all functions represented by the grammar is $\mathcal{U}(\mathcal{G}) = \{u_w : D \to \mathbb{R} \mid u_w = \mathcal{I}(w), w \in \mathcal{L}(\mathcal{G})\}$. We refer to $u_w$ as $u$ in the future to simplify the notation.

**Compositional Ansatz.** When using SIGS on a specific problem, the user may specify a structural Ansatz $F$ that outlines the compositional nature of the proposed solution. For example, one could specify spatiotemporal separability as $u(x,t) = \sum_{j=1}^{K} a_j T_j(t)\phi_j(x)$, leaving the spatial eigenfunctions $\phi_j$ and temporal factors $T_j$, to be chosen by SIGS. In addition to $\phi_j$ and $T_j$, the user may include atoms that encode physical mechanisms at the expression level; such as transport phases, $kx - \omega t$; viscous shock profiles, $\tanh((x_0 + x - ct)/\nu)$; or other motifs known to describe the dynamics of interest exactly or approximately. Localized atoms such as Gaussians can also be included to capture spatially confined phenomena. The Ansatz may include hybrid factors that mix space and time, allowing $u(x,t) = \sum_{j=1}^{K} a_j T_j(t)\phi_j(x)\psi_j(x,t)$ which relaxes separability while retaining controlled, interpretable compositions.

Searching the resulting high-dimensional combinatorial spaces requires a trade-off between generality and complexity. We embed atoms (sub-trees) instead of primitives (unary, binary operators, reals, and variables) to decrease the combinatorial complexity of solutions. In the full Ansatz generality, the solution construction could be performed by considering a number of arbitrary combinations between atoms. This approach would result in a combinatorial explosion, partially losing the benefit of considering atoms. For this reason, we assume that the solutions can be described exactly (or sufficiently well) by the chosen Ansatz. To include the Ansatz into the grammar, we denote by $A : \{\mathcal{L}(\mathcal{G})\}^L \to \mathcal{L}(\mathcal{G})$ the assembly map that composes the individual components into the final solution following the Ansatz. This restricted function class is obtained by activating only those nonterminals and production rules that implement the user's Ansatz and its permitted atom categories, and by enforcing the assembly production dictated by $A$. The Ansatz thus induces a restriction on the language $\mathcal{L}_A(\mathcal{G}) = \{A(w^1, ..., w^L) : w^c \in \mathcal{L}_c(\mathcal{G})\}$ for the component classes $c$ required by the Ansatz. In all cases, $A$ realizes the user's choice by assembling requested categories into a single symbolic candidate that is then scored by the PDE residual. In summary, the Ansatz specifies which families of atoms and couplings are admissible, the CFG generates those atoms and couplings, and the interpretation map turns each derivation into a candidate function over which SIGS optimizes the PDE residual.

**Grammar Variational Autoencoders.** To make the search more efficient, we embed $w \in \mathcal{L}_A(\mathcal{G})$ into a low-dimensional continuous manifold using a Grammar Variational Autoencoder (Kusner et al., 2017). The encoder is defined as $q_\phi(z|w)$ and the decoder $p_\theta(w|z)$, for $z \in Z$ and $w \in \mathcal{L}_A(\mathcal{G})$. The GVAE is trained by minimizing the objective:

$$\mathcal{L} = \mathcal{L}_{\text{recon}} + \gamma \, \text{KL}(q(z|w) \| \, p(z)),$$

where $\mathcal{L}_{\text{recon}}$ the cross-entropy loss between the predicted and the baseline grammar rules, and $\text{KL}(q(z|w)||p(z))$ the KL divergence between the encoder and the prior distributions.

Training the GVAE does not require numerical data, only expressions $w \in \mathcal{L}$. In practice, we handle numerical matrices with entries $\{0, 1\}$, encoding which rules are employed in which order to generate $w$, and impose grammar relations through masking parts of these matrices to only allow related elements to interact. These grammar masks are also required for training the model.

Geometry Regularization. When we sample the latent manifold, we often evaluate latent vectors in regions with little or no support from the training distribution, and can also get trapped in topological artifacts of the latent space. In both cases, the decoder produces degenerate outputs. For this reason, we impose a geometry-aware regularizer that constrains the search inside a data-supported enclosure, removes small topological artifacts at the working resolution, and smooths the decoder so that small latent moves produce predictable output changes.

We augment the GVAE objective with three regularizers (details in App. A.3). A convex-enclosure loss $\mathcal{L}_{\text{Hull}}$ that discourages latents from leaving the data-supported region estimated from training codes Gonzalez (1985); Rockafellar (2015). A persistent-homology loss $\mathcal{L}_{\text{ph}}$ that suppresses small spurious loops/gaps in the latent cloud at a fixed working scale Edelsbrunner & Harer (2010). A decoder-smoothness loss $\mathcal{L}_{\text{smooth}}$ that penalizes large second-order changes in the decoder, so nearby latents decode to predictably similar functions (Hutchinson, 1989). We combine these losses with the reconstruction and the KL loss to define the regularized loss of the TGVAE (Topological Grammar Variational Autoencoder):

$$\mathcal{L} = \mathcal{L}_{\text{recon}} + \gamma \, \text{KL}(q(z|w) \| \, p(z)) + \, \mathcal{L}_{\text{topo}}, \quad \mathcal{L}_{\text{topo}} = \mathcal{L}_{\text{Hull}} + \mathcal{L}_{\text{ph}} + \mathcal{L}_{\text{smooth}}.$$

Solution Discovery. The solution discovery is split in two stages (see Fig. 1D, and details in App. B): In the structure search, we iteratively explore the latent space for a candidate function included in the structural Ansatz while minimizing the PDE residual, and then optimize its numerical constants in a separate stage. For searching, we consider a deterministic encoding $\mathcal{E}(w) = \mu_\phi(w) \in Z$ and decoding $\mathcal{D} : Z \to \mathcal{L}_A(\mathcal{G})$ obtained by the argmax decoding under the grammar mask. Composing with $\mathcal{I}$, we have $\mathcal{I} \circ \mathcal{D} : Z \to \mathcal{U}_A(\mathcal{G})$, so each $z \in Z$ corresponds to a function $u = \mathcal{I}(\mathcal{D}(z)) \in \mathcal{U}_A(\mathcal{G})$. Let $\tau : \mathcal{L}(\mathcal{G}) \to \mathcal{T}$ be a semantic map that assigns tags, e.g. variables, deterministically computed from the parse tree $w$, computed once after training and used for any downstream solution problem. For a given differential equation, we choose the admissible tag set, e.g. any function with $x, y$ arguments, and restrict the search to the type-constrained latent subspace $Z' = \{z \in Z : \tau(D(z)) \in \mathcal{T}' \subseteq \mathcal{T}\}$.

Let $\kappa : Z' \to \{1, ..., m\}$ be a clustering map in the latent space and denote the clusters $C_j = \kappa^{-1}(j)$. We cluster a given subspace based on $z \in Z'$, and then solve a discrete selection problem to choose the cluster that contains the most promising solution forms for each $T$ and $\phi$, $j^* = \arg \min_{1 \le j \le m} [\inf_{z \in Z_j \subset C_j} \mathcal{R}(\mathcal{D}(z))]$, where $Z_j$ can be constructed by either only the expressions from the training set that fall in $Z'$ or the expressions together with samples from the generative model. Within the best cluster $C_{j^*}$, a global latent search is performed:

$$z^* = \arg \min_{z \in C_{j^*}} \mathcal{R}(\mathcal{D}(z)),$$

either by a global optimizer or iterative clustering, performing discrete selection, and sampling from the most promising cluster until $\mathcal{R}(\mathcal{D}(z))$ drops below a threshold. The solution takes a parametric form $u^*(\cdot, p)$, including constants $p$ that are only represented in the grammar with limited precision. Thus, we perform a parameter refining step. We consider a gradient based method (Adam, Kingma & Ba, 2014), and minimize the loss until a termination criterion is triggered, $\mathcal{R}(u) \le 10^{-8}$. The loss $\mathcal{R}(u)$ is augmented here by the hull loss $\mathcal{R}'(u) = \mathcal{R}(u) + \mathcal{L}_{\text{hull}}$ to penalize whichever latent falls out of the hull defined during training.

## 3 Experiments and Results

We conduct comprehensive experiments to evaluate SIGS against state-of-the-art symbolic methods for solving PDEs. Our evaluation comprises three components: cross-validation on benchmarks sourced from the literature (Table 12), assessment on more complex PDE problems with and without known analytical solutions (Table 14), and an ablation study that examines how topology-aware regularization improves sampling efficiency. Details on our implementation of the grammar and GVAE can be found in Appendix A.

Experimental Setup. Our benchmark suite comprises seven PDEs of hyperbolic, parabolic, and elliptic families. Four problems admit known analytical solutions: viscous Burgers', 1D Diffusion, 1D Wave, and 2D Damped Wave equations. For the case with no known analytic solution, we consider three Poisson problems with superposition of different numbers of Gaussian source terms to test the approximation capabilities of the method.

We compare against two recent symbolic discovery methods: HD-TLGP Cao et al. (2024), and SSDE Wei et al. (2024). Both of these methods sample discrete trees by combinatorially combining elements of a user defined dictionary. Moreover, HD-TLGP, considers an Ansatz where the solution is separable in dimension, e.g. $g(x, y) = f(x)g(y)$, as well as the solution in one dimension as prior knowledge. SSDE considers a recursive single-variable decomposition Ansatz, e.g. $u(x, y) = g(x, f(y, c))$ and couples reinforcement learning with a hierarchical approach that resolves each recursion depth sequentially. Both methods search for expressions satisfying differential equations directly through physics-aware losses similar to $\mathcal{R}(u)$. The efficiency of these discovery methods lies both in the way they sample trees and the way they compose solutions. For this reason, we consider two evaluation protocols for HD-TLGP which considers an Ansatz that is similar to SIGS (details are given in Appendix C.3.2). Protocol 1 is the same as in the original work: The algorithm is fed a dictionary of primitives, sin, cos, log, etc., to compose solutions. In Protocol 2, the algorithm is fed atoms from SIGS instead of primitive functions from a dictionary, see Appendix C.3 for more details. The objective is to show that naive search in the space of models does not work even when the algorithm combines atoms, which are mathematically proper subtrees. For SSDE, we tailor the dictionary of terms for each problem to contain only the primitives, meaning functions, and variables contained in the solution. For example, if $u(x) = \sin(\pi x) + \cos(\pi y)$ the dictionary contains only $\sin, x, y, \cos$ and integers. In this way, we show that for sophisticated search methods, if the dictionary considers primitives instead of atoms, the method cannot find an admissible solution when we consider complex problems. The complete primitive specifications appear in Appendix C.3.1. Neural baselines (PINNs (Raissi et al., 2019), FBPINNs (Moseley et al., 2023)) and numerical solvers (FEniCS; Alnæs et al., 2015, ; see details in Appendix C.4) are included for reference. For the Poisson-Gauss problems, no analytical solutions is available. Therefore, we assume the FEniCS with P4 elements on a $128 \times 128$ mesh as the ground truth. We perform a mesh convergence study to confirm the convergence of the solution at the chosen resolution. Complete problem specifications, analytical solutions, and discovered symbolic forms relevant to all the problems in the suite appear in Appendix C, accompanied by additional figures in Appendix C.7.

### 3.1 Experiments

Cross-validation on benchmarks from literature. First, we test SIGS on a subset of problems considered by Cao et al. and Wei et al. to show how combining the grammar-atoms approach together with adaptive search is more accurate than alternative approaches. For this purpose, we chose one-dimensional Poisson and Advection PDEs (HD-TLGP), and a two-dimensional Wave PDE (SSDE). We consider exactly the same problem specification, that is, the domain, boundary, and initial conditions, for the comparison. To make the methods comparable, we impose an Ansatz within SIGS that considers a different function per variable, e.g. $u(x, t) = g(x) \, f(t)$. The results are presented in Table 1. While the baseline approaches achieve high accuracy (HD-TLGP: $4.36 \times 10^{-4}$ for the Poisson, and $1.01 \times 10^{-2}$ for the Wave, SSDE: $1.04 \times 10^{-16}$), SIGS achieves exact solutions on all problems, as it contains $\pi$ as a symbol and does not approximate it numerically.

Table 1: We compare the accuracy, in terms of relative $L_2$ error against the exact solution, of SIGS and baselines on a collection of PDEs presented in the HD-TLGP and SSDE papers.

| Problem (method) | Original Method | SIGS (ours) |
|---|---|---|
| Poisson (HD-TLGP) | $4.36 \times 10^{-4}$ | exact solution |
| Advection (HD-TLGP) | $1.01 \times 10^{-2}$ | exact solution |
| Wave (SSDE) | $1.04 \times 10^{-16}$ | exact solution |

Comparison for Complex PDEs with known solutions. What makes the following collection of experiments complicated is not only that the solution contains many terms, but also that the method needs to find solutions that are very precise. For example, even if an algorithm discovers a solution that describes a viscous shock for the Burgers equation, slight imprecision in the location of the shock will result in a very large relative $L_2$ error against the exact solution. This phenomenon also holds true for the damped wave, as the problem is sensitive to the coefficients governing the diffusion time. For SIGS, we consider general solution Ansatze of the forms: We present the results in Table 2. We observe that

| | |
|---|---|
| Burgers: | $u(x,t) = a\psi(x,t)$ |
| Wave: | $u(x,y,t) = a\phi^1(x)\phi^2(y)T(t)$ |
| PG-2/3/4: | $u(x,y) = \sum_{i=1}^{K} a_i\psi_i(x,y)\varphi_i(x,y)$ |
| Advection: | $u(x,t) = a\psi(x,t)$ |
| Shallow Waters: | $\rho(x,y,t) = \psi^1(x,y,t)\psi^2(x,y,t)T(t)$ |
| | $u(x,y,t) = \psi^3(x,y,t)\rho(x,y,t)$ |
| | $v(x,y,t) = \psi^4(x,y,t)\rho(x,y,t)$ |

| | |
|---|---|
| Diffusion: | $u(x,y,t) = \sum_{i=1}^{4} a_i\phi_i(x)T_i(t)$ |
| Damping Wave: | $u(x,y,t) = a\psi(x,y,t)T(t)$ |
| KdV: | $u(x,t) = \sum_{i=0}^{2} a_i\psi_i(x,t)^k$ |
| Poisson: | $u(x,y) = a^1\phi^1(x) + a^2\phi^2(y)$ |

SIGS recovers exact analytical solutions, achieving machine precision on all problems with relative errors ranging from $6.64 \times 10^{-14}$ to $1.22 \times 10^{-13}$. The discovered expressions match analytical forms up to numerical precision, see Appendix C.5.

Both HD-TLGP and the SSDE methods fail to find a solution within the time budget that is accurate or close to the exact, see Appendix C.5. HD-TLGP in the case that we consider atoms in the dictionary, Protocol 1, returns relative $L_2$ errors in the range $2.04 - 423.40\%$, demonstrating the importance of the optimization method in discovering an accurate solution. Protocol 2 performs worse, with errors in the ranges of $35.68 - 178.77\%$ which shows how the results deteriorate without atoms. SSDE produces errors in the range of $45.62 - 5.87 \times 10^3\%$ even though the primitives are tailored for each problem. Requiring complex and precise solutions, translates to most of the loss landscape being flat with a very high value except for a small area where the loss is small. The failure of SSDE can almost certainly be attributed to the reinforcement learning algorithm failing to find this small region, as in the classic sparse-rewards problem. This result indicates how sophisticated optimizers fail completely when the dictionary does not contain elements that support aggressive exploration of the space of candidate models. Neural methods achieve moderate accuracy (2.56-6.09), while numerical solvers (FEniCS) present very accurate results. A visual comparison of the predictions of different methods are provided in Figure 3.

Table 2: Comparison of methods on PDEs with known analytical solutions. Reported are relative $L^2$ errors.

| PDE Problem | SIGS | HD-TLGP P1 | HD-TLGP P2 | SSDE | PINNs | FBPINNs | FEniCS |
|---|---|---|---|---|---|---|---|
| Burgers | $6.64 \times 10^{-14}$ | 2.04 | 35.68 | 45.62 | 6.09 | 28.26 | $8.69 \times 10^{-3}$ |
| Diffusion | $7.16 \times 10^{-13}$ | 33.34 | 79.73 | $5.87 \times 10^3$ | 2.56 | 55.54 | $2.26 \times 10^{-3}$ |
| Damping Wave | $1.22 \times 10^{-13}$ | 423.30 | 178.77 | $1.19 \times 10^3$ | 5.56 | 71.36 | $2.28 \times 10^{-2}$ |

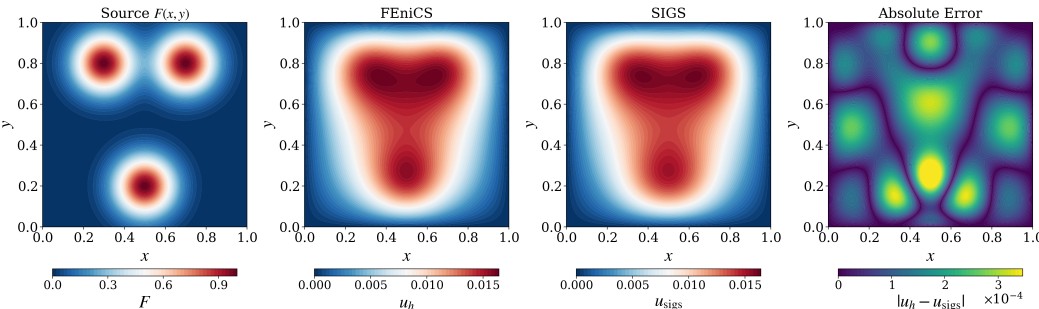

Figure 2: From left to right: source term $F(x, y)$ for the Poisson–Gauss problem; finite-element solution $u_h$ (FEniCS); symbolic approximation $u_{\text{sigs}}$ (SIGS); absolute error $|u_h - u_{\text{sigs}}|$

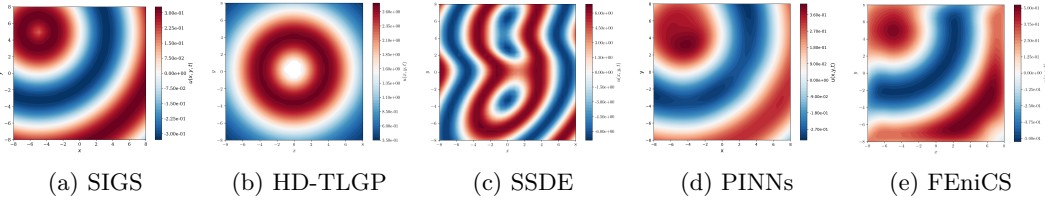

(a) SIGS      (b) HD-TLGP      (c) SSDE      (d) PINNs      (e) FEniCS

Figure 3: Comparison of different methods for solving the damped wave equation at $t = 2.5$. All methods show the same physical domain $x, y \in [-8, 8]$ with wave center at $(-5, 5)$. Parameters: $k = 0.5$, $\omega = 0.4$, $\alpha = 0.45$.

**Symbolic approximation without known solutions.** For the PDEs we considered so far, we manufactured, and therefore had access to, the exact solution. This allowed us to make educated guesses about the form of the Ansatz. In this example, we test how well SIGS and the baselines approximate the solution when an exact solution and a strong prior on the Ansatz does not exist. We investigate the Poisson equations with a Gaussian forcing term, which do not admit an exact analytical solution. For SIGS we choose the Ansatz as $u(x, y) = \sum_{j=1}^{N} \phi_j(x, y)\psi_j(x, y)$, with $j \in [3, 4, 8]$ for PG2, PG3, PG4, respectively. Here $\psi_j(x, y)$ are eigenfunction of the elliptic operator $\psi(x, y) = \sin(\pi x)\sin(\pi y)$ that impose the homogenous Dirichlet boundary conditions, and $\phi_j$ is randomly sampled from the available atoms. We present the results for all the methods in Table 3. SIGS achieves $1 - 3\%$ relative $L_2$ errors with improving accuracy as complexity increases and number of modes increases (from $2.66\%$ for 2 Gaussians to $1.05\%$ for 4 Gaussians), which suggests that SIGS correctly leverages the superposition of the Gaussian atoms. HD-TLGP failed to find a solution withing the time budget, and produced NaNs in our tests, probably due to numerical instabilities for Protocol 1 while for Protocol 2 generates errors exceeding $10^7$. SSDE achieves errors in the range 58-70%, which translates to missing the precise superposition of Gaussians.

The results support that successfully discovering the solution of complex PDEs requires a combination of structured atoms and a global(to explore)-local(to discover precise arrangements) optimization algorithm. Moreover, Table 4 shows how the approach of SIGS is practically viable as the solutions are found in seconds to minutes.

### 3.2 Ablation Studies

**Atoms vs. Primitives.** We previously stated that considering atoms or combining primitives can have a decisive effect in finding PDE solutions. We test this hypothesis by considering the damped wave PDE with the Ansatz $u(x, y, t) = \sum_{j=1}^{N} \psi_j(x, y, t)\phi_j(x)T_j(t)$ and instead of considering atoms, we sample $\phi, T, \psi$ with uniform probability over rules of the same grammar as before. If we sample a function with the correct arity, e.g. $\psi$ containing

Table 3: Approximation on Poisson-Gauss problems without analytical solutions. Relative L2 errors against FEniCS references.

| Problem | SIGS | HD-TLGP P1 | HD-TLGP P2 | SSDE |
|---------|------|------------|------------|------|
| PG-2 | 2.66 | 200.9 | 98.94 | 69.29 |
| PG-3 | 1.54 | NaN | $5.61 \times 10^7$ | 69.64 |
| PG-4 | 1.05 | NaN | $5.45 \times 10^7$ | 58.70 |

Table 4: Wall-clock time (CPU). SIGS reports time-to-$\varepsilon$; others report time-at-termination. Notation: $\checkmark$ reached $\varepsilon$; $\dagger$ hit budget / failed to reach $\varepsilon$. HD-TLGP budget: 20 generations, SSDE budget: 25 generations.

| Problem | SIGS$^\checkmark$ | HD-TLGP P1$^\dagger$ | HD-TLGP P2$^\dagger$ | SSDE$^\dagger$ | PINNs$^\dagger$ | FEniCS$^\checkmark$ |
|---------|------|---------|---------|------|-------|--------|
| Burgers | 13.5 sec | > 239 m36 sec | > 200 m57 sec | > 6 m34 sec | 8.8 sec | 2.2 sec |
| Diffusion | 39.2 sec | > 192 m41 sec | > 181 m49 sec | > 8 m6 sec | 2 m2 sec | 1.4 sec |
| Damping Wave | 30.2 sec | > 88 m40 sec | > 37 m8 sec | > 6 m19 sec | 29.5 sec | 3.4 sec |
| PG-2 | 1 m30 sec | > 182 m25 sec | > 90 m43 sec | > 11 m45 sec | n/a | 19.3 sec |
| PG-3 | 1 m51 sec | > 120 m7 sec | > 97 m16 sec | > 11 m4 sec | n/a | 6.9 sec |
| PG-4 | 1 m23 sec | > 145 m32 sec | > 80 m50 sec | > 12 m32 sec | n/a | 3.4 sec |

$x, y, t$, we consider the function admissible. We sampled $50,000$ functions, out of which only 133 were admissible, which means that it would be impossible to start and adaptive optimization procedure due to the admissible sampling rate being so low. Moreover, the admissible functions with the lowest loss provides $\mathcal{R}(u) \approx 366\%$ relative $L^2$ error to the exact solution. This clearly demonstrates the necessity of atoms, and thus the embedding, to the whole process.

TGVAE vs. vanilla GVAE. We measure sampling efficiency using a race-to-$k$-valid benchmark, which counts the total attempts required to generate $k = 1000$ syntactically valid expressions by sampling random latent vectors $z \in Z$. To assess the quality of the latent space, the latent vectors are decoded to analytical expressions $w$, which are rejected if they fail to meet the grammar-based and mathematical consistency checks in Section A.1.3. We expect both VAEs to be more stable in regions surrounding $z_i = \mathcal{E}(w_i)$. Hence, we only consider latent vectors $z$ with a minimal distance of $\tau = 0.8$ away from any training sample $z_i$ in terms of the Mahalanobis norm (App. C.6). We sample $15,000$ admissible latent vectors and split them into ten disjoint sets. We provide each set to the GVAE and the TGVAE and count the total decode attempts required to obtain $1,000$ valid expressions. The GVAE required $1486.2 \pm 19.5$ attempts, while our topology-regularized TGVAE needed $1433.2 \pm 27.3$ attempts, a $3.56\% \pm 1.81$ relative reduction. This indicates that geometric regularization (hull loss, persistent homology, smoothness penalties) yields a more navigable latent space with fewer degenerate decodes.

## 4 Discussion and Conclusion

Discussion. This work advances solution discovery for PDEs by demonstrating that grammar-guided neuro-symbolic methods can reliably and efficiently recover analytical solutions. SIGS consistently improves the state-of-the-art, both in accuracy and speed, often by several orders of magnitude. Its success stems from two complementary design choices: (i) constructing a latent manifold of solution components, which enables smooth and efficient exploration of admissible expressions; and (ii) employing a hierarchical Ansatz+atom approach that reduces search complexity by structuring the solution space into manageable placeholders, later refined into concrete symbolic elements. This is in contrast to the baselines explored in this work, which do not address the combinatorial explosion inherent in symbolic solution discovery. HD-TLGP (Cao et al., 2024) transfers structures from one-

dimensional solutions to higher dimensions, but still relies on stochastic recombination of primitives, which quickly becomes intractable as complexity grows. SSDE (Wei et al., 2024) instead uses reinforcement learning to guide the construction of candidate solutions, but its flat search space remains prohibitively large without strong priors. As our experiments show, both methods degrade sharply when such priors are absent. In contrast, the hierarchical Ansatz+atom design of SIGS separates global structure from local symbolic details, making tractable what would otherwise be an unmanageable search. In this way, SIGS not only advances but fundamentally redefines the state-of-the-art for solution discovery. Beyond these empirical gains, we view SIGS as part of a broader shift toward neuro-symbolic foundation models for PDEs. Current foundation approaches (Herde et al., 2024; Hao et al., 2024; Sun et al., 2024; Alkin et al., 2024; Shen et al., 2024) rely on extensive pretraining and often serve as black-box predictors for downstream tasks. In contrast, SIGS requires only a one-time pretraining step to construct its manifold, after which it transfers directly to new problems without retraining. Moreover, it produces analytical expressions that incorporate physical priors (e.g., eigenfunctions), yielding interpretable solutions rather than opaque approximations. This suggests that grammar-based neuro-symbolic models could complement or even provide an alternative to purely data-driven foundation models in scientific computing.

Limitations.  Despite these contributions, SIGS faces two main limitations. First, scalability to complex engineering problems remains challenging. PDEs involving discontinuities, multiscale structure, or turbulence may require grammars enriched with special functions that cannot be easily decomposed into smaller atoms, or long expressions that increase search complexity. Hybrid approaches that combine symbolic structures with numerical bases (e.g., POD-derived eigenfunctions, or Neural Operators) may provide a path forward, particularly for multiscale phenomena, as well as for problems with irregular geometries or boundary conditions. Second, the framework depends on the joint design of grammar, Ansatz, and latent space. A richer Ansatz can offset a simpler grammar, while a more expressive grammar requires larger latent spaces and more sophisticated optimization. Currently, the Ansatz still reflects human expert choices. This can be advantageous in domains with strong theoretical foundations (e.g., Burgers or Poisson equations), but limits applicability in less understood settings. A promising direction is to leverage large language models (e.g., Romera-Paredes et al., 2024) to automate Ansatz construction, learning general solution structures directly from governing equations.

Conclusion.  In this work, we introduced the Symbolic Iterative Grammar Solver (SIGS), a grammar-guided neuro-symbolic framework for discovering analytical solutions to differential equations. By unifying classical compositional methods with modern latent-space optimization through the Topological Grammar VAE, SIGS systematically explores the space of admissible solutions, enabling efficient search and refinement of closed-form expressions. Our approach achieves state-of-the-art performance on recent benchmarks, recovering exact solutions when available, and producing interpretable symbolic approximations for PDEs without known closed form solutions. These results highlight the potential of grammar-based neuro-symbolic methods as a scalable and interpretable alternative to purely data-driven approaches, opening new directions for automated solution discovery in scientific computing.

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

# Appendix

## A  Grammar and GVAE

### A.1  Library Generation

We construct symbolic component libraries that serve as input vocabulary for the TGVAE architecture in the discovery of DE solutions. The fundamental challenge in symbolic regression for DEs is that naive search over arbitrary mathematical expressions is computationally intractable and often produces physically meaningless results. Our library generation approach addresses this by creating curated collections of symbolic components that correspond directly to different atoms that compose solutions of a wide range of differential operators. The library consists of individual analytical building blocks rather than complete DE solutions. Each component represents a fundamental mathematical pattern such as spatial eigenfunctions $\sin(k\pi x)$, temporal factor $e^{-\lambda t}$, or their separable product $\sin(k\pi x)e^{-\lambda t}$ that naturally arises in the decomposition of certain operator. This modular design enables the neural architecture to learn complex solution structures through principled combinations of mathematically meaningful primitives, rather than searching over the vast space of arbitrary symbolic expressions. The key insight driving our approach is that different DE operators admit characteristic families of atoms that reflect their underlying mathematical structure. This principled approach transforms the solution discovery task from an open-ended search problem into a structured exploration of mathematically principled solution components.

### A.1.1  Atom Generation

The atoms of the library represent temporal factors, eigenfunctions of operators, expressions that describe dynamics of interest, and random compositions.

Assume a bounded Lipschitz domain $\Omega \subset \mathbb{R}^d$ and a linear second order operator with homogeneous boundary conditions, that is self-adjoint and non-negative, e.g. $\mathcal{S} = -\Delta$ with

Table 5: Closed-form Laplacian eigenfamilies $A = -\Delta$ on common domains. Here $\Omega$ is the spatial domain, $\phi$ the eigenfunction, and $\mu$ the eigenvalue in $A\phi = \mu\phi$. For Neumann on boxes the constant mode has $\mu_0 = 0$.

| Domain / BC | Eigenfunction $\phi$ | Eigenvalue $\mu$ | Index set |
|---|---|---|---|
| Rectangles / periodic boxes $\Omega = \prod_{d=1}^{D}[0, L_d]$ | | | |
| Periodic (torus) | $\phi_{\mathbf{k}}(x) = \exp\left(i\,2\pi\sum_{d=1}^{D}\frac{k_d}{L_d}\,x_d\right)$ | $\mu_{\mathbf{k}} = 4\pi^2\sum_{d=1}^{D}\frac{k_d^2}{L_d^2}$ | $\mathbf{k} \in \mathbb{Z}^D$ |
| Dirichlet | $\phi_{\mathbf{k}}(x) = \prod_{d=1}^{D}\sin\left(\frac{k_d\pi}{L_d}\,x_d\right)$ | $\mu_{\mathbf{k}} = \pi^2\sum_{d=1}^{D}\frac{k_d^2}{L_d^2}$ | $k_d \in \mathbb{N}$ |
| Neumann | $\phi_{\mathbf{k}}(x) = \prod_{d=1}^{D}\cos\left(\frac{k_d\pi}{L_d}\,x_d\right)$ | $\mu_{\mathbf{k}} = \pi^2\sum_{d=1}^{D}\frac{k_d^2}{L_d^2}$ | $k_d \in \mathbb{N}_0$ |
| | | (includes constant mode $\mathbf{k} = \mathbf{0}$ with $\mu_{\mathbf{0}} = 0$) | |
| Disks and balls (Dirichlet conditions) | | | |
| 2-D disk, radius $R$ | $\phi_{mn}(r, \theta) = J_m\left(\frac{j_{mn}}{R}\,r\right) \times \begin{cases}\cos(m\theta), \\ \sin(m\theta),\end{cases}$ | $\mu_{mn} = \frac{j_{mn}^2}{R^2}$ | $m \in \mathbb{Z}_{\geq 0},\ n \in \mathbb{N}$ |
| 3-D ball, radius $R$ | $\phi_{\ell mn}(r, \theta, \varphi) = j_\ell\left(\frac{\alpha_{\ell n}}{R}\,r\right) Y_{\ell m}(\theta, \varphi)$ | $\mu_{\ell n} = \frac{\alpha_{\ell n}^2}{R^2}$ | $\ell \in \mathbb{Z}_{\geq 0},\ |m| \leq \ell,\ n \in \mathbb{N}$ |

Dirichlet, Neumann, or periodic boundary conditions. Then there exists an $L^2$-orthonormal eigen basis $\{\phi\}_{j\geq 1} \subset L^2(\Omega)$ and eigenvalues $\{\mu\}_{j\geq 1} \subset [0, \infty)$ with $\mathcal{S}\phi_j = \mu_j\phi_j$. For example, the Diffusion equation $u_t - \kappa\Delta u = 0$ has the scalar ODE $T'(t) + \kappa\mu_j T_j(t) = 0$ as the temporal rule, with solution $T_j(t) = e^{-\kappa\mu t}$. For the spatial rule, we can consider the rectangle box $\Omega = \prod_{d=1}^{D}[0, L_d]$ with Dirichlet boundary conditions. Given indices $k \in \mathbb{N}^D$, the grammar produces:

$$\phi_k(x) = \prod_{d=1}^{D}\sin(k_d\pi x_d/L_d), \qquad \mu_k = \pi^2\sum_{d=1}^{D}\frac{k_d^2}{L_d^2}.$$

Composing together with the temporal model, we get $u(x, t) = \sum_k a_k e^{\kappa\mu t}\phi_k(x)$, which solves $u_t + \kappa\Delta u = 0$ exactly. The amplitudes $a_j$ are drawn from prior $a_j \sim \mathcal{N}(0, \sigma^2\rho)$ where $\rho$ decays exponentially to control the regularity or the spectrum. This construction generalizes for multiple classes of known operators, see Table 6. For constant coefficient operators on separable geometries we have explicit $\{\phi_k, \mu_k\}$ eigenfunctions as shown in Table 5.

As we discussed, the grammar can also produce expressions that describe dynamics of interest such as viscous shocks $\tanh\left(\frac{(u_l - u_r)(x - st - x_0)}{4\nu}\right)$, transport $g(kx - \omega t)$, heat kernels $\frac{1}{(4\pi kt)^{d/2}}\exp\left(\frac{\|x\|^2}{4kt}\right)$, Gaussian bumps $\exp\left(\frac{\|x - x_0\|^2}{2k}\right)$ and others. Moreover, atoms are polynomials, and combinations of the above.

### A.1.2 Formal Grammar Specifics

The grammar $\mathcal{G} = (V, \Sigma, R, S)$ contains 51 production rules that provide the complete symbolic vocabulary for DE eigenfunction families used in the experiments herein. The grammar systematically generates expressions through the application of production rules $R$, including

- compositional rules $S \to S + T \mid S \times T \mid S/T \mid S - T \mid T \mid -T$ that build complex mathematical structures,
- function application rules $T \to (S) \mid (S)^2 \mid \sin(S) \mid \exp(S) \mid \log(S) \mid \cos(S) \mid \sqrt{S} \mid \tanh(S)$ that provide the transcendental functions essential for eigenfunction representation,
- variable and monomial specifications $T \to T^D \mid \pi \mid x \mid y \mid t \mid x^2 \mid x^3 \mid y^2 \mid y^3$ that capture spatial and temporal dependencies,

Table 6: Modal time factors for common PDE families. Here $A$ is a nonnegative self-adjoint spatial operator with eigenpairs $A\phi_\mathbf{k} = \mu_\mathbf{k}\phi_\mathbf{k}$, $\mu_\mathbf{k} \geq 0$. Projecting the PDE onto $\phi_\mathbf{k}$ yields the scalar ODE for $T_\mathbf{k}(t)$ shown in the middle column and its solution in the right column.

| PDE family | Modal ODE (after projection) | Temporal factor $T_\mathbf{k}(t)$ |
|---|---|---|
| Heat / diffusion | $T'_\mathbf{k} + \kappa\,\mu_\mathbf{k}\,T_\mathbf{k} = 0$ | $e^{-\kappa\,\mu_\mathbf{k}\,t}$ |
| Stokes (divergence-free) | $T'_\mathbf{k} + \nu\,\mu_\mathbf{k}\,T_\mathbf{k} = 0$ | $e^{-\nu\,\mu_\mathbf{k}\,t}$ |
| Undamped wave | $T''_\mathbf{k} + c^2\,\mu_\mathbf{k}\,T_\mathbf{k} = 0$ | $\cos\left(c\sqrt{\mu_\mathbf{k}}\,t\right)$ or $\sin\left(c\sqrt{\mu_\mathbf{k}}\,t\right)$ |
| Damped wave (telegraph) | $T''_\mathbf{k} + 2\gamma\,T'_\mathbf{k} + c^2\,\mu_\mathbf{k}\,T_\mathbf{k} = 0$ | Underdamped $c^2\mu_\mathbf{k} > \gamma^2$: $e^{-\gamma t}\left(C_1\cos(\omega_\mathbf{k}t) + C_2\sin(\omega_\mathbf{k}t)\right)$, $\omega_\mathbf{k} = \sqrt{c^2\mu_\mathbf{k} - \gamma^2}$. Critical $c^2\mu_\mathbf{k} = \gamma^2$: $e^{-\gamma t}(C_1 + C_2 t)$. Overdamped $c^2\mu_\mathbf{k} < \gamma^2$: $C_1 e^{-(\gamma - \sqrt{\gamma^2 - c^2\mu_\mathbf{k}})\,t} + C_2 e^{-(\gamma + \sqrt{\gamma^2 - c^2\mu_\mathbf{k}})\,t}$. |
| Biharmonic diffusion | $T'_\mathbf{k} + \kappa\,\mu_\mathbf{k}^2\,T_\mathbf{k} = 0$ | $e^{-\kappa\,\mu_\mathbf{k}^2\,t}$ |
| Damped plate/beam | $T''_\mathbf{k} + 2\gamma\,T'_\mathbf{k} + c^2\,\mu_\mathbf{k}^2\,T_\mathbf{k} = 0$ | As for damped wave, with $c^2\mu_\mathbf{k}$ replaced by $c^2\mu_\mathbf{k}^2$ |
| Klein–Gordon (damped) | $T''_\mathbf{k} + 2\gamma\,T'_\mathbf{k} + (c^2\mu_\mathbf{k} + m^2)\,T_\mathbf{k} = 0$ | As for damped wave, with $c^2\mu_\mathbf{k}$ replaced by $c^2\mu_\mathbf{k} + m^2$ |
| Fractional diffusion | $T'_\mathbf{k} + \kappa\,\mu_\mathbf{k}^s\,T_\mathbf{k} = 0, \ s \in (0, 1]$ | $e^{-\kappa\,\mu_\mathbf{k}^s\,t}$ |
| Reaction–diffusion (linear part) | $T'_\mathbf{k} + (\kappa\,\mu_\mathbf{k} - \rho)\,T_\mathbf{k} = 0$ | $e^{-(\kappa\,\mu_\mathbf{k} - \rho)\,t}$ |
| Allen–Cahn (linearized) | $T'_\mathbf{k} + (\kappa\,\mu_\mathbf{k} - \alpha)\,T_\mathbf{k} = 0$ | $e^{-(\kappa\,\mu_\mathbf{k} - \alpha)\,t}$ |
| Cahn–Hilliard (linearized) | $T'_\mathbf{k} + M\,\mu_\mathbf{k}(\mu_\mathbf{k} + \sigma)\,T_\mathbf{k} = 0$ | $e^{-M\,\mu_\mathbf{k}(\mu_\mathbf{k} + \sigma)\,t}$ |
| Kuramoto–Sivashinsky (linearized) | $T'_\mathbf{k} + (\nu\,\mu_\mathbf{k}^2 - \kappa\,\mu_\mathbf{k})\,T_\mathbf{k} = 0$ | $e^{-(\nu\,\mu_\mathbf{k}^2 - \kappa\,\mu_\mathbf{k})t}$ |
| Maxwell in a PEC cavity | $\varepsilon\,T''_\mathbf{k} + \sigma\,T'_\mathbf{k} + c^2\,\mu_\mathbf{k}\,T_\mathbf{k} = 0$ | Vector modes; as damped wave (if $\sigma = 0$: $\cos/\sin$ with $\omega_\mathbf{k} = c\sqrt{\mu_\mathbf{k}}$) |
| Isotropic linear elasticity | $T''_\mathbf{k} + \omega_{B,\mathbf{k}}^2\,T_\mathbf{k} = 0, \ B \in \{T, L\}$ | Two branches: $\omega_{T,\mathbf{k}} = c_T\sqrt{\mu_\mathbf{k}}, \ \omega_{L,\mathbf{k}} = c_L\sqrt{\mu_\mathbf{k}}; \ T = \cos/\sin$ |

- numeric construction $T \to D \mid D.D \mid -D \mid -D.D \mid TD$ with digit generation $D \to D0 \mid D1 \mid \ldots \mid D9 \mid 0 \mid 1 \mid \ldots \mid 9$,
- and scientific notation $D \to$ e-1 $\mid$ e-2 $\mid$ e-3 $\mid$ e-4 for numerical stability across multiple scales.

The terminal alphabet hence encompasses

$$\Sigma = \{x, y, t, \pi\} \cup \{\sin, \cos, \exp, \log, \tanh, \sqrt{\cdot}, (,)\} \cup \{+, -, \times, /, \widehat{\ }\} \cup \{0, 1, \ldots, 9\} \cup \{\text{e-1}, \ldots, \text{e-4}\}.$$

### A.1.3 Mathematical checks on generated functions

Each generated component undergoes rigorous symbolic validation to guarantee syntactic and mathematical sense of the generated expressions. In case a generated expression does not satisfy the checks, it is rejected, and a new one is generated.

**Syntactic requirements.** We enforce strict variable presence requirements where ODE problems must contain $\{x\} \subseteq \text{Vars}(u)$, spatial DE problems require $\{x, y\} \subseteq \text{Vars}(u)$, and spatiotemporal problems need $\{x, t\} \subseteq \text{Vars}(u)$ or $\{x, y, t\} \subseteq \text{Vars}(u)$. Function domain restrictions prevent undefined operations through logarithmic function constraints $\log(f) \Rightarrow f > 0$ on $\overline{\Omega}$, square root function requirements $\sqrt{f} \Rightarrow f \geq 0$ for spatial components, and division safety ensuring denominators remain bounded away from zero. To ensure symbolic rather than constant generation, we forbid purely numeric arguments to transcendental functions so that $\sin(\alpha), \cos(\alpha), \exp(\alpha), \log(\alpha) \notin \text{Lang}(\mathcal{G}_\mathbb{D})$ for $\alpha \in \mathbb{R}$. Integer powers are restricted to degree $\leq 3$ to preserve $H^1(\Omega)$ membership on bounded domains, ensuring that for polynomials $u = \sum_{|\beta| \leq 3} c_\beta x^\beta$ we have $\|u\|_{H^1(\Omega)} < \infty$ when $\Omega$ is bounded. Function compositions are validated for smoothness preservation where admissible functions $f \in \{\sin, \cos, \exp, \tanh\}$ applied to arguments $g$ with controlled growth maintain $C^\infty$ regularity on bounded domains.

**Boundary conditions.** For boundary condition compatibility, homogeneous Dirichlet conditions $u|_{\partial\Omega} = 0$ are enforced by multiplying spatial components with boundary-vanishing envelopes such as $\psi_{\text{env}}(x, y) = \sin\left(\frac{\pi x}{L_x}\right)\sin\left(\frac{\pi y}{L_y}\right)$ for rectangular domains, ensuring $u_{\text{modified}} \in H_0^1(\Omega)$. Neumann compatibility for problems requiring $\frac{\partial u}{\partial n}|_{\partial\Omega} = 0$ uses cosine spatial modes that naturally satisfy zero normal derivative conditions.

Table 7: GVAE architecture summary. LN: LayerNorm over $[C, L]$; all linear/conv layers use bias=False unless noted.

| Block | Layer | Dims / Kernel / Len | Act/Norm |
|---|---|---|---|
| Input | Tensor | $C{=}53$, $L{=}72$ | – |
| Encoder | Conv1D | $53 \to 64$, $k{=}2$, $L$: $72 \to 71$ | ELU |
| | Conv1D | $64 \to 128$, $k{=}3$, $L$: $71 \to 69$ | ELU |
| | Conv1D | $128 \to 256$, $k{=}4$, $L$: $69 \to 66$ | ELU |
| | Linear | $256{\times}66{=}16,896 \to 256$ | ELU |
| | Heads | $256 \to 32$ ($\mu$), $256 \to 32$ ($\log \sigma^2$) | – |
| Decoder (positional) | Linear | $32 \to 512$ | ELU |
| | GRU | input$= 512$, hidden$= 512$, layers$= 1$ | – |
| | TimeDense | $512 \to 53$ (per position, $L{=}72$) | – |
| Latent dim / samples | | $z$-dim $= 32$; decoder samples per input $= 1$ | |

Table 8: Lightning module summary (train mode).

| Name | Type | Params | Mode |
|---|---|---|---|
| model | GrammarVAE | 6.1 M | train |
| Total trainable params | | 6.1 M (24.495 MB) | |

**Constants.** Numerical stability is maintained through exponential scaling control using scientific notation coefficients with mantissa $m \in [0.001, 999]$ and exponent $e \in [-4, 4]$ to prevent overflow and underflow. Floating point precision involves rounding numeric literals to 3 decimal places for most components and 6 decimal places for wave modes, converting to rational representations when possible to avoid precision degradation.

**Uniqueness of expressions.** Expression canonization includes converting fractional powers to $\sqrt{\cdot}$ notation when $p = 1/2$, transforming reciprocal notation $x^{-1} \mapsto 1/x$, and simplifying coefficients such as $(2 \times 3)x \mapsto 6x$. Uniqueness is enforced through syntactic equivalence classes where we define $s \sim s'$ if their canonized forms coincide after symbolic simplification, maintaining exactly one representative per equivalence class $[s] \in \mathcal{L}(\mathcal{G}_{\mathbb{D}})/\sim$ using a global hash table that tracks all generated canonical forms.

### A.2 GVAE Model and Training Details

We employ a Grammar Variational Autoencoder (GVAE) that maps one-hot sequences of CFG production rules to a continuous latent space and decodes back to valid rule sequences. Inputs are $x \in \mathbb{R}^{B \times C \times L}$ with $C{=}53$ rules and $L{=}72$ time steps (dataset shape $N \times C \times L = \mathbf{23,682} \times 53 \times 72$); targets are $y = \arg\max_c x \in \{0, \dots, 52\}^{B \times L}$.

**Architecture.** The encoder stacks three valid (no-pad) 1D convolutions with ELU activations, followed by a linear layer and two bias-free heads producing $\mu, \log \sigma^2 \in \mathbb{R}^{32}$. The decoder is non-autoregressive ("positional"): it lifts $z \in \mathbb{R}^{32}$ to a hidden state, runs a GRU across positions, then applies a time-distributed linear projection to rule logits. Shapes and hyperparameters are summarized in Table 7. Lightning reports 6.1 M trainable parameters (model size 24.495 MB; see Table 8).

**Losses and regularization.** The objective is

$$\mathcal{L} = \mathcal{L}_{\text{recon}} + \beta(t) \underbrace{\text{KL}\left(q(z|x) \,\|\, p(z)\right)}_{\text{latent}} + \gamma(e)\left(0.8\,\mathcal{L}_{\text{Hull}} + 0.8\,\mathcal{L}_{\text{ph}} + 10^{-4}\,\mathcal{L}_{\text{smooth}}\right),$$

with $\mathcal{L}_{\text{recon}}$ being equal to cross entropy loss of $\mathcal{L}_{\text{recon}}(\overline{\text{logits}}, y)$, where $\overline{\text{logits}}$ being the mean over decoder samples (here $= 1$). The KL weight uses a linear warmup $\beta(t) =$

Table 9: Training hyperparameters and $\mathcal{L}_{\text{topo}}$ term weights. ReduceLROnPlateau monitors the balanced ELBO.

| Item | Value | Details |
|---|---|---|
| Optimizer | AdamW | lr $= 3 \times 10^{-4}$, weight decay $= 10^{-5}$ |
| Batch size (train/val) | 64 / 64 | 4 dataloader workers |
| Precision | 16-mixed (AMP) | global grad clip $= 1.0$ |
| Scheduler | ReduceLROnPlateau | factor $= 0.2$, patience $= 5$ |
| Epochs / Early stop | 200 / 10 | monitor (validation's set ELBO) |
| KL warmup $\beta(t)$ | to 1.0 by 7000 updates | $\beta_0 = 0.01$ |
| Topo loss activation | at val-acc $\geq 20\%$ | ramp $\gamma$ over 5 epochs |
| Topo loss schedule | train/val every 50 / 12 | sparse to limit cost |
| Topo loss weights | $w_{\text{Hull}}=0.8$, $w_{\text{ph}}=0.8$ | $w_{\text{smooth}}=10^{-4}$ |
| Ph settings | max points $= 24$, max dim $= 1$ | Rips on CPU, scales $\{0.10, 0.50\}$ |
| Hull directions | $K=256$ | fixed $U_K \subset \mathbb{S}^{d-1}$ |

$\beta_0 + (1 - \beta_0) \min\left(\frac{t}{7000}, 1\right)$ with $\beta_0 = 0.01$. The geometric topological block activates once validation sequence-exact accuracy reaches 20%, then ramps $\gamma(e)$ from 0 to 1 over 5 epochs. Topological loss' terms (Hull, ph@scale on CPU, smooth) are computed in fp32 and scheduled sparsely (train every 50 steps and validate every 12 batches). Upon $\mathcal{L}_{\text{topo}}$ activation, the LR scheduler's best-score baseline is reset to the new balanced ELBO.

**Data and splits.** We train on an HDF5 corpus of one-hot sequences under a typed CFG. Random split with seed 42 into train/val/test of 70%/20%/10% yields the counts in Table 10.

Table 10: Dataset and splits for GVAE training ($C=53$, $L=72$).

| Split | # Sequences |
|---|---|
| Train | 16,578 |
| Val | 4,736 |
| Test | 2,368 |

**Environment and software.** Experiments ran on an NVIDIA RTX 5080 Laptop GPU (16 GB VRAM). Key versions are summarized in Table 11.

Table 11: Compute environment.

| Component | Spec |
|---|---|
| CPU | Intel Core Ultra 9 275HX, 24C/24T @ 2.7 GHz |
| RAM | 32 GB |
| GPU | NVIDIA GeForce RTX 5080 Laptop GPU 16 GB VRAM) |
| Python | 3.10.18 |
| PyTorch / Lightning | 2.7.1+cu128 / 2.5.2 |
| CUDA / cuDNN | 12.8 / 90800 |

**Training procedure and metrics.** We train on a single GPU with AMP and gradient clipping. The primary validation metric is the val ELBO, combining CE, KL (with warmup), and $\mathcal{L}_{\text{topo}}$ (when enabled). We also log CE, KL, ELBO variants, Topo loss components, and sequence-exact accuracy. Early stopping halts after 10 epochs without improvement in val elbo full.

**Runtime observations.** Before the activation of $\mathcal{L}_{\text{topo}}$, epochs take a few seconds. Around the activation point (48th epoch), training duration is $\sim 3\,\text{s}$, however at the (50th epoch) that $\mathcal{L}_{\text{topo}}$ starts getting calculated training duration raises to $\sim 262\,\text{s}$. This spike is expected. The $\mathcal{L}_{\text{topo}}$ builds a Vietoris–Rips complex and computes persistent homology on the CPU. Constructing distance matrices and boundary operators. Differentiating through

them (ph/smooth-Hessian), dominates wall-clock time and introduces CPU↔GPU synchronization overhead.

Decoding & evaluation settings.  Non-autoregressive (positional) decoding with one latent sample, max length 72, vocabulary size 53. When applicable, a CFG mask enforces per-step validity.  Report sequence-exact accuracy, validity rate, CE, KL, and ELBO on validation/test sets.

## A.3    Geometry Regularization

Here, we provide details on the additional loss terms added to the GVAE loss to form the Topological GVAE (TGVAE).

Convex hull loss.  Let $z \in \mathbb{R}^d$ be a latent vector and $Z = \{z_i\}_{i=1}^B$ the current batch. We maintain a reservoir of latent vectors $R_t \subset \mathcal{R}^d$ using a farthest point insertion with distance $\delta > 0$ (Gonzalez, 1985). At each iteration $t$, we freeze $R_t^{\text{prev}}$, compute losses, and update the reservoir only if a latent is not $\delta$ close to any $z$ seen thus far. Considering fixed unit directions $\{n_k\}_{k=1}^K \in \mathbb{S}^{d-1}$ and a support function $h_Z(n) = \sup_{z \in Z}\langle n, z\rangle$ (Rockafellar, 2015) we define:

$$\mathcal{L}_{\text{Hull}}(R_t^{\text{prev}}, Z_t) = \frac{1}{BK}\sum_{i=1}^B \sum_{k=1}^K [\langle n_k, z_i\rangle - h_{R_t^{\text{prev}}}(n_k)]^2+, \quad \text{where } [\cdot]+ = \max\{\cdot, 0\}.$$

If $\mathcal{L}_{\text{Hull}} = 0$ then every $z_i$ lies in the an explicit convex enclosure of the frozen reservoir $\cap_{d=1}^D \{z : \langle n_m, z\rangle \le h_{R_t^{\text{prev}}}\}$. Inside those bounds, we remove small spurious loops and holes at the working resolution set by $\delta$.

Persistent homology loss.  Let $P_t = R_t^{\text{prev}} \cup Z_t$, and $V_k(P)$ the persistence diagram of Vietoris-Rips homology across the scale $\epsilon$ (Edelsbrunner & Harer, 2010). We set the working radius $r = \sqrt{2}\delta$ using the clamped lifetime $\ell_r(b, d) = \max\{0, \min(d, r) - \min(b, r)\}$, and define the persistent homology loss:

$$\mathcal{L}_{\text{ph}}(P_t) = \sum_{(b,d)\in V_1(P_t)} \ell_r(b, d)^2 + a_0 \sum_{(b,d)\in V_0(P_t)} \ell_r(b_0, d_0)^2,$$

which suppresses small loops $H_1$ and micro-clusters $H_0$ at resolution $r$.

Smoothing loss.  We also penalize large Hessian energies to prevent sharp decode curvatures, to ensure that small moves in the latent space produce stable changes. For decoder $\mathcal{D}_\theta : \mathbb{R}^d \to \mathbb{R}^M$, we set $f(z) = \mathbf{1}^\top \mathcal{D}_\theta$ and $H_f(z) = \nabla^2 f(z)$, with $v \sim \mathcal{N}(0, I_d)$ we define:

$$\mathcal{L}_{\text{smooth}} = \mathbb{E}_{z \in Z_t}\|H_f(z)v\|_2^2,$$

which is estimated using the probes as in Hutchinson (1989).

## B    Iterative Search and Refinement

This appendix details the two-stage search procedure outlined in Section 2. We provide the mathematical formulation and implementation details for both structure discovery and coefficient refinement.

### B.1    Notation and Setup

When using SIGS on a specific problem, the user may specify a structural Ansatz $F$ consisting of the compositional nature the proposed solutions should follow. For example, one could specify spatio-temporal separability as $u(x, t) = \sum_{j=1}^K \phi_j(x)\psi_j(t)$, leaving $K$ spatial and $K$ temporal functions, overall $L = 2K$ components, to be chosen by SIGS. We denote by $A : \{\mathcal{L}(\mathcal{G})\}^L \to \mathcal{L}(\mathcal{G})$ the assembly map that composes the single components into the

final solution following the Ansatz, and refer to the component indices as $\mathbb{N}_L = \{1, \ldots, L\}$. Recall the TGVAE encoder $\mathcal{E} : \mathcal{L}(\mathcal{G}) \mapsto Z$ and decoder $\mathcal{D} : Z \to \mathcal{L}(\mathcal{G})$ that assign strings (functions) from the language to latent vectors in $Z \subset \mathbb{R}^d$. For a given component $c \in \mathbb{N}_L$, the Ansatz specifies the variables $c$ should contain. This prior knowledge is incorporated by filtering the library to contain only valid component candidates $\mathcal{L}^{(c)}$ and restricting $Z$ to the component-specific latent set $Z^{(c)} = \{z_i^{(c)}\}_{i=1}^{N_c}$ by applying the encoder $\mathcal{E}$ to $\mathcal{L}^{(c)}$.

B.2   Target loss: Discretized PDE residual

Based on the continuous augmented PDE residual $\mathcal{R}(u)$ from Equation equation 2, we formulate its discretized form

$$R(u) = \frac{1}{|\mathcal{M}|} \sum_{x \in \mathcal{M}} \frac{1}{|\mathcal{T}|} \sum_{t \in \mathcal{T}} \left( \mathcal{S}[u](x,t) \right)^2 \tag{3}$$

$$+ \beta_1 \frac{1}{|\mathcal{M}_{IC}|} \sum_{x \in \mathcal{M}} \left( u(x,0) - u_0(x) \right)^2 \tag{4}$$

$$+ \beta_2 \frac{1}{|\mathcal{M}_{BC}|} \sum_{x \in \mathcal{M}_{\partial\Omega}} \frac{1}{|\mathcal{T}|} \sum_{t \in \mathcal{T}} \left( \mathbb{B}[u](x,t) - g(x,t) \right)^2, \tag{5}$$

where $\mathcal{M}$ is the discretization grid inside the domain, $\mathcal{M}_{BC}/\mathcal{M}_{IC}$ are the discretization points on the domain boundary and initial conditions, respectively, and $\mathcal{T}$ is the time discretization to evaluate the PDE and boundary operators on. For any candidate decoded function $u_w = \mathcal{I}(\mathcal{D}(z))$, we use $R$ as the target metric throughout all steps of the solution search pipeline. Within our experiments, we choose the spatial discretization to be 128 and 128 as a time discretization for all the problems except Damping Wave where we use 64 and 64 respectively.

B.3   Stage I: Structure discovery by iterative clustering

Component-wise libraries.   The Ansatz function $A$ specifies the number of components as well as which variables should be present per component (and possibly other syntactic requirements). We therefore filter the initial library $\mathcal{L}$ for each component to retain only viable candidate expressions to obtain $\mathcal{L}^{(c)}$, and the corresponding encoded latent vectors $Z^{(c)} = \{\mathcal{E}(w) : w \in \mathcal{L}^{(c)}\}$.

Initial clustering.   We then iteratively partition the latent subspaces $Z^{(c)}$ for each of the components separately into $K^c$ clusters by k-means clustering, sample from each the clusters, and assemble solution candidates. Let $\mathbb{N}_{K^{(c)}} = \{1, \ldots, K^c\}$ denote the cluster indices for component $c$, and by $\mathcal{K} = \mathbb{N}_{K^{(1)}} \times \cdots \times \mathbb{N}_{K^{(L)}}$ the cluster index set of all possible cross-component cluster combinations. For example, in case of spatio-temporal separability $u(x,t) = f(x) \cdot g(t)$, spatial and temporal components are separated into component-wise libraries, clustered, and sampled independently and solutions are assembled from pairs of clusters $(k_x, k_t)$.

Cluster selection.   We sample $M$ tuples of cluster indices $k_i = (k_i^{(1)}, \ldots, k_i^{(L)}) \in \mathcal{K}$, where $k_i^{(c)}$ denotes the index of the cluster used for the $c$-th component in the $i$-th sample. For each component $c$, we choose a latent vector $z_i^{(c)}$ from the current encoded library vectors in cluster $k_{i,c}$, decode $w_i^{(c)} = \mathcal{D}(z_i^{(c)})$, and assemble $w_i = A(w_i^{(1)}, \ldots, w_i^{(L)})$. Then, we evaluate the discretized residual for each candidate, $r_i = R(\mathcal{I}(w_i))$. Finally, we select the candidate with minimal residual and record the cluster indices $k^* = k_i = (k_i^{(1)}, \ldots, k_i^{(L)})$ of the candidate with minimal residual $w^* = w_i$ as the current best clusters.

Iterative subclustering.   Each of the component-wise subclusters selected as current best clusters in the previous iteration are partitioned into $K_c$ sub-clusters by k-means clustering on the latent vectors. The cluster selection is repeated (sample combinations of clusters,

decode and assemble expressions, evaluate the residual, choose best cluster combination) and the best cluster combination $k^*$ is updated from the new, refined clusters. Iteratively, the size of the resulting clusters shrinks, focusing in on the final best cluster combination. This procedure is repeated until a target residual is reached, $r^\star \leq \varepsilon_{\text{struct}}$, or an evaluation budget on the number of iterations is exhausted.

**Generation of additional latent vectors.** As the size of the latent clusters decreases, there are fewer latent vectors of the initial training library $z_i$. New latent vectors can be generated for these clusters, further exploring the latent space beyond what the GVAE has seen during training. We generate these samples by convex interpolation of its members with small isotropic jitter (decodable latent interpolation).

### B.4 Stage II: Coefficient Refinement

Given the best symbolic structure $w^\star$ from Stage I, we freeze the form and expose only its numeric literals as trainable parameters $p \in \mathbb{R}^P$, where we protect constants such as $\pi$, $e$, and integer exponents. We minimize the PDE residual from R(u) on the resulting parametric function family $u^\star(\cdot; p) = \mathcal{I}(w^\star(p))$ to obtain the best constants

$$p^* = \arg\min_p R(u^\star(\cdot; p))$$

and the corresponding final (exact or approximate) solution $u^\star(\cdot) = u^\star(\cdot; p^\star)$.

**Implementation.** We compile $u(\cdot; p)$ in float64 JAX, obtain the required derivatives by automatic differentiation to evaluate $\mathcal{S}[u]$ on the grids $(\mathcal{M}, \mathcal{T})$, and compute $R(u)$ batched over all points.

We parse the numeric literals in $w^\star$ to form $\bar{p}$. For single-start, set $p^{(0)} = \bar{p}$. For multi-start, draw

$$p^{(0,s)} \sim \mathcal{N}\big(\bar{p}, \ \text{diag}\big((\eta|\bar{p}|)^2\big)\big), \qquad s = 1, \ldots, S,$$

and optimize all starts in parallel (JAX vmap). We use Adam (Optax) with exponential learning-rate decay and JIT. Early stopping triggers when $\sqrt{R(u)} < \varepsilon_{\text{tol}}$ or a budget is reached.

---

**Algorithm 1 SIGS: Symbolic Iterative Grammar Solver (overview)**

---

**Require:** Grammar $\mathcal{G}$, assembly map $A$, trained TGVAE $(\mathcal{E}, \mathcal{D})$, discretized residual $R$, budgets $(M, T_{\max})$, thresholds $(\varepsilon_{\text{struct}}, \varepsilon_{\text{tol}})$
**Ensure:** Refined symbolic solution $u^\star$ with coefficients $p^*$
 1: Stage 0 (amortized): Initial_Sampling $\rightarrow (w^\star, z^\star, k^\star, r^\star)$
 2: Stage I (structure): Subcluster_Refine$(w^\star, z^\star, k^\star, r^\star) \rightarrow (w^\star, z^\star, k^\star, r^\star)$
 3: Stage II (coeffs): Coefficient_Refinement$(w^\star, \varepsilon_{\text{tol}}) \rightarrow p^*$
 4: return $w^\star(p^*)$

---

## C  Experiments

### C.1  Problem Definitions

We evaluate our method on five representative PDEs spanning steady-state and time-dependent settings. Following the general formulation in equation 1, we specify the differential operator $\mathbb{D}$ in Table 14, together with the computational domain and mesh used to evaluate the residual $\mathcal{R}(u)$ and to compute discretized solutions with FEM and PINN methods. The forcing term **f** and initial/boundary conditions for each test problem are specified in the following, where we distinguish cases with known (manufactured) and unknown analytic solutions.

---

**Algorithm 2** Stage 0 (amortized): Library clustering and initial assembly

---

**Require:** Assembly map $A$, encoder/decoder $(\mathcal{E}, \mathcal{D})$, discretized residual $R$, draw budget $M$
**Ensure:** Best candidate $(w^\star, z^\star, k^\star, r^\star)$
1: For each component $c \in \mathbb{N}_L$: enforce variable constraints $\mathcal{C}_c$ from the Ansatz to filter the library $\mathcal{L}^{(c)}$
2: Encode and cluster: $Z^{(c)} = \{\mathcal{E}(w) : w \in \mathcal{L}^{(c)}\}$, partition into $K^c$ clusters; let $\mathbb{N}_{K^{(c)}} = \{1, \ldots, K^c\}$
3: Initialize incumbent $r^\star \leftarrow +\infty$
4: **for** $i = 1$ to $M$ **do**
5: $\quad$ Sample cluster tuple $k_i = (k_{i,1}, \ldots, k_{i,L}) \in \mathbb{N}_{K^{(1)}} \times \cdots \times \mathbb{N}_{K^{(L)}}$
6: $\quad$ **for** each $c$ **do**
7: $\quad\quad$ Draw $z_i^{(c)}$ from cluster $k_{i,c}$; decode $w_i^{(c)} = \mathcal{D}(z_i^{(c)})$
8: $\quad$ **end for**
9: $\quad$ Assemble $w_i = A(w_i^{(1)}, \ldots, w_i^{(L)})$
10: $\quad$ Score $r_i = R(\mathcal{I}(w_i))$
11: $\quad$ **if** $r_i < r^\star$ **then**
12: $\quad\quad$ $(w^\star, z^\star, k^\star, r^\star) \leftarrow (w_i, [z_i^{(1)}; \ldots; z_i^{(L)}], k_i, r_i)$
13: $\quad$ **end if**
14: **end for**
15: **return** $(w^\star, z^\star, k^\star, r^\star)$

---

**Algorithm 3** Stage I: Focused subclustering and structure refinement

---

**Require:** Incumbent $(w^\star, z^\star, k^\star, r^\star)$ from Stage 0, encoder/decoder $(\mathcal{E}, \mathcal{D})$, residual $R$, assembly $A$, budgets $(T_{\max})$, threshold $\varepsilon_{\text{struct}}$
**Ensure:** Updated $(w^\star, z^\star, k^\star, r^\star)$
1: For each $c \in \mathbb{N}_L$: restrict to latents in cluster $K^{(c),\star}$ and partition into $H_c$ subclusters
2: $t \leftarrow 0$
3: **while** $r^\star > \varepsilon_{\text{struct}}$ and $t < T_{\max}$ **do**
4: $\quad$ Sample subcluster tuple $h = (h_1, \ldots, h_L) \in \mathbb{N}_{H^{(1)}} \times \cdots \times \mathbb{N}_{H^{(L)}}$
5: $\quad$ **for** each $c$ **do**
6: $\quad\quad$ **if** subcluster $h_c$ is too small **then**
7: $\quad\quad\quad$ generate samples in $h_c$ by convex interpolation plus small isotropic jitter
8: $\quad\quad$ **end if**
9: $\quad\quad$ Draw $z^{(c)}$ from subcluster $h_c$
10: $\quad\quad$ Decode $w^{(c)} = \mathcal{D}(z^{(c)})$
11: $\quad$ **end for**
12: $\quad$ Assemble $w = A(w^{(1)}, \ldots, w^{(L)})$
13: $\quad$ Score $r = R(\mathcal{I}(w))$
14: $\quad$ **if** $r < r^\star$ **then**
15: $\quad\quad$ $(w^\star, z^\star, k^\star, r^\star) \leftarrow (w, [z^{(1)}; \ldots; z^{(L)}], h, r)$
16: $\quad$ **end if**
17: $\quad$ $t \leftarrow t + 1$
18: **end while**
19: **return** $(w^\star, z^\star, k^\star, r^\star)$

---

### C.1.1 Construction of known analytical solutions

For the four problems with known analytical solutions, we employ the method of manufactured solutions to construct the test problems and ensure exact error quantification. Given a chosen analytical solution $u_{\text{true}}$, we construct the forcing term via $\mathbf{f} = -\mathbb{D}[u_{\text{true}}]$ to guarantee that $u_{\text{true}}$ satisfies the PDE exactly. Initial and boundary conditions are then prescribed from $u_{\text{true}}$ to complete the well-posed problem formulation.

The specific analytical solutions are detailed in Table 15. These solutions are chosen to exhibit diverse mathematical behaviors: the Burgers' equation features a smooth shock profile with nonlinear advection, the diffusion equation uses a multi-mode separated solution with

**Algorithm 4** Stage II: Multi-start coefficient refinement (JAX)

---

**Require:** Best structure $w^\star$, residual $R$, tolerance $\varepsilon_{\text{tol}}$, starts $S$, noise scale $\eta$
**Ensure:** Optimized coefficients $p^*$ and refined $w^\star(p^*)$
 1: Parse numeric literals in $w^\star$ to get $\bar{p}$
 2: **for** $s = 1$ to $S$ **do**
 3:    Initialize $p^{(0,s)} \sim \mathcal{N}\big(\bar{p},\ \text{diag}((\eta|\bar{p}|)^2)\big)$
 4: **end for**
 5: Optimize all starts with Adam (JAX, float64, JIT) and exponential LR decay; at each step evaluate $R\big(\mathcal{I}(w^\star(p^{(t,s)}))\big)$
 6: Early-stop when $\sqrt{R} < \varepsilon_{\text{tol}}$ or budget reached; keep the best $p^* = \arg\min_s R\big(\mathcal{I}(w^\star(p^{(\cdot,s)}))\big)$
 7: **return** $p^*$ and $w^\star(p^*)$

---

Table 12: Canonical problems reproduced from prior work. Dimension notation: $n+m$D denotes $n$ spatial and $m$ temporal variables.

| Problem (paper) | Operator $\mathbb{D}$ | Dim | Domain | Mesh | Ground truth $u^\star$ |
|---|---|---|---|---|---|
| Poisson1 (HDTLGP) | $u_{xx} + u_{yy}$ | 2D | $[0,1]^2$ | $64^2$ | $\sin(\pi x)\sin(\pi y)$ |
| Advection3 (HDTLGP) | $u_t + u_x + u_y$ | 2+1D | $[0,1]^2 \times [0,2]$ | $64^2 \times 64$ | $\sin(x-t) + \sin(y-t)$ |
| Wave2D (SSDE) | $u_{tt} - (u_{xx} + u_{yy})$ | 2+1D | $[-1,1]^2 \times [0,1]$ | $8^2 \times 8$ | $e^{x^2}\sin(y)\,e^{-0.5t}$ |

exponential decay, the wave equation employs a truncated Fourier series, and the damping wave incorporates both temporal decay and spatial wave propagation in two dimensions.

### C.1.2   Problem without known analytic solution

The Poisson–Gauss problem represents a realistic scenario where no analytical solution is available, making it particularly valuable for assessing method performance in practical applications. The problem consists of the steady-state Poisson equation $\nabla^2 u = \mathbf{f}$ on the unit square $[0,1]^2$ with homogeneous Dirichlet boundary conditions $u = 0$ on $\partial[0,1]^2$.

The forcing term $\mathbf{f}$ is constructed as a superposition of $n$ isotropic Gaussian sources:

$$\mathbf{f}(x,y) = \sum_{i=1}^{n} \exp\left(-\frac{(x - \mu_{x,i})^2 + (y - \mu_{y,i})^2}{2\sigma^2}\right) \tag{6}$$

with fixed width $\sigma = 0.1$ and deterministically chosen centers:

- PG-2: $(0.3, 0.8)$, $(0.7, 0.2)$
- PG-3: $(0.3, 0.8)$, $(0.7, 0.2)$, $(0.5, 0.2)$
- PG-4: $(0.3, 0.8)$, $(0.7, 0.2)$, $(0.5, 0.2)$, $(0.4, 0.6)$

This configuration creates localized source regions with smooth spatial variation, testing the method's ability to capture multi-scale features and handle problems without ground truth solutions. For evaluation on this problem, we rely on mesh convergence studies and physics-based consistency checks rather than direct error computation against an analytical reference.

### C.2   Solution Ansatz specific to our experiments

Our framework generates eigenfunction components for five distinct operator classes, each producing characteristic mathematical patterns with specific parameter ranges that ensure physical relevance and numerical stability.

- Wave operators $\mathbb{D}_{\text{wave}} = \nabla^2 - \frac{1}{c^2}\frac{\partial^2}{\partial t^2}$ generate oscillatory eigenmodes $a_k \sin(k\pi x)\cos(ck\pi t)$ where mode indices $k \in \{1, 2, \ldots, K\}$ determine spatial harmonic frequencies, wave speeds $c \in [0.1, 0.8]$ control temporal oscillation rates,

Table 13: Closed-form parity on canonical problems from prior work. We list the operator, domain, evaluation mesh, the ground-truth solution $u^\star$, the closed form printed in the original paper, and the expression found by SIGS, together with relative $L^2$ errors (discrete, uniform grid).

| Problem (source) | Baseline expression | SIGS expression |
|---|---|---|
| Poisson1 (HDTLGP) | $\sin(3.141\,x)\sin(3.142\,y)$ | $\sin(\pi x)\sin(\pi y)$ |
| Advection3 (HDTLGP) | $-\sin(0.9838t - x) - \sin(0.9979t - y)$ | $\sin(x - t) + \sin(y - t)$ |
| Wave2D (SSDE) | $e^{x^2 - 0.5t}\sin(y)$ | $e^{x^2}\sin(y)\,e^{-0.5t}$ |

Table 14: Summary of benchmark problems. Dimension notation: $n{+}m$D denotes $n$ spatial and $m$ temporal dimensions.

| Problem | Operator $\mathbb{D}$ | Dim | Domain | Mesh | Key Parameters |
|---|---|---|---|---|---|
| Burgers' | $u_t + uu_x - \nu u_{xx}$ | 1+1D | $[-5,5]\times[0,2]$ | $128 \times 128$ | $\nu = 0.01$ |
| Diffusion | $u_t - \kappa u_{xx}$ | 1+1D | $[0, 1.397]\times[0,1]$ | $128 \times 128$ | $\kappa = 0.697$ |
| Damping wave | $u_{tt} + u_t - c^2(u_{xx} + u_{yy})$ | 2+1D | $[-8,8]^2 \times [0,4]$ | $32 \times 32 \times 32$ | $c = 0.8$ |
| Poisson–Gauss | $-(u_{xx} + u_{yy})$ | 2D | $[0,1]^2$ | $100 \times 100$ | |

and amplitude coefficients $a_k = \frac{m \times 10^e}{k} \times \frac{\pi}{K}$ use scientific notation with mantissa $m \in [5, 9]$ and exponential damping to ensure numerical stability across multiple scales.

- Diffusion operators $\mathbb{D}_{\text{diff}} = \nabla^2 - \frac{\partial}{\partial t}$ produce separable heat modes $\frac{2M_0}{L}\sin\left(\frac{(2n+1)\pi x}{L}\right)e^{-\frac{(2n+1)^2\pi^2 Dt}{L^2}}$ where amplitude coefficients $M_0 \in [1,3]$ set initial magnitudes, domain lengths $L \in [0.1, 1.5]$ determine spatial scales, diffusivities $D \in [0.01, 1]$ control temporal decay rates, and odd harmonic indexing $(2n + 1)$ corresponds to homogeneous Dirichlet boundary conditions with mode numbers $n \in \{0, 1, 2\}$ generating the first three eigenmodes.

- Viscous Burgers operators $\mathbb{D}_{\text{Burgers}} = u\frac{\partial u}{\partial x} - \nu\frac{\partial^2 u}{\partial x^2}$ create shock transition profiles consisting of average components $\frac{u_L + u_R}{2}$ and shock components $\frac{u_L - u_R}{2}\tanh\left(\frac{(x - x_0 - st)(u_L - u_R)}{4\nu}\right)$ where left asymptotic states $u_L \in [1, 3]$ and right asymptotic states $u_R \in [-1, 1]$ define the shock amplitude, propagation speeds $s \in [0.1, 2]$ control shock movement, initial positions $x_0 \in [-1, 1]$ set shock locations, and kinematic viscosities $\nu \in [0.01, 1]$ determine shock width.

- Poisson-Gauss operators $\mathbb{D}_{\text{Poisson}} = \nabla^2$ with source terms generate localized Gaussian profiles $e^{-\alpha((x - x_0)^2 + (y - y_0)^2)}$ for superposition of Gaussian source terms and polynomial harmonic functions for steady-state equilibrium configurations, where decay parameters $\alpha$ control Gaussian widths, center coordinates $(x_0, y_0)$ determine localization, and multiple Gaussians can be superposed as source terms.

- Outgoing damped wave operators $\mathbb{D}_{\text{out-wave}} = \nabla^2 - \frac{1}{c^2}\frac{\partial^2}{\partial t^2} + \gamma\frac{\partial}{\partial t}$ combine envelope functions $h\left(e^{((x - x_0)^2 + (y - y_0)^2)/(w(1+t))} + 1\right)^{-1}$, oscillatory kernels $\cos(k\sqrt{(x - x_0)^2 + (y - y_0)^2} - ct)$, and decay factors $e^{-at}$ where amplitudes $h \in [0.01, 0.5]$, envelope width parameters $w \in [0.3, 1.0]$, radial wave numbers $k \in [0.5, 4.0]$, phase velocities $c \in [0.1, 1.0]$, temporal decay rates $a \in [0.02, 0.8]$, and center coordinates $(x_0, y_0) \in [-6, 6]^2$ control the composite spatiotemporal structure.

Our grammar-based approach produces eigenfunction components at different structural levels including elementary eigenmodes corresponding to individual spatial harmonics $\phi_k(x)$ and temporal factors $\psi_k(t)$, separable products representing complete eigenfunctions $\phi_k(x)\psi_k(t)$ generated when the grammar produces expressions containing both spatial and

Table 15: Analytical (manufactured) solutions for benchmark problems.

| Problem | Analytical Solution $u_{\text{true}}$ | Constants |
|---|---|---|
| Burgers' | $0.86 + 0.6 \tanh(25.8\,t - 30.0\,x + 9.9)$ | − |
| Diffusion | $A[\sin\left(\frac{\pi x}{L}\right) e^{-\frac{\pi^2 \kappa}{L^2} t} - \sin\left(\frac{3\pi x}{L}\right) e^{-\frac{9\pi^2 \kappa}{L^2} t}$ $+ \sin\left(\frac{5\pi x}{L}\right) e^{-\frac{25\pi^2 \kappa}{L^2} t}]$ | $A = 3.974,\ L = 1.397$ |
| Damping wave | $e^{-\alpha t} \cos(\omega t - KR(x,y)),$ where $R(x,y) = \sqrt{(hx+1)^2 + (hy-1)^2}$ | $h = 0.2,\ K = 2.5,\ \omega = 0.4,$ $\alpha = 0.45$ |

temporal variables, and composite structures like non-separable patterns $\cos(\sqrt{x^2 + y^2}/t)$ that the grammar can generate through its compositional rules but cannot be factorized. For ODE problems and linear spatial DEs such as Poisson and Laplace equations where component structure is simpler, we supplement operator-informed generation with probabilistic grammar expansion using a context-free grammar that recursively builds expression trees by selecting binary operations with probability 0.6, unary functions with probability 0.3, and terminal symbols with probability 0.1.

### C.3   Configuration of baseline methods

#### C.3.1   SSDE Primitive Sets

To ensure fair comparison, SSDE receives primitive sets derived from the same structural Ansatz used by SIGS. For problems expecting separated variable forms (e.g., $f(x) \cdot g(t)$ for spatiotemporal PDEs), we provide SSDE with functions that appear in the corresponding variable-specific clusters within SIGS's grammar.

Table 16: SSDE primitive sets derived from SIGS's structural Ansatz

| Problem | Expected Form | Variables | Function Set |
|---|---|---|---|
| Burgers | $f(x,t)$ | $(x,t)$ | $\{+, -, \times, \div, \exp, \tanh, \sin, \cos\}$ |
| Diffusion | $\sum_i f_i(x) \cdot g_i(t)$ | $(x,t)$ | $\{+, -, \times, \div, \exp, \tanh, \sin, \cos, \log\}$ |
| Damping Wave | $f(x,y) \cdot g(t)$ | $(x,y,t)$ | $\{+, -, \times, \div, \exp, \sin, \cos, \sqrt{\ }\}$ |
| PG-2/3/4 | $f(x,y)$ | $(x,y)$ | $\{+, -, \times, \div, \exp, \log, x^n, \sin, \cos\}$ |

**Rationale for Primitive Selection.**   The primitive sets are determined by analyzing which functions appear in SIGS's variable-specific clusters:

- For separated forms $f(x) \cdot g(t)$: We include functions from both the spatial cluster (containing $x$) and temporal cluster (containing $t$)

- For spatiotemporal problems: $\{\sin, \cos\}$ from spatial modes, $\{\exp\}$ from temporal decay, $\{\tanh\}$ for shock profiles (Burgers-specific)

- For wave equations: Exclude exp since the temporal cluster for waves contains only oscillatory functions

- For spatial-only problems (PG): Include functions from the $(x, y)$ spatial cluster

This ensures both methods access identical function spaces, SIGS through its structured grammar clusters and SSDE through explicit primitive specification. The key difference lies in search strategy: SIGS restricts combinations to physically motivated forms, while SSDE explores all possible tree compositions.

**SSDE Hyperparameters.**   All problems use consistent RL hyperparameters: learning rate 0.0005, entropy weight 0.07, batch size 1000, 200,000 training samples, and expression length constraints between 4 and 30 tokens (extended to 60 for Diffusion due to its multimodal structure requiring more complex expressions).

### C.3.2 HD-TLGP Protocols

We evaluate HD-TLGP under two protocols that parallel the conditions for SSDE and SIGS.

**Protocol 1: Knowledge-Based Initialization.** HD-TLGP receives problem-specific solution components in its knowledge base:

- Diffusion: First mode with exact amplitude $A\sin(\pi x/L)\exp(-\pi^2 Dt/L^2)$, templates for modes 3 and 5, and 2-3 mode combinations

- Burgers: Core shock $\tanh(\alpha(x - x_0 - st))$ and scaled variant

- Damping Wave: Radial motif $\cos(k\sqrt{(x-x_0)^2+(y-y_0)^2}-\omega t)$ and separable template $\sin(\pi x)\sin(\pi y)\cos(\omega t)\exp(-\gamma t)$

- PG-2/3/4: Boundary mask $\sin(\pi x)\sin(\pi y)$, individual Gaussians for each center, and sum-of-Gaussians template

These components test whether genetic programming can extend partial solutions (Diffusion/Wave), refine parametric forms (Burgers), or combine spatial structures (Damping Wave, PG).

**Protocol 2: Primitive-Only Discovery.** HD-TLGP starts from random expressions using exactly the same primitive set as SSDE for each problem:

- 1D problems (Diffusion, Wave, Burgers): $\{+, -, \times, \div, \sin, \cos, \exp, \tanh\}$

- Damping Wave: $\{+, -, \times, \div, \sin, \cos, \exp, \tanh, \sqrt{}\}$

- PG-2/3/4: $\{+, -, \times, \div, \sin, \cos, \exp, \log, \sqrt{}\}$

No knowledge base components are provided, requiring complete discovery from elementary functions. This ensures all symbolic methods explore identical function spaces.

**Implementation Details.** Population size 200 (1D) or 50 (2D), crossover 0.6, mutation 0.6, KB transfer 0.6 (Protocol 1 only), maximum 25 generations or 120-1200 seconds, local optimization enabled for constant tuning, peephole simplification for expression reduction.

### C.4 FEniCS Validation for Reference Solutions

For the Poisson-Gauss problems lacking analytical solutions, we establish numerical ground truth through rigorous finite element analysis. We solve the Poisson equation $-\nabla^2 u = f$ with homogeneous Dirichlet boundary conditions on the unit square, where $f$ consists of superposed Gaussian sources:

$$f(x, y) = \sum_{i=1}^{n} \exp\left(-\frac{(x - \mu_{x,i})^2 + (y - \mu_{y,i})^2}{2\sigma^2}\right)$$

**Verification Methodology.** To validate our FEniCS reference solutions, we employ three convergence criteria:

1. Mesh convergence: Solutions computed on progressively refined meshes ($16\times16$ through $128\times128$) with P4 elements

2. Energy balance: The weak form identity $a(u_h, u_h) = L(u_h)$ must hold to machine precision, where $a(u, v) = \int_{\Omega} \nabla u \cdot \nabla v\, dx$ and $L(v) = \int_{\Omega} fv\, dx$

3. Residual minimization: The strong-form PDE residual $\|-\nabla^2 u_h - f\|_{L^2}$ decreases at the expected rate $O(h^{p+1})$

Table 17: FEniCS Convergence Study for Poisson-Gauss (PG-2) Problem using P2 Elements

| Problem | Mesh | DOF | Runtime (s) | PDE Residual ($\mathbf{R(u)}$) |
|---|---|---|---|---|
| | 32×32 | 4225 | 0.0672 | 2.0525e−02 |
| | 64×64 | 16641 | 0.2103 | 1.0238e−02 |
| | 100×100 | 40401 | 0.5271 | 6.5487e−03 |
| PG-2 | 128×128 | 66049 | 0.9463 | 5.1154e−03 |
| | 256×256 | 263169 | 3.5499 | 2.5573e−03 |
| | 512×512 | 1050625 | 14.9201 | 1.2787e-03 |
| | 1024×1024 | 4198401 | 57.0198 | 6.3970e−04 |

**Validation Results.** Table 17 shows the convergence study for the Poisson-Gauss (PG-2) problem using $P2$ elements. This study focuses on validating the stability and efficiency of the Finite Element solution. The *Runtime* shows the computational cost increases proportionally with the DOF. Most importantly, the *PDE Residual* ($R(u)$) demonstrates stable, clear convergence, decreasing by a factor of approximately 2 with each mesh refinement. This confirms a consistent $O(h^1)$ decay rate for the residual, proving that the solution is systematically converging to satisfy the strong form of the Poisson equation and the Dirichlet boundary conditions.

**Evaluation of Discovered Expressions.** Symbolic expressions discovered by SIGS and baseline methods are evaluated against these FEniCS references through Galerkin projection. Given a discovered expression $u_{\text{sym}}(x, y)$, we compute its projection onto the finite element space and measure the relative L² error:

$$\text{Error} = \frac{\|u_h^{\text{FEM}} - \Pi_h u_{\text{sym}}\|_{L^2}}{\|u_h^{\text{FEM}}\|_{L^2}}$$

where $\Pi_h$ denotes the L² projection operator onto the P4 finite element space. This provides a rigorous, mesh-independent measure of solution quality for problems without analytical ground truth.

### C.5 Discovered Symbolic Expressions

Tables 18–20 present the symbolic expressions discovered by each method. The structural differences are immediately apparent: SIGS produces compact, physically interpretable expressions that directly reflect PDE solution structures—separated variables for diffusion, traveling waves for Burgers, and properly masked Gaussians for Poisson problems. In contrast, both HD-TLGP and SSDE generate deeply nested compositions of elementary functions. HD-TLGP Protocol 1, despite receiving solution components, wraps them in other operations (e.g., $\sin(\sin(\cos(\exp(\cdot))))$ around the Burgers shock), while Protocol 2 often collapses to trivial constants for complex problems. SSDE consistently produces expressions with extreme nesting depth—up to 500+ operations for Damping Wave, that represent brute-force fitting rather than discovery of underlying mathematical structure. These expressions, while potentially achieving low training error fail to generalize and provide no insight into the PDE dynamics.

### C.6 Details on the ablation Study

**Mahalanobis distance.** Given $x \in \mathbb{R}^d$, mean $\mu$, covariance $\Sigma \succ 0$, the Mahalanobis distance is $d_M(x, \mu) = \sqrt{(x - \mu)^\top \Sigma^{-1}(x - \mu)}$. For each model, we compute training encoder means $\{\mu_i\}_{i=1}^N$, estimate $\Sigma$ from these means, and define $d_{\min}(z) = \min_i d_M(z, \mu_i)$. Our filter accepts a candidate $z$ iff $d_{\min}^{(\text{with})}(z) \geq \tau$ and $d_{\min}^{(\text{w/o})}(z) \geq \tau$ (we use $\tau{=}0.8$).

Table 18: Symbolic expressions discovered by SIGS

| Problem | Discovered Expression |
|---|---|
| Burgers | $0.86 + 0.6\tanh(30(x - 0.33 - 0.86t))$ |
| Diffusion | $3.974(\sin(2.15\pi x)e^{-3.21\pi^2 t} + \sin(0.71\pi x)e^{-0.36\pi^2 t} + \sin(3.58\pi x)e^{-8.93\pi^2 t})$ |
| Damping Wave | $\cos(0.5\sqrt{(x + 5.0)^2 + (y - 5)^2} - 0.4t)e^{-0.45t}$ |
| PG-2 | $\sin(\pi x)\sin(\pi y) \cdot \big[0.0080\exp\big(\frac{0.424((x-0.923)^2+(y-0.760)^2)}{2.136\cdot 0.573^2}\big)$ 
 $+0.0251\exp\big(\frac{-1.071((x-0.794)^2+(y-0.054)^2)}{2.245\cdot 0.201^2}\big)$ 
 $+0.0105\exp\big(\frac{-1.110((x-0.248)^2+(y-0.496)^2)}{1.862\cdot 0.185^2}\big)\big]$ |
| PG-3 | $\sin(\pi x)\sin(\pi y) \cdot \big[0.0079\exp\big(\frac{0.461((x-0.500)^2+(y+0.217)^2)}{2.152\cdot 0.508^2}\big)$ 
 $+0.0137\exp\big(\frac{-0.816((x-0.750)^2+(y-0.873)^2)}{1.898\cdot 0.138^2}\big)$ 
 $+0.0137\exp\big(\frac{-0.851((x-0.250)^2+(y-0.873)^2)}{2.505\cdot 0.123^2}\big)$ 
 $+0.0206\exp\big(\frac{-1.092((x-0.500)^2+(y-0.043)^2)}{1.738\cdot 0.221^2}\big)\big]$ |
| PG-4 | $\sin(\pi x)\sin(\pi y) \cdot \big[0.0068\exp\big(\frac{-1.489((x-0.731)^2+(y-0.502)^2)}{1.553\cdot 0.195^2}\big)$ 
 $+0.0112\exp\big(\frac{-1.123((x-0.500)^2+(y-0.124)^2)}{1.804\cdot 0.159^2}\big)$ 
 $+0.0294\exp\big(\frac{-0.031((x-0.665)^2+(y-0.887)^2)}{2.025\cdot 0.584^2}\big)$ 
 $+0.0069\exp\big(\frac{-0.992((x-0.267)^2+(y-0.502)^2)}{1.664\cdot 0.155^2}\big)$ 
 $+0.0286\exp\big(\frac{-1.024((x-0.501)^2+(y+0.276)^2)}{1.569\cdot 0.190^2}\big)\big]$ |

Table 19: Symbolic expressions discovered by SSDE

| Problem | Expression Found |
|---|---|
| Burgers | $\exp(\tanh(-1743.845x - 76821.176)/\sin(\exp(\tanh(\exp(x))))) \cdot$ 
 $\exp(-\tanh(-t\exp(t) - 3t + \tanh(-1743.845x - 76821.176)/\sin(\exp(\tanh(\exp(x))))))$ |
| Diffusion | $\cos(t + x + \cos(112.185x^3\tanh(x^2 + x) - 118.201x^3 + 8.824x) - \tanh(t))/(2t + x/\cos(-t + x + \cos(112.185x^3\tanh(x^2 + x) - 118.201x^3 + 8.824x)) + \cos(112.185x^3\tanh(x^2 + x) - 118.201x^3 + 8.824x)/\cos(-t + x + \cos(112.185x^3\tanh(x^2 + x) - 118.201x^3 + 8.824x)))$ |
| Damping Wave | Expression with 500+ operations including nested functions, (full expression exceeds reasonable display length) |
| PG-2 | $sin(x(-0.02582816 + \frac{0.01654789}{cos(x2+cos(y(y+exp(x\cdot cos(y+0.42096704)))))}))$ |
| PG-3 | $x(-0.802y^2 + y)(1.092cos(sin(sin(sin(x)))) - 0.82463681261637)$ |
| PG-4 | $x(\log(\sin(\sin(\cos(\sin(\sin(\sin(\sin(\sin(\sin(x))))))))))) + 0.383) - 0.007$ |

Table 20: Symbolic expressions discovered by HD-TLGP

| Problem | Protocol | Expression Found |
|---------|----------|------------------|
| Burgers | P1 | $\sin(\sin(\cos(\exp(-\tanh(0.996\tanh(25.8t - 30.0x + 9.9)))))) + 0.569$ |
|  | P2 | $\tanh(\exp(-\cos(\tanh(\tanh(x) + 0.5) + \tanh(1.649\exp(-x)))) + \tanh(\sin(x + 48.558) + \tanh(x)))$ |
| Diffusion | P1 | $3.974 \cdot (\exp(88.121t + 3.974\exp(-3.525t)\sin(63.495/x) \cdot \sin(2.249x))\sin(2.249x) + \sin(11.244x))\exp(-91.646t)$ |
|  | P2 | $\sin(\tanh(\tanh(x))) + 1.840\tanh(t + \sin(0.540 \cdot \exp(\frac{0.5}{\sin(\cos(\sin(x+\pi))/((1.623t-5.100)(-t+2x+1.0)))}\tanh(1.019\cos(1.649t + 0.824)))))$ |
| Damping Wave | P1 | $\exp(\cos(0.542t - 0.357((x-0.135)(x+0.269) + (y-0.940)(y+0.495))^{1/2}))$ |
|  | P2 | Complex nested expression with 150+ operations including imaginary unit |
| PG-2 | P1 | $0.0181\sin(\exp(1.638(26.282 + \exp(-0.992/(x^2 - 40x + y^2 - 26.598y + 531.969)))^{1/2}) - 1.638\exp(-0.191/(7.560x \cdot \exp(1.670x) + 117.703x - 370.656y + 61.605\exp(1.670x) - 1833.212))\sin(x - y))$ |
|  | P2 | $0.000105$ |
| PG-3 | P1 | $(y - 0.468)(y - 0.440)(\sin((\tanh(y) + 9.870)\exp(0.0120/((x - 0.538)(x - 0.032) + (y - 1.662)(y - 1.548)))))^{1/2}$ |
|  | P2 | $9944.705\sin(x) + 1.218 \times 10^{-12}$ |
| PG-4 | P1 | $12945.616/\sin(\exp(\exp(\exp(-1.095\exp(3.535/((x + 0.0183)^2 + (y - 0.389)(y - 0.368)))/(-1.527y + 1.527(x - 0.764)^2\exp(3.535/((x + 0.0183)^2 + (y - 0.389)(y - 0.368))) - 5.261)))))$ |
|  | P2 | $9505.982$ |

C.7  Solution Visualizations

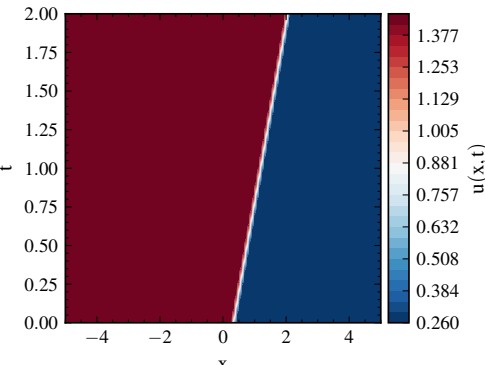

Figure 4: Contour plot of the learned solution $u(x, t)$ for the Burgers equation. The horizontal axis represents the spatial domain $x \in [-5, 5]$, the vertical axis represents the temporal domain $t \in [0, 2]$, and the colormap indicates the solution magnitude ranging from 0.26 to 1.46. The solution is computed on a $128 \times 128$ discretization grid

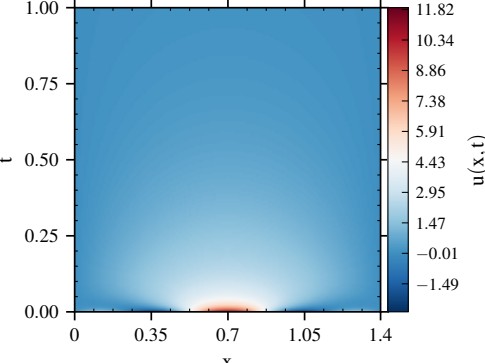

Figure 5: Contour plot of the learned solution $u(x, t)$ for the Diffusion equation. The horizontal axis represents the spatial domain $x \in [0, 1.4]$, the vertical axis represents the temporal domain $t \in [0, 1]$, and the colormap indicates the solution magnitude ranging from $-1.5$ to 11.9. The solution is computed on a $128 \times 128$ discretization grid.

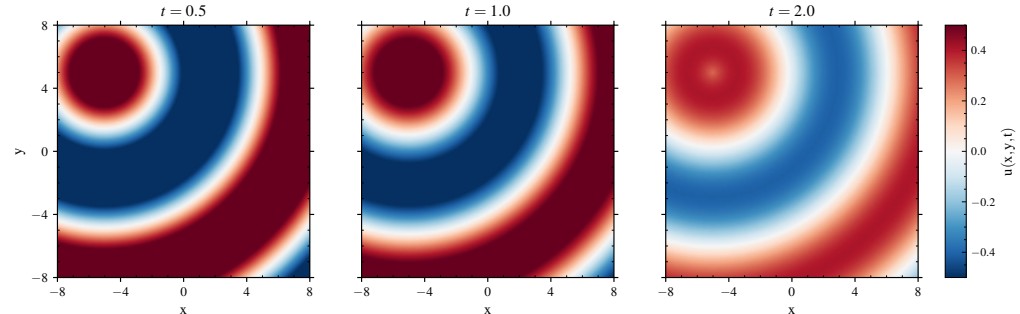

Figure 6: Contour plots of the learned solution $u(x, y, t)$ for the Damped Wave equation at time instances $t \in \{0.5, 1.0, 2.0\}$. The spatial domain is $(x, y) \in [-8, 8]^2$, and the colormap indicates the solution magnitude ranging from $-0.5$ to $0.5$. The solution is computed on a $128 \times 128$ discretization grid.

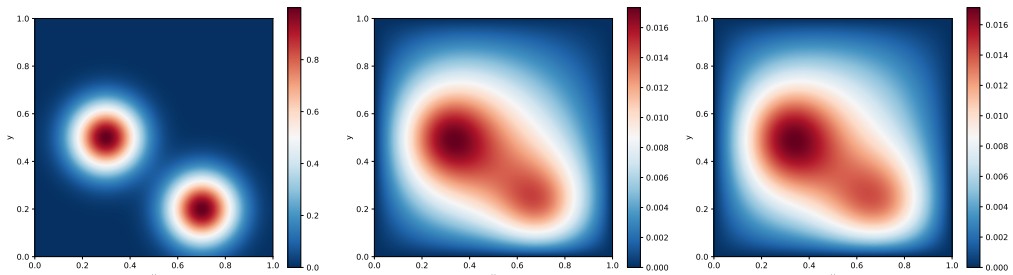

Figure 7: Comparison of numerical approximation and symbolic solution for the Poisson equation with 2 Gaussian source centers. (Left) Source term $F(x, y)$ consisting of 2 Gaussian functions centered at $(0.3, 0.5)$ and $(0.7, 0.2)$ with $\sigma = 0.1$. (Right) Solution obtained using the SIGS method. (Middle) Reference solution obtained by Finite Element Method (FEM) solution computed using FEniCS on a $100 \times 100$ mesh with P2 elements. The spatial domain is $(x, y) \in [0, 1]^2$, visualized on a $400 \times 400$ grid. The colormap indicates solution magnitude with maximum values of approximately 0.035.

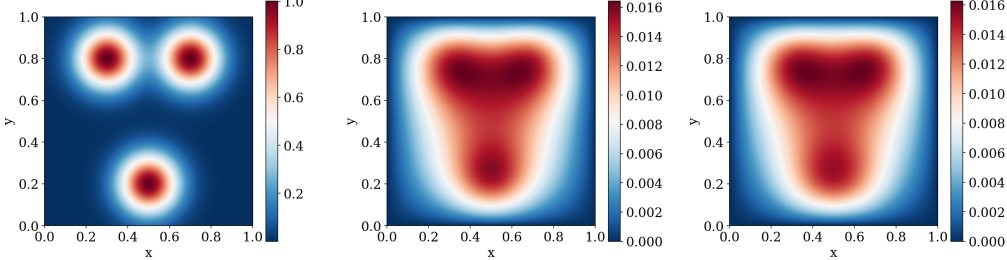

Figure 8: Comparison of numerical approximation and symbolic solution for the Poisson equation with 3 Gaussian source centers. (Left)Source term $F(x, y)$ consisting of 3 Gaussian functions centered at $(0.3, 0.8)$, $(0.7, 0.8)$, and $(0.5, 0.2)$ with $\sigma = 0.1$. (Middle) Reference solution obtained by Finite Element Method (FEM) solution computed using FEniCS on a $100 \times 100$ mesh with P2 elements .(Right) Solution obtained using the SIGS method. The spatial domain is $(x, y) \in [0, 1]^2$, visualized on a $400 \times 400$ grid. The relative $L^2$ error between FEM and SIGS solutions is approximately 1%.

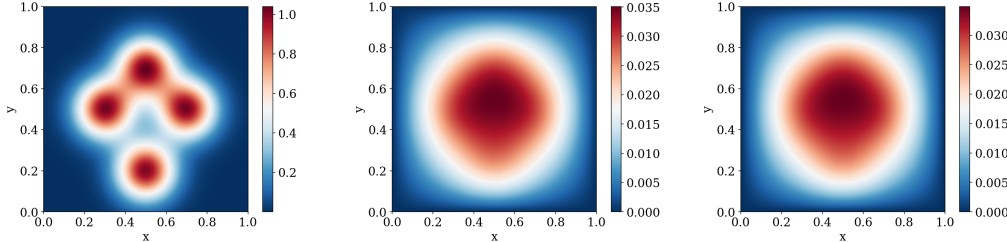

Figure 9: Comparison of numerical approximation and symbolic solution for the Poisson equation with 4 Gaussian source centers. (Left)Source term $F(x, y)$ consisting of 4 Gaussian functions centered at $(0.3, 0.5)$, $(0.7, 0.5)$, $(0.5, 0.2)$, and $(0.5, 0.7)$ with $\sigma = 0.1$s. (Middle) Reference solution obtained by Finite Element Method (FEM) solution using FEniCS on a $100 \times 100$ mesh with P2 element .(Right) Solution obtained using the SIGS method. The spatial domain is $(x, y) \in [0, 1]^2$, visualized on a $400 \times 400$ grid. The symmetric arrangement of sources produces a cross-like pattern in the solution field.

### C.8 Additional results during the review process

#### C.8.1 Korteweg-de Vries (KdV) Equation

We are going to study how grammar misspecification, e.g. missing primitive functions, and the choice of the Ansatz affect the solution capabilites of SIGS. For this purpose, we consider the Korteweg-de Vries (KdV) quation. The KdV equation is a PDE which models one-dimensional nonlinear dispersive nondissipative waves, or solitons, defined as:

$$u_t + 6uu_x + u_{xxx} = 0, \tag{7}$$

The one-soliton setup of this problem considers a single wave as it moves across the domain, and has an analytic solution of the form

$$u(x,t) = \frac{2}{\cosh^2(x-4t)}, \tag{8}$$

where cosh is not included in the grammar. We define the domain $\Omega \times T \in [-10, 10] \times [0, 1]$, and the Ansatz as

$$u(x,t) = \sum_{k=0}^{2} \phi(x,t)^k. \tag{9}$$

SIGS found the following solution:

$$0.0003734 \cdot \tanh(0.001355 \cdot t - 0.0006419 \cdot x + 0.0002936)$$

$$-2.0 \cdot \tanh^2(4.0 \cdot t - 1.0 \cdot x + 2.237e - 7) + 2.0.$$

This is a very interesting result. If we apply trigonometric identities, we see that:

$$\frac{2}{\cosh^2(x-4t)} = 2\,\mathrm{sech}^2(x-4t) = 2\,(1-\tanh^2(x-4t)) = 2-2\,\tanh^2(x-4t), \tag{10}$$

which is very close to what SIGS returns, with relative $L_2$ error $\approx 6.6 \times 10^{-6}$. Despite the fact that cosh is missing from the grammar, we can find an equivalent form very fast, wall-clock time is 36 sec.

#### C.8.2 Shallow Water Equations

The SWE equations are a hyperbolic system of PDEs which describe the flow below a pressure surface in a fluid. We use the method of manufactured solutions to construct analytical equations for the density and velocities. The solutions are coupled because of the dependence of the velocities on the density. The density $\rho(x,y,t)$ is modeled as:

$$\rho(x,y,t) = 1 + h\exp\left(-\frac{r}{w(1+t)}\right)\cos\left(k\sqrt{r} - ct\right)e^{-\alpha t},$$

where $r = (x-x_0)^2 + (y-y_0)^2$ the center of the droplet, $h$ represents the amplitude, $w$ is the Gaussian width, $k$ is the wave number, $c$ is the wave speed, and $\alpha$ is the decay rate. The terms $x_0$ and $y_0$ define the initial center of the wave. The velocity components $u_x(x,y,t)$ and $u_y(x,y,t)$ are derived using the linear shallow waters theory as:

$$u_x(x,y,t) = \rho(x,y,t) \cdot \frac{x \cdot c}{H \cdot \sqrt{r}},$$

$$u_y(x,y,t) = \rho(x,y,t) \cdot \frac{y \cdot c}{H \cdot \sqrt{r}},$$

where $H$ is a velocity scaling factor. The governing equations for the shallow water system are:

$$\text{Mass conservation:} \qquad \frac{\partial \rho}{\partial t} + \frac{\partial(\rho u_x)}{\partial x} + \frac{\partial(\rho u_y)}{\partial y} = f_\rho(x,y,t),$$

$$\text{x-momentum:} \qquad \frac{\partial(\rho u_x)}{\partial t} + \frac{\partial(\rho u_x^2 + \frac{1}{2}g\rho^2)}{\partial x} + \frac{\partial(\rho u_x u_y)}{\partial y} = f_x(x,y,t),$$

$$\text{y-momentum:} \qquad \frac{\partial(\rho u_y)}{\partial t} + \frac{\partial(\rho u_x u_y)}{\partial x} + \frac{\partial(\rho u_y^2 + \frac{1}{2}g\rho^2)}{\partial y} = f_y(x,y,t).$$

The forcing terms $f_x(x, y, t)$ and $f_y(x, y, t)$ ensure the manufactured solutions remain valid over the specified domain and time interval by compensating for natural wave decay and dissipation. The parameters of these expressions ranges provide sufficient flexibility to capture diverse wave behaviors $h \sim U(0.5, 2.0), w \sim U(0.3, 3.0), k \sim U(0.5, 8.0), c \sim U(0.3, 3.0), \alpha \sim U(0.02, 0.8), x_0, y_0 \sim U(-6.0, 6.0), H \sim U(1.0, 5.0)$. To discover a new solution of the shallow water equations, we consider $h = 0.97, w = 0.88, k = 1.78, c = 1.28, \alpha = 0.72, (x_0, y_0) = (3.77, 2.34), H = 4.46, (x, y) \in [-10, 10], t \in [0, 5]$, and $r_0 = (x - 3.77)^2 + (y - 2.34)^2$. We consider periodic boundary conditions and initial conditions the function values at time $t = 0$.

We consider Ansatze for each equation:

$$\rho(x, y, t) = f(x, y, t)g(x, y, t)h(t),$$
$$u(x, y, t) = \rho(x, y, t)s_x(x, y),$$
$$v(x, y, t) = \rho(x, y, t)s_y(x, y),$$

where $f, g, s_x s_y$ atoms from the grammar. We consider $\rho$ as part of the ansatz of $u, v$ due to the dependency between the velocity and the density we discussed earlier. We report both the true manufactured solution and the solution found by SIGS, which has the correct structural form, in terms of the values of $\theta$ in Table 21. The errors reported are:

$$\text{Rel}L^2(\rho) \approx 1.8731 \times 10^{-4}(0.0187\%),$$
$$\text{Rel}L^2(u) \approx 2.2310 \times 10^{-4}(0.0223\%), \tag{11}$$
$$\text{Rel}L^2(v) \approx 4.1783 \times 10^{-4}(0.0418\%).$$

By the problem definition, we know that the initial condition has a local support around a center $r_0$ which means that it is not physically meaningful for different atoms to have different centers. For this reason, the optimization centers are made identical for the local optimization by choosing the center of the traveling wave which is the dominant physical feature of the problem. The overall optimization time to get the solution is 3 minutes and 23 seconds.

| | $A_\rho$ | $A_{sx}$ | $A_{sy}$ | $c_{\rho x}$ | $c_{\rho y}$ | $c_{sx}$ | $c_{sy}$ | $c_{syx}$ | $c_{syy}$ | $\sigma$ | decay | freq | phase |
|------|------|------|------|------|------|------|------|------|------|------|------|------|------|
| True | 1.142 | 1.142 | 1.142 | 0.4 | -0.4 | 0.4 | -0.4 | 0.4 | -0.4 | 1.1 | 0.6 | 2.6 | 0.7 |
| Init | 1.64 | 1.91 | 0.93 | 1.02 | -0.49 | 1.02 | -0.49 | 1.02 | -0.49 | 1.96 | 0.75 | 1.92 | 0.11 |
| SIGS | 1.142 | 1.14 | 1.14 | 0.39 | -0.4 | 0.40 | -0.39 | 0.39 | -0.39 | 1.09 | 0.59 | 2.60 | 0.70 |

Table 21: Values of the parameters $\theta$ in the solutions to the SWE system: The true manufactured solution (True), the values returned after the SIGS structural search with values of $c$ adjusted (Init), and the final values found after the parameter optimization (SIGS).

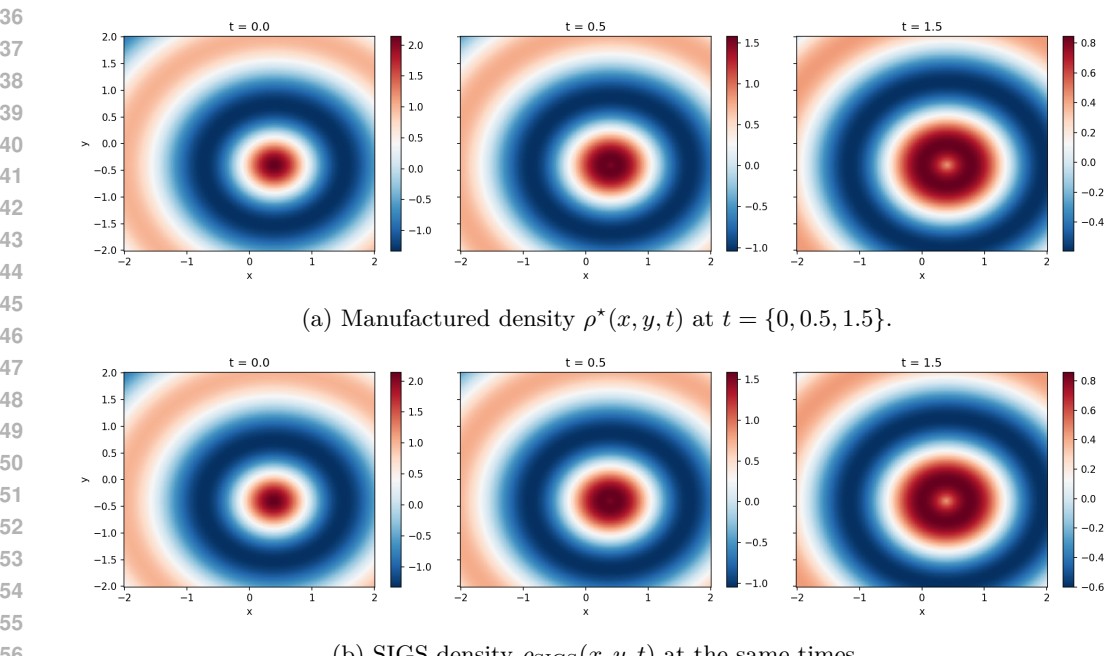

(a) Manufactured density $\rho^\star(x, y, t)$ at $t = \{0, 0.5, 1.5\}$.

(b) SIGS density $\rho_{\mathrm{SIGS}}(x, y, t)$ at the same times.

Figure 10: Shallow-water density fields. Top: manufactured solution from Eq. (SWE-MS); bottom: SIGS-refined solution. The three panels correspond to $t = 0$, 0.5, and 1.5 on the domain $(x, y) \in [-2, 2]^2$ with periodic boundary conditions.

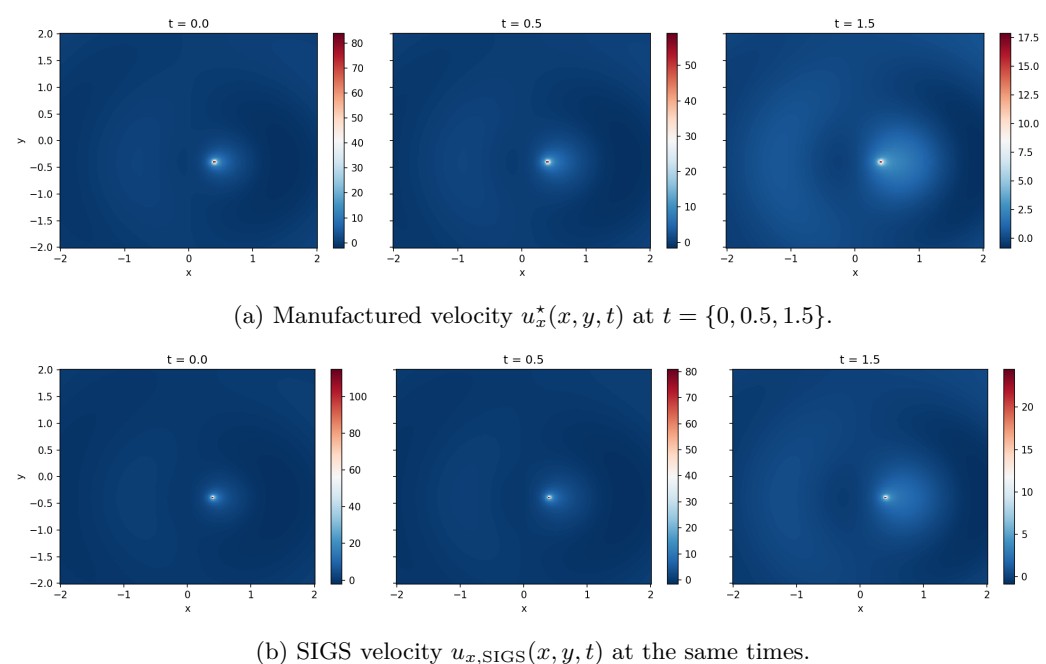

(a) Manufactured velocity $u_x^\star(x, y, t)$ at $t = \{0, 0.5, 1.5\}$.

(b) SIGS velocity $u_{x,\mathrm{SIGS}}(x, y, t)$ at the same times.

Figure 11: Shallow-water $x$-velocity fields. Top: manufactured solution; bottom: SIGS solution. Panels show $t = 0$, 0.5, and 1.5.

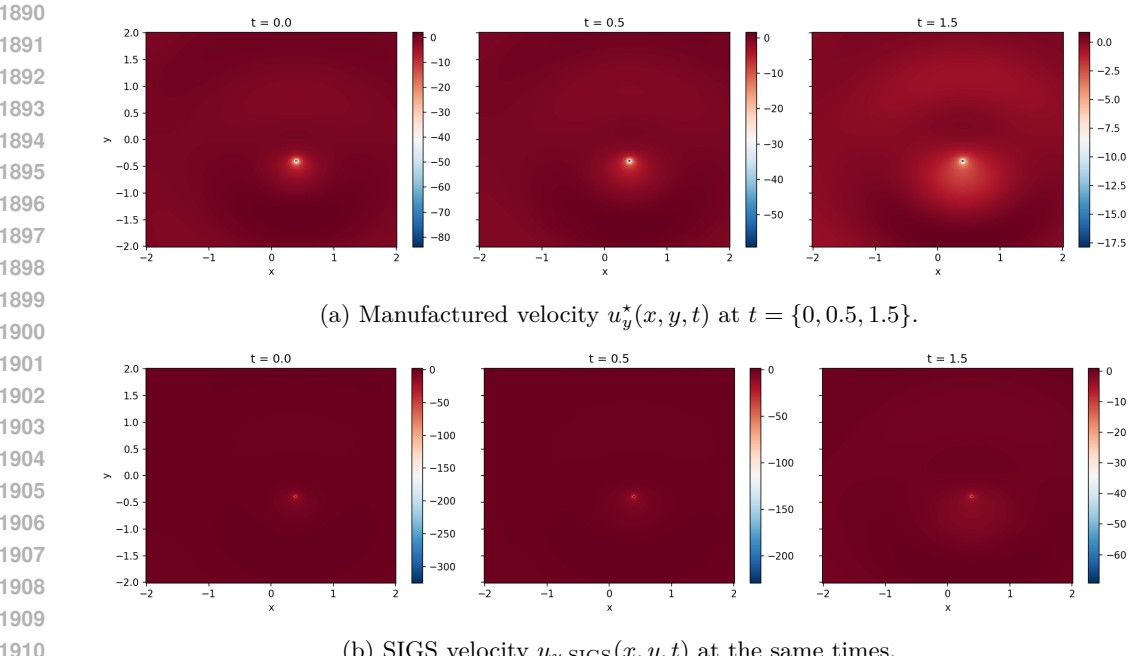

(a) Manufactured velocity $u_y^\star(x, y, t)$ at $t = \{0, 0.5, 1.5\}$.

(b) SIGS velocity $u_{y,\text{SIGS}}(x, y, t)$ at the same times.

Figure 12: Shallow-water $y$-velocity fields. Top: manufactured solution; bottom: SIGS solution. Panels show $t = 0$, 0.5, and 1.5.

### C.8.3   Compressible Euler equations (CE)

We consider the two-dimensional compressible Euler equations (in steady state) on a periodic spatial domain. We use the method of manufactured solutions (MMS) to construct closed-form analytical baselines, and derive consistent forcing terms so that the manufactured fields satisfy the Euler system exactly. SIGS is then used to rediscover these fields by minimizing PDE residuals.

Let $\Omega = [0, 1]^2$ with $(x, y) \in \Omega$. We define the conservative state

$$U(x, y) = \begin{pmatrix} \rho(x, y) \\ \rho(x, y)\, u(x, y) \\ \rho(x, y)\, v(x, y) \\ E(x, y) \end{pmatrix}, \qquad \rho > 0, \quad (u, v) \in \mathbb{R}^2,$$

where $\rho$ is density, $(u, v)$ are the velocity components, and $E$ is the total energy density. In the MMS setting we use steady (time–independent) solutions, hence $\partial_t(\cdot) = 0$. The forced steady Euler system reads

$$\nabla \cdot F(U) = \begin{pmatrix} f_\rho \\ f_u \\ f_v \\ f_E \end{pmatrix} \qquad \text{in } \Omega, \tag{12}$$

with flux

$$F(U) = \begin{pmatrix} \rho u & \rho v \\ \rho u^2 + p & \rho u v \\ \rho u v & \rho v^2 + p \\ u(E + p) & v(E + p) \end{pmatrix}.$$

Equivalently, in component form,

$$\partial_x(\rho u) + \partial_y(\rho v) = f_\rho,$$
$$\partial_x(\rho u^2 + p) + \partial_y(\rho uv) = f_u,$$
$$\partial_x(\rho uv) + \partial_y(\rho v^2 + p) = f_v,$$
$$\partial_x\big(u(E+p)\big) + \partial_y\big(v(E+p)\big) = f_E.$$

We close the system with the ideal-gas equation of state

$$p = (\gamma - 1)\big(E - \tfrac{1}{2}\rho(u^2 + v^2)\big), \qquad \gamma = 1.4. \tag{13}$$

We impose periodic boundary conditions on all primitive variables: for $\phi \in \{\rho, u, v, p\}$,

$$\phi(0, y) = \phi(1, y), \qquad \phi(x, 0) = \phi(x, 1).$$

The Ansätze used by SIGS for each variable are sums of grammar atoms, with exponential envelopes for $\rho$ and $p$:

$$\rho(x, y) = \exp\Big(\sum_{i=1}^{6} f_i(x, y)\Big),$$

$$u(x, y) = \sum_{i=1}^{6} g_i(x, y),$$

$$v(x, y) = \sum_{i=1}^{6} h_i(x, y),$$

$$p(x, y) = \exp\Big(\sum_{i=1}^{6} k_i(x, y)\Big),$$

where $f_i, g_i, h_i, k_i$ are spatial atoms generated by the grammar.

| Field | Manufactured | | Optimized | | Rel. $L^2$ (%) |
|---|---|---|---|---|---|
| $\rho$ | $\exp\big(-0.0887\,\sin(\pi x)\sin(\pi y)$ | $+$ | $\exp\big(0.273\,\sin(\pi x)\sin(2\pi y)$ | $+$ | 10.8 |
| | $0.504\,\sin(\pi x)\sin(2\pi y)$ | $+$ | $0.217\,\sin(2\pi x)\sin(\pi y)$ | $+$ | |
| | $0.259\,\sin(2\pi x)\sin(\pi y)$ | $+$ | $0.229\,\sin(2\pi x)\sin(2\pi y)$ | $+$ | |
| | $0.140\,\sin(2\pi x)\sin(2\pi y)\big)$ | | $2.18 \times 10^{-3}\,\sin(2\pi y)\cos(\pi x)\big)$ | | |
| $u$ | $\pi\big(-0.243\,\sin(\pi x)\sin(\pi y)$ | $-$ | $-0.929\,\sin(\pi x)\sin(\pi y)$ | $-$ | 9.93 |
| | $0.385\,\sin(\pi x)\sin(2\pi y)$ | $-$ | $1\,\sin(\pi x)\sin(2\pi y)$ | $+$ | |
| | $0.494\,\sin(2\pi x)\sin(\pi y)$ | $+$ | $0.0518\,\sin(\pi x)\cos(3\pi y)$ | $-$ | |
| | $0.518\,\sin(2\pi x)\sin(2\pi y)\big)$ | | $1.50\,\sin(2\pi x)\sin(\pi y)$ | $+$ | |
| | | | $1.56\,\sin(2\pi x)\sin(2\pi y)$ | | |
| $v$ | $\pi\big(0.0715\,\sin(\pi x)\sin(\pi y)$ | $+$ | $0.718\,\sin(\pi x)\sin(2\pi y)$ | $+$ | 9.84 |
| | $0.233\,\sin(\pi x)\sin(2\pi y)$ | $-$ | $2.05\,\sin(2\pi x)\sin(2\pi y)$ | $-$ | |
| | $0.536\,\sin(2\pi x)\sin(\pi y)$ | $+$ | $0.307\,\sin(4\pi x)\sin(\pi y)$ | $-$ | |
| | $0.665\,\sin(2\pi x)\sin(2\pi y)\big)$ | | $1.22\,\sin(\pi y)\cos(\pi x)$ | $+$ | |
| | | | $1\,\sin(\pi y)\cos(3\pi x)$ | | |
| $p$ | $\exp\big(0.235\,\sin(\pi x)\sin(\pi y)$ | $-$ | $\exp\big(-0.389\,\sin(\pi x)\sin(2\pi y)$ | $-$ | 12.1 |
| | $0.322\,\sin(\pi x)\sin(2\pi y)$ | $-$ | $0.354\,\sin(2\pi x)\sin(\pi y)$ | $-$ | |
| | $0.356\,\sin(2\pi x)\sin(\pi y)$ | $-$ | $0.410\,\sin(2\pi x)\sin(2\pi y)\big)$ | | |
| | $0.448\,\sin(2\pi x)\sin(2\pi y)\big)$ | | | | |

Table 22: Manufactured vs. optimized fields with coefficients truncated to three significant figures.

### C.8.4 Results from classical methods

We assess the difficulty of the PDEs considered in this manuscript by trying to solve them using different automated procedures, such as the state-of-the-art Computer Algebra System Mathematica Wolfram Research, Inc. (2024). Since manual solving might also lead to success

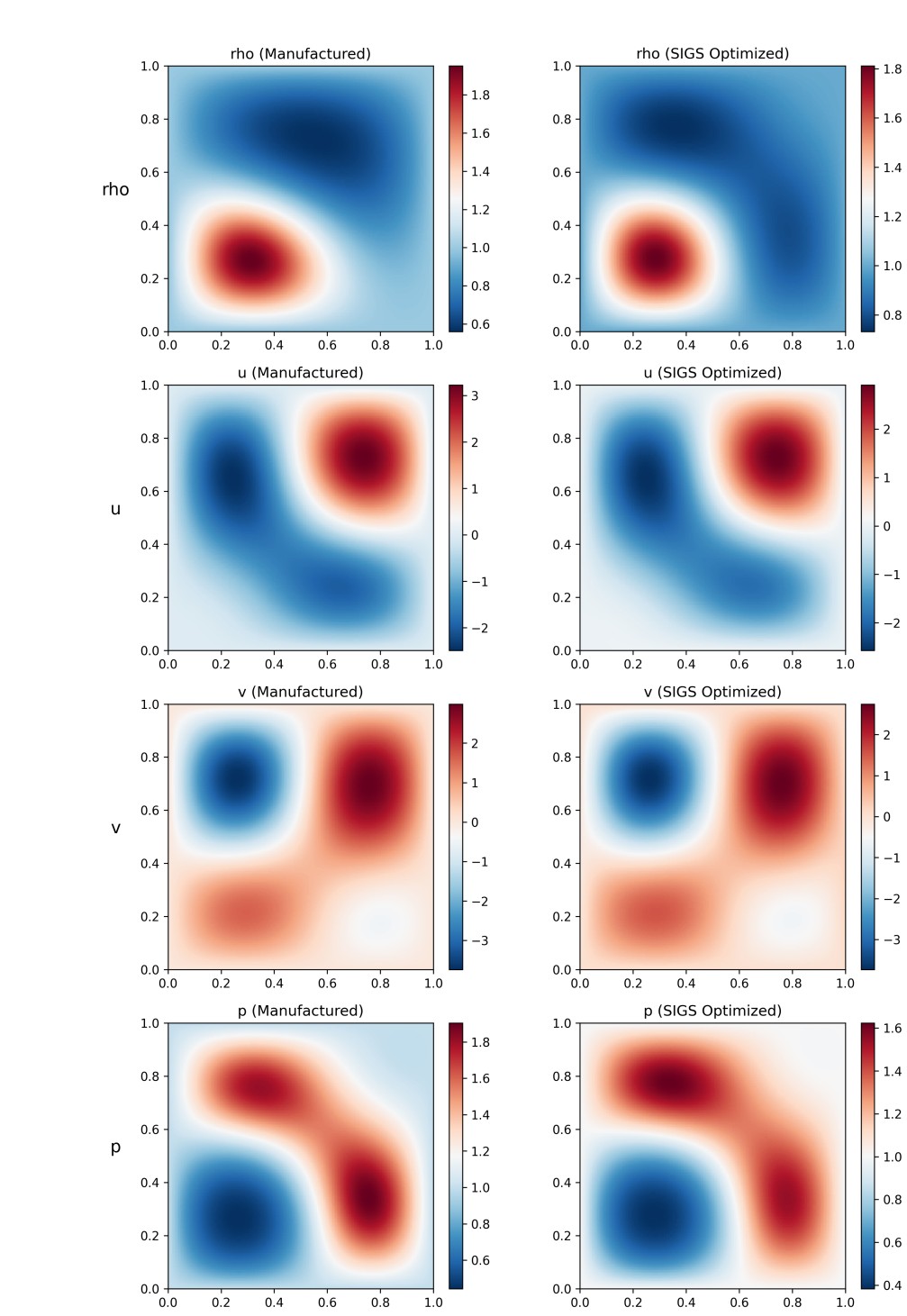

Figure 13: Compressible Euler manufactured vs. SIGS-optimized fields. Each row corresponds to one state variable $(\rho, u, v, p)$, and columns show the manufactured reference (left) and the SIGS-refined solution (right) evaluated on the same grid. SIGS recovers the spatial structure and magnitude of all four coupled fields simultaneously, yielding a near-indistinguishable match to the manufactured solution.

in different cases, we ask ChatGPT 5.1 (Extended Reasoning) to solve the problems as a proxy for human ingenuity. We use the DSolveValue function of Mathematica on the same suite of problems that we tested SIGS on. We report the results in Table 23. The following solutions were found:

**Mathematica: Dirichlet heat equation.** Mathematica returns the standard Fourier–sine series solution of the Dirichlet heat problem:

$$u(x,t) = \sum_{n=1}^{\infty} b_n \exp\left(-D\left(\tfrac{n\pi}{L}\right)^2 t\right)\sin\left(\tfrac{n\pi x}{L}\right), \tag{14}$$

where the coefficients $b_n$ are the sine–series coefficients of the initial condition (computed symbolically or numerically by DSolveValue):

$$b_n = \frac{2}{L}\int_0^L u_0(\xi)\sin\left(\tfrac{n\pi\xi}{L}\right)d\xi. \tag{15}$$

The infinite series solution found is written as:

$$\begin{aligned}
u(x,t) = \sum_{K=1}^{\infty} &\frac{e^{-3.52tK^2}\sin(2.24K)}{1.69\times10^{63}K^6 - 5.94\times10^{64}K^4 + 4.40\times10^{65}K^2 - 3.82\times10^{65}} \\
&+ \frac{\left(-1.28\times10^{64}K^4 + 2.57\times10^{64}K^2 - 8.38\times10^{65}\right)\sin(3.14K)}{1.69\times10^{63}K^6 - 5.94\times10^{64}K^4 + 4.40\times10^{65}K^2 - 3.82\times10^{65}} \\
&+ \frac{K\left(-1.68\times10^{49}K^4 + 2.78\times10^{50}K^2 - 3.62\times10^{50}\right)\cos(3.14K)}{1.69\times10^{63}K^6 - 5.94\times10^{64}K^4 + 4.40\times10^{65}K^2 - 3.82\times10^{65}}\Bigg].
\end{aligned} \tag{16}$$

where $K$ the mode index n in a sine expansion in, meaning $\sin(\pi n x)$, matching Dirichlet BC in $x$.

**Mathematica: Poisson-Gauss (2 centers).** Mathematica returns an eigenfunction/Green's-function representation based on the sine basis in $x$ and the corresponding 1D Green's function in $y$:

$$u(x,y) = 2\sum_{n=1}^{\infty}\sin(n\pi x)\int_0^1 G_n(y,\eta)\left(\int_0^1\sin(n\pi\xi)f(\xi,\eta)d\xi\right)d\eta, \tag{17}$$

where, for each Fourier mode $n \geq 1$, $G_n(y,\eta)$ is the 1D Green's function for the operator $\partial_{yy} - (n\pi)^2$ on $y \in (0,1)$ with homogeneous Dirichlet boundary conditions:

$$G_n(y,\eta) = \frac{1}{n\pi\sinh(n\pi)}\begin{cases}\sinh(n\pi y)\sinh\left(n\pi(1-\eta)\right), & 0 \leq y \leq \eta \leq 1, \\ \sinh(n\pi\eta)\sinh\left(n\pi(1-y)\right), & 0 \leq \eta \leq y \leq 1.\end{cases} \tag{18}$$

This is exactly the Green's-function representation

$$u(x,y) = \iint_{(0,1)^2} G(x,y;\xi,\eta)f(\xi,\eta)d\eta, \tag{19}$$

with $G$ expanded in the sine basis in $x$.

**ChatGPT: Poisson-Gauss (2 centers).** ChatGPT returned the following solution after a reasoning time of 8 m 16 s:

$$\begin{aligned}
u_{ChatGPT}(x,y) = -0.01\Big(&\log\big(\sqrt{(x-0.3)^2 + (y-0.5)^2}\big) \\
&- 0.5\,\mathrm{Ei}\Big(-\frac{(x-0.3)^2 + (y-0.5)^2}{0.02}\Big) \\
&+ \log\big(\sqrt{(x-0.7)^2 + (y-0.2)^2}\big) \\
&- 0.5\,\mathrm{Ei}\Big(-\frac{(x-0.7)^2 + (y-0.2)^2}{0.02}\Big)\Big),
\end{aligned} \tag{20}$$

shown in Fig. 14 and corresponding to a relative $L^2$ error of 1.576e+02%. Upon questioning the result, it admits: 'Exactly: the closed-form solution I gave you did not enforce the Dirichlet BC=0, u=0'.

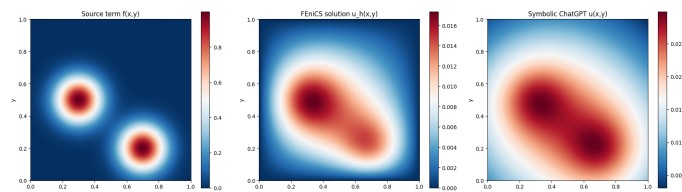

Figure 14: Performance of ChatGPT on the PG-2 problem: Initial condition, FEniCS solution, and approximation returned by ChatGPT.

ChatGPT: Burgers.   ChatGPT correctly finds the analytical solution

$$u_{ChatGPT}(x, y) = 0.86 - 0.6 \tanh\big(30x - 25.8t - 9.9\big). \tag{21}$$

Here we need to clarify that ChatGPT actually "cheats" in the sense that it doesn't really solve the PDEs with manufactured solutions, but it uses the initial and boundary conditions to reverse engineer the manufactured solution.

Table 23: Success of classical solution methods on a selection of problems.

| Problem | Mathematica | ChatGPT |
|---|---|---|
| Burgers | No | Yes |
| Damped Wave | No | |
| Diffusion equation | Yes (Infinite Series) | |
| Poisson–Gauss (2 centers) | Yes (Infinite Series) | Approximation |
| KdV equation | No | |

Table 24: Performance metrics for ChatGPT for two mathematical problems.

| Problem | Relative $L^2$ Error | Reasoning Time |
|---|---|---|
| Burgers | 0.0% | 7 m 25 s |
| Poisson–Gauss (2 centers) | 1.576e+02% | 8 m 16 s |

## C.9   Additions to the computational performance assessment

In Table 3, the relative $L^2$ error between the SIGS and FEniCS solutions was computed on the native $100 \times 100$ grid. We add the residuals $R(u)$ of both methods on different grids for the solutions by SIGS and FEniCS in Table 25. In general, FEniCS has a lower residual error, justifying to consider it as the reference in the relative error computation. For finer meshes, we see the FEniCS residual decreasing, while is stays constant for SIGS. An extended convergence study of $R(u)$ for FEniCS can be found in Table 17.

Table 25: Comparison of residuals $R(u)$ between SIGS and FEniCS for problems PG-2, PG-3 and PG-4, evaluated on two mesh resolutions.

| Mesh | Model residual | PG-2 | PG-3 | PG-4 |
|---|---|---|---|---|
| $128 \times 128$ | $R(u_{\text{FEniCS}})$ | $2.924 \times 10^{-4}$ | $3.663 \times 10^{-4}$ | $4.070 \times 10^{-4}$ |
| $128 \times 128$ | $R(u_{\text{SIGS}})$ | $4.491 \times 10^{-2}$ | $4.476 \times 10^{-2}$ | $3.617 \times 10^{-2}$ |
| $256 \times 256$ | $R(u_{\text{FEniCS}})$ | $8.048 \times 10^{-5}$ | $1.034 \times 10^{-4}$ | $1.133 \times 10^{-4}$ |
| $256 \times 256$ | $R(u_{\text{SIGS}})$ | $4.493 \times 10^{-2}$ | $4.490 \times 10^{-2}$ | $3.618 \times 10^{-2}$ |

In terms of runtimes, the model runs in Table 4 arrive at different levels of accuracy. In order to facilitate the comparison, we stop the SIGS optimization at errors comparable to the FEniCS solution in Table 2 and report this cropped SIGS runtime in Table 17.

Table 26: Updated SIGS runtimes: time to discover an analytical solution at comparable relative $L^2$ error $\approx 10^{-2}$–$10^{-3}$.

| PDE problem | Rel. $L^2$ (SIGS) | Total SIGS time (s) |
|---|---|---|
| Burgers | $4.85 \times 10^{-3}$ | 11.62 |
| Diffusion | $2.59 \times 10^{-3}$ | 14.67 |
| Damped wave | $1.44 \times 10^{-2}$ | 8.95 |

LLMs usage in the manuscript   The authors used LLMs to polish grammar and spelling through Overleaf tools and independent LLMs services.

