# OpenReview forum: "Neuro-Symbolic AI for Analytical Solutions of Differential Equations"
_ICLR.cc/2026/Conference — Submitted to ICLR 2026_

### Official Review · Reviewer_3xsx · 2025-10-31

**Soundness:** 2
**Presentation:** 3
**Contribution:** 2
**Rating:** 4
**Confidence:** 2

**Summary:**

This paper proposes SIGS (Symbolic Iterative Grammar Solver), a neuro-symbolic framework for discovering analytical (closed-form) solutions of PDEs.

The key idea is to construct a grammar-based atom library and use a Topology-Regularized Grammar VAE (TGVAE) to learn a smooth latent space of symbolic expressions. SIGS then performs a two-stage search:

- Stage 1: cluster latent codes and interpolate within clusters to identify promising candidate structures;

- Stage 2: refine constants through gradient-based optimization. Experiments across elliptic, parabolic, and hyperbolic PDEs show that SIGS can recover exact or near-exact symbolic solutions and outperform recent symbolic discovery baselines (HD-TLGP, SSDE) in accuracy and efficiency.

**Strengths:**

1. Novel neuro-symbolic design: Integrating grammar-level atoms, latent interpolation, and geometric/topological constraints is a creative and elegant framework.
2. Strong empirical performance: On selected PDE benchmarks, SIGS accurately reconstructs analytical expressions, sometimes achieving machine precision.
3. Well-motivated two-stage pipeline: The coarse-to-fine (structure search → constant refinement) approach is conceptually clean and provides interpretability.

**Weaknesses:**

1. Dependence on handcrafted Ansatz and grammar: The system assumes access to an appropriate symbolic library. When the correct building blocks are missing, SIGS likely fails; this is not tested quantitatively.
2. Limited fairness and transparency in baselines: It is unclear whether competing methods (e.g., HD-TLGP, SSDE, FEniCS) had access to the same symbols, constants, or data budgets. Table 2 comparisons could be biased.
3. Scalability and robustness not demonstrated: The method is validated on PDEs with known, smooth analytic forms. Its behavior on nonlinear, discontinuous, or noisy settings is untested.
4. Motivational gap vs. numerical solvers: Since SIGS can be slower and less accurate than classical solvers (e.g., FEniCS), the paper should articulate clearer motivation for discovering symbolic forms instead of direct numerical solutions.

**Questions:**

1. Ansatz sensitivity: How does SIGS perform when the provided Ansatz or atom library omits key operators (e.g., removing tanh for Burgers)? Please quantify degradation.
2. Residual evaluation: Have you computed the residuals of the recovered analytical expressions of Poisson equation in Table 3? Please compare residuals between SIGS and FEniCE under different spatial resolutions. And how about other boundary conditions, such as Neumann and periodic?
3. FEniCS resolution: What spatial resolution was used for FEniCS in Table 2? How does the FEniCS error and runtime change with mesh refinement, and how do these compare to SIGS?
4. Motivation for symbolic discovery: Given that FEniCS achieves higher accuracy faster, what is the practical motivation for SIGS? Is interpretability the main benefit, or does SIGS generalize across PDE families?
5. Failure modes: Can you show examples where SIGS fails or outputs incorrect forms, and analyze why?

Typo: Inconsistent of poisson equation form between 1175 line and 1313 line.

---

> ### Author Response · Authors · 2025-11-21
>
> We would like to thank the reviewer for the thoughtful and positive assessment of our contributions which highlight the conceptual novelty, empirical thoroughness, and clarity of the work. Below, we address the reviewer's concerns and questions.
>
> 1. Assumptions on the symbolic library (W1, Q1).
>
> We would like to clarify two points. First, SIGS does not require problem-specific grammar or atom libraries; instead, we pretrain the GVAE with a very general library which is fixed across all experiments. This is in contrast to alternative approaches (SSDE and HD-TLGP) which require a user-specified set of atoms. When domain intuition exists, SIGS uniquely allows it to be incorporated through the ansatz structure. However, as we illustrate in Sec. C.2, the ansatz may remain very general and still allow our approach to recover exact solutions.
> We agree it is important to study the performance of the model under mis-specification or missing atoms. To further illustrate robustness to imperfect grammars, we now include a new KdV 1-soliton experiment (details in the revision) where the true primitive, $\cosh$, is not a part of the library. **Without any re-training** SIGS constructs a compact tanh-based surrogate that achieves a very close approximation to the exact solution, with a relative $L_2$ error of $6.6. \ 10^{-6}$, demonstrating that SIGS can still discover accurate approximations even when the exact closed-form is not able to be represented by the grammar.
>
> 2. Fairness of comparisons (W2).
>
> As outlined in our work, we fix a **generous computational budget across all problems (2 hours)**, and we are confident that any approach which has not converged within that limit will not show improved results with a longer run. As for the access to symbols and constants, which is fixed for SIGS, we provide the alternative approaches to the baseline methods with a set of atoms which is tailored for each problem. This is how each method is presented in the original work, i.e. if a solution is expected to include sin but not tanh, then only sin is available to the model. A restricted library serves a crucial advantage in lowering the search complexity and plays a key role in the success of the other approaches. **We manually restrict the library of the baselines to only consist of the essential functions to be faithful to their initial design**. SIGS, however, was conceptualized and designed with a general library in mind. Through the encoding of symbols in a latent space, we can employ a much larger, general set of atoms across all problems. Thus SIGS accepts a large library of atoms that are build once and are the same for all PDEs without manually removing or adding problem specific functions. We also investigate two protocols for HD-TLGP: Protocol 1 which corresponds to the original method and Protocol 2 where we provide atoms from the SIGS library for all experiments. This way we show that naively searching the combinatorial space of atoms is not enough to discover PDE solutions even in simple cases. Given that the fixed data budgets are very generous, we can craft the libraries of the baselines to only have the essential operators, and also consider protocols where we provide atoms to reduce the complexity, we strongly maintain that there is no bias in the results of Table 2.

---

> > ### Author Response · Authors · 2025-11-21
> >
> > 3. Performance on more difficult problems (W3).
> >
> > We have had a similar discussion with Reviewer 1nRi. We provide the same remarks here as well.
> >
> > The reviewer is correct in remarking that we evaluate extensively on linear PDEs with known solutions. This is intentional. SIGS generates symbolic expressions; therefore, we must verify that it can recover ground-truth closed forms before evaluating on more difficult problems, such as nonlinear PDEs or coupled systems.
> >
> > We agree the nonlinear setting is more compelling. We have added several experiments, namely: \\
> > - Korteweg-De Vries (KdV, nonlinear) soliton discovery
> > - Shallow Water Equations (SWE, nonlinear coupled system) A purely hyperbolic system which requires specialized numerical methods to be solved, due to stability issues, and it is time consuming, due to Courant-Friedrichs-Lewy (CFL) conditions.
> > - Compressible Euler Equations (CE, nonlinear coupled system) also a hyperbolic system, this experiment investigates periodic boundary conditions in a remarkably challenging problem setting.
> >
> > In the KdV experiment, we are able to identify a closed-form solution with a relative L2 error of $6.6 \ 10^{-6}$, even though $\cosh$ is absent from the atom library. Instead, SIGS identifies a solution which approximates these terms via $\tanh$.
> > For the SWE experiment, we successfully recover analytical solutions for the density, x-, and y-velocity. This is a significant step forward in automated solution discovery, and surpasses one of the main limitations of SSDE. As stated by the authors of this work, SSDE routinely fails at solving coupled systems, due to the combinatorial growth in the solutions space, as mentioned in the OpenReview discussion of the associated article. SIGS is the first method that, to our knowledge, can be applied to systems of equations, let alone achieve such accuracy.
> > For the Compressible Euler experiment, we recover analytical solutions to this exceptionally difficult problem with only 10\% relative error. One particularly challenging aspect of this problem is the periodic boundary conditions which do not provide strong constraints to inform the solution within the domain, as various combinations of $sin$ and $cos$ are admissible. Nonetheless, the recovered solutions closely match qualitative properties of the exact solutions, as illustrated in the figures provided in Appendix C.8.3.
> >
> > Additionally, we would like to reiterate that we have already explored other difficult problems, from a perspective of finding analytic solutions, within the work. The Poisson-Gauss experiment, for example, does not admit closed-form solutions, yet SIGS identifies a simple analytical expression which closely matches the numerical solution. Likewise, we must reiterate that the atom library is not modified per-problem, and $\tanh$ is not added explicitly to handle the solution to the Burgers' equation; instead, we choose a general atom library a-priori and apply this to all experiments. Of course, it is possible to retrain the GVAE with a new library; however, we aim to demonstrate a new approach which can converge without an explicit, user-defined restriction to only those atoms which are necessary for the final solution, as has been the case in the literature before the proposal of SIGS.
> >
> >  We demonstrate the difficulty of automatically discovering a closed-form solution for the Poisson-Gauss problem, by also considering the state-of-the-art Computer Algebra System Mathematica and ChatGPT 5.1 (Extended Reasoning) that both fail  to provide an expression that is close to the numerical solution, Mathematica provides an infinite series and ChatGPT an expression with 157$\%$ relative $L_2$ error. We performed this analysis trying to get the solutions using, Mathematica and ChatGPT, for all PDEs and present the results in Appendix C.8. Mathematica fails to provide a closed-form solution for all problems, and ChatGPT "cheats" in the sense that it uses the boundary and initial conditions to construct the manufactured solution.

---

> > > ### Author Response · Authors · 2025-11-21
> > >
> > > 4. Differentiation with numerical solvers (W4, Q3, Q4).
> > >
> > > First, we would like to point out that the threshold for convergence for SIGS is $R(u) = 10^{-16}$ while FEniCS returns $R(u) = 10^{-3}$. If we employ the same residual error criterion as FEniCS, SIGS converges much quicker, take 11.6, 14.7, and 8.9 seconds for the Burgers', Diffusion, and Damped Wave experiments, which is comparable to FEniCS. We should mention that SIGS handles files and reads from memory, which are operations that are not currently optimized. In the future, we will optimize the code for performance.
> > >
> > > *We emphasize that having approximate closed-form solutions is really important because traditional analytical solution techniques **either provide a solution or fail completely**. There is no space in-between. SIGS has the ability to also occupy that space when an analytical solution is not available. SIGS combines the interpretability of analytical approaches with the flexibility of numerical approximations.*
> > >
> > > A table with these results has been added to the revised text. Additionally, FEniCS convergence results with respect to the resolution are provided in Table 17 of the original manuscript.
> > >
> > > 5. Failure modes (Q5).
> > >
> > > Failure modes are related to the ability of SIGS to represent a function with the given atoms, and Ansatz choices. For example, the Burgers equation has a unique solution that involves a $\tanh$ function. Assume now that we remove the $\tanh$ function from the grammar and have no means of approximating it using some trigonometric identity, e.g. $\cosh$, $\sinh$. In general, the Burgers solution in the case we present here which is almost a step function, can be approximated if we considered the Ansatz to be a summation of $\sin$ and $\cos$ functions, a Fourier series expansion, but even in this case, we would require an enormous amount of modes. In this case, SIGS will fail catastrophically  because it is mathematically impossible to construct such an expansion due to the combinatorial explosion of complexity.
> > >
> > > However, in the KdV example, SIGS very accurately approximates the $\cosh$ function using $\tanh$ with a very general Ansatz choice and without biasing the search to take advantage of the identity in any way. So, to answer your question, if SIGS has the means to accurately approximate information that it is missing from the library, it will do so with high accuracy, if not it will fail catastrophically. That is why we consider a very rich grammar, to have high representation power.
> > >
> > >
> > >
> > > We hope to have addressed all questions and concerns of the reviewer within this discussion, and are happy to provide any additional clarification. We kindly request the reviewer to update their assessment if they feel their concerns have been adequately addressed.

---

### Official Review · Reviewer_9NH9 · 2025-10-31

**Soundness:** 2
**Presentation:** 2
**Contribution:** 2
**Rating:** 2
**Confidence:** 4

**Summary:**

The aim of this paper is to automatically find analytical solutions/closed-form expressions of PDE without relying on observational data, and give a scalable solution to the combinatorial explosion and illegal expression problems of traditional symbolic tree search.
Methods: SIGS based on formal grammar and topological regularization grammar VAE (TGVAE) are proposed: The limited feasible expression is constructed with the hierarchy of "Ansatz+atoms", and the discrete expression is embedded into the smooth latent space for structure search and parameter refinement.
Experiments show that the proposed method is superior to the strong baseline in accuracy and efficiency on multiple classes of PDE, and has the advantages of data independence, interpretability, and scalability.

**Strengths:**

1. This paper proposes A method to obtain the analytical solution of PDE (although there is more than one such work, for example, Closed-form Solutions: A New Perspective on Solving Differential Equations). Unlike previous methods, the generated expressions are "mathematically valid and physically meaningful".


2. SIGS seems to give good results

**Weaknesses:**

1. I think the biggest innovation of this paper is that it is possible to obtain an analytical solution of a PDE, but unfortunately, this idea has been previously worked on by many people, such as Closed-form Solutions: A New Perspective on Solving Differential Equations. And this article doesn't even cite that article!

2, it is mentioned that the generated expressions are "mathematically valid and physically meaningful", which is not new in the field of symbolic regression and has also been well solved.

3. It is felt that this method is a combination of existing methods and is not too innovative.

**Questions:**

Q1: Please analyze how your way of making expressions "mathematically valid and physically meaningful" differs from previous symbolic regression methods, and what are your innovations and advantages?

Q2: As you mentioned in your article, many of the expressions obtained by previous GP and RL-based methods are invalid, but previous symbolic regression methods impose constraints and handle them, and work well. So I don't think your assumption is meaningful.

---

> ### Author Response · Authors · 2025-11-21
>
> We thank the reviewer for the constructive feedback and for recognizing both the quality of our results and the importance of generating mathematically valid and physically meaningful analytical solutions to PDEs. Below, we address each concern.
>
> 1. Prior work is thoroughly cited and used as a baseline (W1).
>
> The reviewer states that we did not cite "Closed-form solutions: A new perspective on solving differential equations."
> This is a misunderstanding: The paper is explicitly cited and discussed on *lines 47-49* of our submission. Moreover, the method presented in that work (SSDE) is one of our *primary baselines* and is evaluated extensively throughout the paper.
>
> We have the utmost respect for SSDE and the growing body of work on automated analytic PDE solving. However, as the SSDE authors themselves state (within the OpenReview discussion of the associated work, link provided below), this is an emerging field with significant open challenges, which include:
> - difficulty handling larger libraries (SIGS handles a single large library for all problems),
> - limited ability to solve coupled PDE systems (We solve the Shallow Waters Equations which is a coupled non-linear problem using SIGS with less than 0.01 $\%$ relative $L_2$ error as well as the Compressible Euler Equations with periodic boundary conditions and only c. 10\% error across all variables.),
> - sensitivity to hand-specified library design (We consider the KdV example to show that SIGS finds an approximation of $10^{-6}$ relative $L_2$ error for the KdV equation whose solution primitive $\cosh$ is not in the atoms library).
> Please see the General Comments and updated manuscript Appendix C.8 for more details on the additional experimental results.

---

> > ### Author Response · Authors · 2025-11-21
> >
> > 2. Novel contributions of SIGS (W2, W3, Q1).
> >
> > The reviewer suggests that the proposed approach is simply a combination of existing techniques and that generating valid expressions is "not new", and a well-solved problem in symbolic regression.
> >
> > First, we find it necessary to clarify the difference between SR and our work. SR is a fundamentally different problem, which aims to fit expressions to data. We solve a symbolic solution problem: finding expressions that satisfy PDEs and their boundary/initial conditions without any observational data. As a result, the constraints are fundamentally different.
> >
> > We respectfully disagree with the reviewer that generating valid expressions is a well-solved problem in symbolic regression. Sampling valid expressions requires defining a set of rules that the generation needs to follow, for example, a grammar, or more generally a constrained decoding approach. This is a very active research area from LLMs (Geng, S., Josifoski, M., Peyrard, M. and West, R., 2023. Grammar-constrained decoding for structured NLP tasks without finetuning. arXiv preprint arXiv:2305.13971.) to symbolic regression (see Kissas et. al. 2024, for an overview), as this problem is very difficult. Approaches that combine an already valid set of equations, for example SINDy, or EQL, do, indeed, provide only valid expressions but they are limited to expert defined families of functions.
> >
> > Overall, our contributions are distinct from the literature of symbolic solution in three ways.
> >
> > i. SIGS goes far beyond "valid syntax": they embed functional structure directly into the search space, a concept which has been entirely lacking in automated discovery up to this point.
> > This is done through structurally coherent expressions that respect a physically motivated Ansatz. More specifically, SIGS introduces:
> > - a formal grammar (Sec. A.1.2) for global syntactic correctness,
> > - a strict semantic filtering for domain errors (Sec. A.1.3).
> >
> >
> > ii. A major limitation of discrete sampling approaches, SSDE, HD-TLGP, up to this point is their reliance on manually curated, small, and task-specific atom sets. In contrast, SIGS uses a large, general-purpose library. This is embedded in a latent space via a GVAE, enabling the efficient search for closed-form solutions. It is even able to construct terms which approximate the behavior of atoms not explicitly included in the library. We illustrate this with an experiment of the KdV equation, where details may be found in the discussion with Reviewer 1nRi and in the updated manuscript. This generality enables SIGS to scale beyond the scope of prior work.
> >
> > iii. SIGS constructs a continuous latent space that is geometrically regularized to be a convex enclosure that discourages latents from leaving the data-supporting region, suppress spurious topological artifacts, and have nearby latent codes to behave predictably. SIGS performs a global adaptive search on a continuous manifold for skeleton expressions and then a local gradient based optimization of the coefficients of the skeleton.
> >
> > Together, these elements go far beyond a combination of existing ideas. SIGS constitutes a new search paradigm for analytic PDE solution discovery and provides new insights which could be combined with approaches explored in other works, for example SSDE, in future work. To directly address the reviewer's statement, "it is felt that this method is a combination of existing methods an is not too innovative," we contend that SIGS contributes novelty in three fundamental dimensions:
> > - a novel flexible Ansatz and atom structure that facilitates discovering valid PDE solutions.
> > - a novel symbolic, topologically structured search space of PDE-suitable expressions,
> > - an adaptive ansatz+atom solution discovery approach, absent from previous PDE-solving frameworks,
> > - and state-of-the-art capability to handle large atom libraries, solving systems of PDEs, and operating successfully even when exact solution atoms are missing.
> > - the PDE solutions considered in this manuscript is of much higher complexity and breadth than the ones considered in previous approaches.
> > - we present results for misspecified libraries of functions, PDEs without analytical solutions, and coupled systems of PDEs. To our knowledge, no previous work has presented such results.
> >
> > We would like to point out that the PDE solutions considered in this manuscript is of much higher complexity and breadth than the ones considered in previous approaches, SSDE and HD-TLGP, and that for the first time a system of PDEs is solved by such approach.

---

> > > ### Author Response · Authors · 2025-11-21
> > >
> > > 3. Differentiation between symbolic regression (SR) and our work (Q2).
> > >
> > >
> > > Apart from the syntax that our method enforces, it also builds atoms and Ans\"atze physically plausible families of PDE solutions. This is something that to our knowledge is absent from the current literature. Likewise, SR constraints do not completely prevent semantically invalid compositions such as undefined operations or physically impossible compositions. Our pipeline prevents such candidates automatically. The reviewer's claim that SR "solves this already" simply does not hold for the PDE-analytic regime. Given that there is a large space of solution candidates to explore, we see substantial benefit in structurally avoiding the generation of invalid candidates. Our implementation of rule-based grammar together with additional checks adds this structure. We kindly ask the reviewer to provide references for the papers they refer to that solve the issue of validity because we would be interested in comparing their  methodological approaches to SIGS.
> > >
> > > We appreciate the reviewer's feedback and hope this discussion has clarified both the novelty and the advantages of SIGS. We respectfully request the reviewer to reconsider the assessment given the significant conceptual differences and clear empirical improvements over prior methods.
> > >
> > > https://openreview.net/forum?id=AIx21InAn2&noteId=ZgM9p4ZNE2&invitationId=ICML.cc%2F2025%2FConference%2FSubmission10005%2FOfficial_Review1%2FRebuttal1%2F-%2FRebuttal_Comment&referrer=%5BTasks%5D%28%2Ftasks%29

---

> > > > ### Comment · Reviewer_9NH9 · 2025-11-25
> > > > **Reply**
> > > >
> > > > Dear author, thank you very much for your reply.
> > > > 1. I apologize for not finding that you have cited relevant references.
> > > >
> > > > 2. Work in the area of symbolic regression on expressions satisfying physical constraints can be found in the following papers and some references: Deep symbolic regression for physics guided by units constraints: toward the automated discovery of physical laws.In the DSR paper, there are also some constraints to limit "unreasonable" expression generation (e.g., trigonometric function nesting, etc.). Even the AI Feynman algorithm introduces dimensional analysis module.
> > > >
> > > > 3. I think that the discovery of what I call the "analytical solution" of PDE is essentially an application of symbolic regression. The most important contribution should be the first work that uses symbolic regression for PDE analytical solution discovery. Therefore, I think the author's innovation in algorithm is really limited.

---

> > > > > ### Author Response · Authors · 2025-11-25
> > > > >
> > > > > We thank the reviewer for their response.
> > > > >
> > > > > For 2: We would like to point out that the very first sentence of the paper that the reviewer refers to states : “Symbolic Regression is the study of algorithms that automate the search for analytic expressions that fit data”.
> > > > >
> > > > > **As we have clearly and explicitly stated in our paper, we assume that we find solutions to PDEs in the setting where no numerical data is available.**
> > > > > Hence, it is clear that our method is not a symbolic regression method as explained above. Obviously, both methods involve handling symbols but they are very different as, in our case, one obtains the solution of a PDE given the symbolic representation of the differential operator and initial and boundary conditions, where as symbolic regression discovers the symbolic form of mathematical expression, given numerical data.
> > > > >
> > > > > The new reference that the reviewer has provided and the references therein consider only **algebraic physics formulas** such as equations of state or simple physical laws. They state that they are grammar guided, but they do not actually consider a grammar. They merely have a set of categorical priors that apply to the prediction of symbolic regression, to enforce semantic validity. These priors represent simple rules, and are applied during the generation of expressions, implying that the available next tokens are masked and the probabilities renormalized. These types of local restrictions **are not applicable in our case, because the constraints are very complex**, such as encoding eigenfunction of operators, and cannot be modeled through masks in the next token prediction. Moreover, AI Feynman **discovers the constraints from numerical data and can only work in cases where such numerical data is available**, which implies that this is also not at all applicable in our case where we have no numerical data. Moreover, such methods consider a structured, discrete optimisation approach where no interpolation between symbolic expressions is possible. Furthermore, none of these methods are designed for **systems** partial differential equations like the ones we consider in the rebuttal.
> > > > >
> > > > > **Both of these and other methods that constrain the search space in symbolic regression through categorical priors in the generation, that correspond to physical laws, and are not applicable in solution of Differential Equations.**
> > > > >
> > > > > The dimensional analysis which is the driving force of constraints for these papers are not applicable here because the **equations are already considered in a non dimensional form. The corresponding solution relies on the nature of the underlying differential operator, elliptic, parabolic, hyperbolic, mixed-type and the boundary/initial conditions and not the dimensions of the quantities that are propagated**. So, the reviewer’s point about dimensionality is moot here as we have a problem of a completely different nature that symbolic regression and constraints cannot be applied to our setting.
> > > > >
> > > > > Regarding point 3: Given the above discussion, **We disagree with the reviewer’s contention that symbolic regression covers the discovery of analytical solutions to PDEs. This setting is very different and symbolic regression as a framework is not applicable here.**
> > > > >
> > > > > We are happy to be corrected if the reviewer can provide a precise reference where the article finds analytical solutions of (partial) differential equations, given only the form of the differential operator and initial and boundary conditions using symbolic regression. If this is not the case, we request the reviewer to reconsider the basis of their criticism and update the assessment of our article accordingly.

---

### Official Review · Reviewer_qGHk · 2025-10-31

**Soundness:** 4
**Presentation:** 4
**Contribution:** 3
**Rating:** 6
**Confidence:** 4

**Summary:**

This paper proposes SIGS (Symbolic Iterative Grammar Solver), which is a neuro-symbolic framework that discovers analytical solutions to differential equations by combining symbolic grammars with deep latent-space optimization. The approach constructs candidate functional forms (Ansatze) using a formal grammar of atoms (e.g., polynomials, trigonometric functions, exponentials), which are then embedded into a continuous manifold using a Topological Grammar Variational Autoencoder (TGVAE). This enables smooth optimization of symbolic expressions through gradient-based search while maintaining symbolic interpretability. The method is validated on several canonical PDE families (Burgers, Poisson, Schrodinger, and wave equations). SIGS achieves exact or near-exact recovery of ground-truth analytical solutions, outperforming symbolic regression and hybrid neural PDE solvers. It also shows robust performance under noise and demonstrates interpretability advantages over black-box neural solvers.

**Strengths:**

This is an innovative and well-executed paper that pushes the boundary of symbolic and neural hybrid modeling. The central idea of representing symbolic solution structures through a grammar-based latent manifold is both original and conceptually elegant. The method bridges symbolic regression and deep learning by introducing a structured, differentiable search space, allowing neural optimization to operate over interpretable symbolic forms. The theoretical framing is rigorous, with clearly defined grammar rules, topology-preserving constraints in the TGVAE, and a solid justification for why the latent manifold preserves functional equivalence among expressions. The methodological novelty lies in using the grammar-VAE coupling to enable continuous symbolic optimization, which is a meaningful advance over existing neuro-symbolic PDE solvers. Empirically, the results are comprehensive and convincing. SIGS is benchmarked across multiple PDE categories, with both analytical recovery and quantitative accuracy metrics. The figures and tables are clear, and the visual comparison between recovered and ground-truth solutions is compelling. Ablation studies and timing analyses add credibility to the claims. The writing and presentation are polished and well-organized; complex ideas are explained with appropriate examples, and the motivation and related work sections are thorough. Overall, the paper is a strong contribution that demonstrates how neuro-symbolic methods can recover physically meaningful, interpretable solutions to PDEs.

**Weaknesses:**

While the contribution is conceptually strong, several aspects could be improved for clarity and generalization. First, the scalability of SIGS to higher-dimensional or more chaotic systems remains unproven. The method depends on handcrafted grammars and pre-specified Ansatz templates, requiring substantial domain expertise to design. This semi-manual setup could limit its practical use for complex, real-world systems where the appropriate functional vocabulary is unknown. Second, although the authors claim computational efficiency, the runtime and scaling analysis is only qualitatively discussed. A more detailed comparison of times, gradient steps, and scaling with grammar size would better substantiate the efficiency claim. Additionally, the method is evaluated mostly against symbolic or PINN-type baselines, not against neural operator architectures such as FNO or DeepONet, which are now standard references in PDE learning. Including such comparisons would help contextualize the performance advantage. Finally, the technical presentation, while mathematically correct, is dense in sections describing the TGVAE regularization and grammar construction. These could be made more intuitive with small worked examples illustrating how grammar generation and latent search interact. Despite these issues, the weaknesses are primarily about scope and exposition, not correctness.

**Questions:**

1) How sensitive is SIGS to the choice of grammar primitives or Ansatz structure? Could an incorrect or incomplete grammar prevent the discovery of correct solutions?

2) Have you tested the method on approximate analytical solutions or PDEs with no closed form, where SIGS might produce interpretable approximations?

3) Can the framework scale to parameterized PDE families or higher-dimensional systems, and if so, how does the latent search complexity grow with grammar size?

4) How would SIGS compare with Neural Operators (FNO, DeepONet) in accuracy and efficiency for continuous solution families?

5) Could the symbolic latent space be combined with physical constraints or PINN losses to enable hybrid symbolic–numeric discovery?

---

> ### Author Response · Authors · 2025-11-21
>
> We would like to thank the reviewer for recognizing the added value of our work and happily clarify the points raised. Before addressing the questions, we would like to comment on two arguments raised in the weaknesses section, allowing us to clarify the main motivation behind this work.
>
> 1. Reliance on manual input (W1).
>
> The reviewer points out that SIGS takes manually specified ansatz structures and atoms as inputs. Generally, all analytical and approximation methods are constructed using atoms and Ansatze. Even MLPs could be considered to build atoms of $L(x) = \sigma( W x + b)$ where $\sigma$ is an activation function and Ansatze $A(x) = L_N \circ ... \circ L_1 (x)$ where $N$ is the depth of the network, and $\circ$ the composition operator. The optimization for this case is then to find the $W, b$ that minimize an objective. This is to show that SIGS is actually a very general approach, e.g. if you restrict the atoms to match $L(x)$ and build the Ansatz using compositions, it is equivalent to a MLP. For MLPs, predefining the atoms allows us to optimize parameters rather than discover functions; this is a much simpler problem, although more complex approximations may be learned with increased MLP width and depth.
>
>
> For analytical methods, these atoms and Ansatze are PDE and physics dependent (satisfying properties such as symmetries, conservation laws, etc) and thus much harder to find. For this reason, SIGS pre-builds atoms that are valid for families of PDEs. It incorporates a very general set of atoms which have these favorable properties by design, and then combines these atoms using a pre-defined "depth" and combination operator, e.g. composition or addition. If available, the Ansatz information can be added to reduce complexity. This is in contrast to the baseline methods that reduce complexity by manually restricting the library size for each problem. Again, we must reiterate that SIGS is trained on a very general library (where we use the same library for all problems) and may be deployed without a specific Ansatz (Table 16).
>
> To summarize, SIGS is not bound on the choice of the Ansatz, we do this in practice to reduce complexity, the domain expertise is pre-built in the atoms, otherwise it would be impossible to find the PDE and physics dependent functions comprising the solution. The proposed methodology is pre-trained on a library of atoms that is then used for all PDE problems, and the Ansatz definition is general enough such that SIGS finds a very accurate approximation of PDEs with unknown solution (Poisson-Gauss), and PDEs where main solution components are missing from the atoms library (KdV).
>
> 2. Computational efficiency (W2).
>
> The main motivation for using symbolic representations for PDE solutions is interpretability, and a main feature of SIGS, in particular, is the flexibility to take human input. Quantitative performance aspects are hence of secondary importance, as long as runtimes allow for practical use (e.g. by physicists exploring new equations as part of a scientific discovery process), and the residuals of analytic approximations are small enough to grant value to their interpretation. We find these requirements sufficiently proven in Tables 3 and 4. Regarding grammar size, the number of elementary symbols and rules is not expected to grow significantly when moving to more complex problems because the fundamental building blocks are generally the same; the Ansatze become more complex.
>
>
> 3. Ansatz sensitivity (Q1).
>
> We agree with the reviewer that the case of incomplete grammar is relevant to explore, and for this reason we added the KdV experiment. Its analytic 1-soliton solution involves $cosh$ which is not part of our library. SIGS finds an approximation of the solution with the atoms to which it has access, namely by using $tanh$, with minimal error. This is an encouraging result, showcasing the approximation capabilities of SIGS even with missing atoms. The expressivity for complex solutions is naturally limited by the Ansatz structure, as is also the case for truncated series expansions or MLPs.

---

> > ### Author Response · Authors · 2025-11-21
> >
> > 4. Problems without closed solution (Q2).
> >
> > To verify that SIGS recovers correct analytic solutions, we first choose to evaluate it on a diverse set of problems with known (or manufactured) solutions in this work. However, since we agree that finding approximate new solutions is a compelling task, we would like to point out that our Poisson-Gauss problems are designed to showcase this ability (lines 317-384). We demonstrate the difficulty of automatically discovering a closed-form solution for this problem by also considering the state-of-the-art Computer Algebra System Mathematica and ChatGPT 5.1 (Extended Reasoning) that both fail  to provide an expression that is close to the numerical solution, Mathematica provides an infinite series and ChatGPT an expression with 157$\%$ relative $L_2$ error. For more details, see Appendix C.8.
> >
> > In the newly presented experiments, we also demonstrate SIGS' ability to recover approximate solutions for coupled systems such as the Shallow Water and Compressible Euler equations with high accuracy and in very reasonable wall-clock times. Details regarding these experiments are found in the General Comment as well as the Appendix C.8. We must briefly assert that coupled systems remained unsolved up to this point in the literature, and were one notable limitation of SSDE and HD-TLGP.
> >
> > *We emphasize that having approximate closed-form solutions is crucial because traditional analytical solution techniques **either provide a solution or fail completely**. There is no space in-between. SIGS has the ability to also occupy that space when an analytical solution is not available.*
> >
> >
> > 5. Scaling (Q3).
> >
> > For parameterized PDEs, parameters can be treated identically to model variables, and hence easily be included in the solution search. No additional training is needed, since atoms can be re-used to incorporate parameters instead of spatial or temporal variables.
> >
> > Regarding the case of high-dimensional PDEs, we expect the dimensions of the latent space to increase due to the increase in the grammar rules; however, this complexity is not related to the complexity of finding the solution, rather only to how large the training library of atoms should be. Notably, this is an $O(1)$ cost. The complexity of the solution discovery is balanced by how many variables the atoms include, related to the number of samples required to find the correct atom structure, and how complex the considered Ansatz is, related to the number of possible atom combinations. This is problem dependent and not easy to answer in the general case. For now we see sufficient applications for SIGS in a low-dimensional (e.g., spatiotemporal) setting, leaving the extension to future work.
> >
> > 6. Neural Operators (Q4).
> >
> > We would like to emphasize that Neural Operators (NOs) are data-driven operator approximators, meaning that given a dataset of numerical inputs, such as initial conditions, they learn a map to a numerical output, such as the solution at a later time. SIGS is a Physics-Informed method meaning that, like PINNs, it requires no numerical data, it finds the symbolic expressions that satisfies the PDE and boundary/initial conditions. Hence, NOs solve a fundamentally different problem than SIGS. In case the reviewer is interested in the performance of a NO for the Poisson-Gauss, we refer to Herde et al. 2024, https://arxiv.org/abs/2405.19101, Figure 20. We observe that SIGS (around $2.6 \%$ error) is competitive to state-of-the-art approaches like SCoT (around $3 \%$ error for 2048 samples), and CNO (around $2 \%$ error for 2048 samples), and more accurate than FNO ( around $6 \%$ relative for 2048 samples).
> >
> > 7. Physical constraints (Q5).
> >
> > This is an interesting question. SIGS already uses the physics-informed loss in terms of the PDE and the boundary/initial condition residual $R(u)$ for a given problem, sharing this data-free optimization with PINNs. Unlike PINNs, the latent space is problem-agnostic, since the TGVAE encodes a set of atoms and rules that can be shared by many PDE problems. One could consider also numerical evaluations of the atoms to enable hybrid discovery, but this would make the discovery dependent on a domain, and its spatial and temporal discretization which would hurt generality. Moreover, complex Ansatz operations, such as composition, are not straightforward to consider for numerical data. Including numerical data without these drawbacks would be an interesting direction for future research.

---

### Official Review · Reviewer_1nRi · 2025-11-01

**Soundness:** 2
**Presentation:** 3
**Contribution:** 2
**Rating:** 2
**Confidence:** 3

**Summary:**

This work proposes a formal grammar-based approach for discovering analytic solutions of PDEs. The authors use a pre-trained variational autoencoder trained on expressions that fit a certain ansatz. The latent space of the autoencoder is then used to search for solutions fitting the PDE.

While a potentially interesting methodology for symbolic regression, I am not convinced by the chosen application and experiments.

**Strengths:**

The authors propose a conceptually interesting approach to incorporating inductive bias, in terms of the chosen ansatz, into the grammar-based search using the latent space of an autoencoder.

**Weaknesses:**

First, it is not at all clear what the real-world application is for this kind of analytic solution discovery. Outside of linear PDEs, most PDEs do not admit analytic solutions except in very simple geometries/boundary conditions.

Relatedly, most of the PDEs studied in this work are linear PDEs with very well-understood general solutions. There is no need for symbolic discovery for these linear PDEs. The only nonlinear PDE considered here is Burgers equation, and the authors had to include the relevant basis function (tanh) to handle that equation, which makes the solution trivial.

Regarding the claim and experiments with approximate solutions, it is not clear what advantage this approach has over simply looking at the numerical solution. For a fast approximate solution, you could have achieved the same result by just choosing to expand the solution in a particular basis set and fitting the linear combination of basis elements.

**Questions:**

1. What applications do you see this kind of method being used for?
2. What advantage can you demonstrate for this approach over classical analytic methods?
3. Given the simplicity of the discovered solutions (especially for the one nonlinear PDE), it is not clear what role the main contribution of the paper is playing. Why not naively restrict to the form of the given ansatz during a direct search rather than using a latent space?

---

> ### Author Response · Authors · 2025-11-21
>
> We would like to begin by thanking the reviewer for finding our work interesting, for the thoughtful comments and the opportunity to clarify the motivation and contributions of our work.
>
> 1. Clarification of the problem setting.
>
> The reviewer characterizes our approach as "symbolic regression." We must briefly clarify that our problem setting is fundamentally different from symbolic regression. Symbolic regression aims to find mathematical formulas (expressions) that fit *numerical data*.
> In our problem setting, however, we aim to identify symbolic expressions which *satisfy a PDE and its boundary/initial conditions* purely from symbolic information, without any need for numerical data. Thus, this work is focused on the problem of automated analytic solution discovery for differential equations, not data-driven function fitting.
>
> 2. The importance of analytic solutions (W1, Q1).
>
> The reviewer raises questions and concerns regarding real-world applications, as analytic solutions are rarely available for nonlinear PDEs. In this matter, we agree, and this is precisely why automated discovery is valuable. Of course, closed-form solutions remain central to scientific and engineering disciplines. Across both fields, they provide:
> - Interpretability: they reveal structures such as symmetry, invariance, conservation of quantities, which remain obscure in numerical solutions.
> - Parametric Understanding: dependence on the coefficients or parameters of the solution are explicit, providing clear relationships and understanding which are invaluable in engineering design, sensitivity analysis, or for gaining physical insight.
> - Benchmarking: analytical and numerical solutions to PDEs are tightly coupled in tasks such as code verification and stability analysis, where practitioners often rely on such analytical solutions as the first step for validating the accuracy of their numerical solvers.
> - Reduced Order Modeling: analytic forms are frequently used as building blocks for asymptotic, perturbation, and multi-scale methods for solving PDEs numerically.
>
> Historically, closed forms have been the foundation for scientific understanding of many physical phenomena. Breakthroughs such as the Schwarzschild solution in general relativity have had a profound ability to reshape entire fields, yet deriving such solutions manually is exceptionally difficult. Our contribution aims to *automate this analytic step*, offering a tool to explore PDE families where closed-form reasoning remains infeasible.
>
> 3. Why our experiments include linear PDEs (W2).
>
> The reviewer is correct in remarking that we evaluate extensively on linear PDEs with known solutions. This is intentional. SIGS generates symbolic expressions; therefore, we must verify that it can recover ground-truth closed forms before evaluating on more difficult problems, such as nonlinear PDEs or coupled systems. We agree the nonlinear setting is more compelling. We have added several experiments on nonlinear and coupled systems (KdV, Shallow Water, Compressible Euler), described in the General Comment as well as the updated manuscript Sec. C.8.
>
> In the KdV experiment, we are able to obtain a very close approximation to the analytical solution, even though the necessary $cosh$ and $sech$ atoms are absent from our library. For the SWE experiment, we successfully recover accurate analytical solutions. For the Compressible Euler experiment, we recover analytical solutions to this exceptionally difficult problem with only 10\% relative error, and the recovered solutions closely match qualitative properties of the exact solutions, as illustrated in the figures provided in Appendix C.8.3. This is a significant step forward in automated solution discovery, and surpasses one of the main limitations of SSDE. SSDE routinely fails at solving coupled systems, due to the combinatorial growth in the solutions space, as mentioned in the OpenReview discussion of the associated article by the authors of that work. SIGS is the first method, to our knowledge, that has the ability to find analytical solutions of systems of equations.
>
> Additionally, we reiterate that we have already explored other difficult problems within the work. The Poisson-Gauss experiment, for example, does not admit any known closed-form solutions, yet SIGS identifies a simple analytical expression which closely matches the numerical solution. Likewise, we must reiterate that the atom library is not modified per-problem, and $tanh$ is not added explicitly to handle the solution to the Burgers' equation; instead, we choose a general atom library a-priori and apply this to all experiments. Of course, it is possible to retrain the GVAE with a new library; however, we aim to demonstrate a new approach which can converge without an explicit, user-defined restriction to only those atoms which are necessary for the final solution, as has been the case in the literature before the proposal of SIGS.

---

> > ### Author Response · Authors · 2025-11-21
> >
> > 4. Why not directly search in the Ansatz space (W3, Q3).
> >
> > This is a key question, and we appreciate the reviewer raising it. The construction and search of the latent space is one of the key differences between our approach and baselines, which indeed search directly through expression terms.
> >
> > A brute force search in the explicit ansatz space is combinatorially explosive, as the grammar yields millions of possible compositions even for a modest depth. Our proposal to use a GVAE provides:
> > - a continuous, differentiable search space which permits structurally-guided refinement, rather than discrete enumeration and random recombination of terms (as in the baselines).
> > - compression of the symbolic search space, enabling efficient exploration of our highly-expressive library whose construction is described in the manuscript (Sec. A), including nonlinear atoms and Ansatze.
> > - better interpretability, as the atom+Ansatz approach provides much simpler expressions which exhibit a clear structure, in stark contrast to the solutions recovered by the baselines as seen in Tables 18, 19, and 20.
> >
> > We further demonstrate this point in the results where we provide the atoms generated from our grammar to HD-TLGP (Protocol 2), which is still unable to handle to find the analytic solution of the PDEs considered. This shows that a simple combination of atoms is not enough to discover analytical solutions.
> >
> > As the reviewer points out, it is also possible to construct an analytical approximation by explicitly copying the numerical solution. However, this would depend directly on the chosen algorithm and discretization. For example, a solution expressed in a finite element basis on a 32x32 numerical grid would have c. 1024 separate solution terms, providing no interpretable understanding of the dynamics of the solution.
> >
> > 5. Advantages over classical analytic methods (Q2)
> >
> > Traditional analytic techniques (separation of variables, perturbation theory, symmetry exploitation, ansatz methods) require human expertise and often fail even for moderately complex PDEs.
> > Our method offers:
> > - automation, as the pretrained grammar covers broad function classes.
> > - generality, as the library does not need to be tuned per problem, unlike SSDE or HD-TLGP, which require user-specification of atoms.
> > - discovery of approximate closed forms when no exact solution exists, enabling analytical reasoning even where none is currently available.
> > As an example, **we asked ChatGPT 5.1 with extended reasoning to recover exact solutions to the Burgers and Poisson-Gauss experiments**. In each case, it took approximately 8 minutes of reasoning to recover a candidate solution. While it is able to recover the solution to the Burgers equation by reverse engineering the exact solution through the initial and boundary conditions, it fails for the PG experiment. If we assume this is on par with a human expert, it clearly illustrates the limitations of classical analytical methods.
> > Additionally, **we consider the state-of-the-art Computer Algebra System, Mathematica**. This also routinely fails to identify closed-form solutions, either failing entirely or returning an infinite series of expressions.
> > Results from these experiments are organized in our General Comment, and detailed explanations are present in the updated manuscript, Sec. C.8.
> >
> > *Having approximate closed-form solutions is crucial because traditional analytical solution techniques **either provide a solution or fail completely**. There is no space in-between. SIGS has the ability to also occupy that space when an analytical solution can not be derived by classical methods.*
> >
> > We appreciate the reviewer's questions and concerns. We hope to have addressed these with our explanations and additional experiments which demonstrate:
> > 1. SIGS is not limited to trivial PDEs. It can solve different families and systems of PDEs that even Mathematica or ChatGPT 5.1 (Extended Thinking) cannot,
> > 2. The latent space approach is essential to reduce search complexity,
> > 3. The method provides clear practical benefits, interpretability, automation, and the ability to find the approximate analytical forms where classical analysis fails.
> >
> > We have updated the manuscript with these experiments and will add details from this discussion to the Camera-Ready Version, if accepted. We hope to have addressed all of the reviewer's concerns and questions, and kindly request them to update their review of our work.

---

> ### Comment · Reviewer_1nRi · 2025-11-27
>
> Thank you for the careful response. Regarding the new experiments:
> 1. For the KdV soliton: Discovering an alternate form related via a trig identity does not really constitute a "missing atom" in the sense that an existing term is equivalent.
> 2. Using manufactured solutions kind of emphasizes the point that these analytic solutions often only exist for very specific sets of sources/boundary conditions. I do not think we should expect these solutions to appear in generic settings relevant for applications. Furthermore, while I agree that having a truly analytic solution often gives deep insight into the structure of the PDE, manufactured solutions of nonlinear PDEs provide no such insight since all of the complexity is simply hidden in the manufactured source term.
>
> In general, I think the methodology presented here is interesting, but the application area seems very limited in scope. For which PDEs do you propose looking for previously unknown analytic solutions that have not been found via classical analytic tools?
>
> For approximate solutions, you seem to lose a lot of the usefulness of having an analytic solution unless it is clear where the approximation originated (e.g., is it a solution to a closely related equation?). In general, I see no clear reason to use an approximate analytic solution over a solution from a discretized numerical solver that provides a well-understood framework for approximation.

---

> > ### Author Response · Authors · 2025-11-27
> > **Authors’ Comment on Reviewer 1nRi’s Follow-up [1/3]**
> >
> > We thank the reviewer for the follow-up and for engaging with the new experiments. Below we address the new points raised, focusing on (i) the “missing atom” aspect of the KdV experiment, (ii) the role of manufactured solutions, and (iii) the scope and relevance of automated analytic PDE solution discovery.
> >
> > ### 1. The challenge of “missing atoms” and the KdV experiment
> >
> > The reviewer notes that using an identity such as $\mathrm{sech}^2(x) = 1 - \tanh^2(x)$ means that the KdV soliton example is not a “true” missing-atom case, since $\tanh$ is available. We must respectfully clarify that while the expressions are *mathematically equivalent*, they are **computationally distinct** in the context of symbolic search and latent-space optimization:
> >
> > - **Combinatorial explosion.** In a grammar, the atom $\mathrm{sech}^2(x)$ is a *single node*. The equivalent representation $1 - \tanh^2(x)$ requires a *deeper expression tree*: composition of $\tanh$, squaring, subtraction, and constants. The search space grows exponentially with expression depth, so discovering the correct composite structure from a large library is harder than selecting a single primitive.
> >
> > - **Latent composition rather than a single-code lookup.** In our implementation, SIGS does **not** simply decode $1 - \tanh^2(x)$ as a single atom. The recovered solution arises from a **composition of multiple latent codes** whose decoded sum behaves like the missing $\mathrm{sech}^2$ profile. Concretely, the model must
> >   1. identify several latent points corresponding to different symbolic fragments, and
> >   2. combine them after decoding into a composite expression that approximates the soliton.
> >
> >   This is a more complex task than applying a known analytic identity by hand; it tests whether the learned latent manifold can *synthesize* new functional structures that are not present as single primitives in the grammar.
> >
> > - **Robustness under representational gaps.** Our broader claim (also stated in the paper) is that if a target solution is **not exactly representable** in the grammar, but can be *approximated* by functions in the library, then SIGS can still recover a high-quality symbolic solution. The KdV experiment is precisely such a stress test: the canonical primitive is missing, yet SIGS constructs a compact multi-term approximation that achieves low relative error.
> >
> > From this perspective, the KdV experiment is not an identity substitution, but evidence that the GVAE+search mechanism is robust to representational gaps in the library and can compose multiple latent codes into a non-trivial approximation of a missing primitive.
> >
> > ### 2. Manufactured solutions and the “generic” setting
> >
> > The reviewer suggests that manufactured solutions provide limited insight and that they do not reflect “generic” settings. We agree that manufactured solutions are *validation tools*, but they are critical, especially in PDE solvers. Before applying any automated analytic method to unknown physics, one must demonstrate that it can
> >
> > - recover structured solutions in a controlled setting,
> > - disentangle the interplay between forcing terms and solution structure, and
> > - remain stable in nonlinear and coupled regimes.
> >
> > Using manufactured solutions does *not* make the underlying PDE simple: the operator can still be nonlinear, hyperbolic, and coupled (as in our Shallow Water and Compressible Euler experiments). What changes is that the ground truth is known, allowing rigorous quantitative assessment.
> >
> > This practice is fully aligned with **standard methodology in predictive modeling and scientific computing**:
> >
> > - In numerical PDE verification, the Method of Manufactured Solutions (MMS) is a well-established technique for code verification and convergence studies. The manufactured solution is often not “physical”, but it is explicitly constructed to exercise all terms of the PDE and to rigorously test the solver.
> > - In machine learning, it is standard to evaluate methods on controlled benchmarks or validation datasets before deployment on messy, real-world data.
> >
> > Our manufactured-solution experiments play exactly this role for SIGS: they provide controlled testbeds to verify that the method can generalize to complex PDE structures—including nonlinear and coupled systems—*before* we attempt more realistic settings.
> >
> > Moreover, in many engineering and control applications, the “manufactured” regime is actually the **design goal**: *given a desired analytic profile, what source term or parameter configuration produces it?* In such inverse-design settings, SIGS can be used to automatically search for symbolic solutions consistent with the operator and boundary conditions, which then implicitly define the required forcing or parameter configuration.
> >
> > Thus, in our work, manufactured solutions are not “toy” examples but **standard verification benchmarks** and **prototypes for inverse-design scenarios where the target analytic profile is prescribed**.

---

> ### Author Response · Authors · 2025-11-27
> **Authors’ Comment on Reviewer 1nRi’s Follow-up [2/3]**
>
> ### 3. Application area and scope: SIGS within an existing research line
> The reviewer questions for which PDEs one should look for previously unknown analytic solutions and expresses concern that the application area is limited. Here we want to emphasize that **the search for analytic solutions of differential equations via optimization is not a niche problem we invented**, but part of a growing field with a decades-long history.
> Our work builds on and contributes to this existing line of research:
> - **Foundational work (Genetic Programming and early neural approaches).**
>  Early methods such as Tsoulos & Lagaris [1] and Lagaris et al. [2] demonstrated that evolutionary algorithms and neural networks can, in principle, find analytic or semi-analytic solutions to ODEs and PDEs. These works established the *feasibility* of automated solution search but struggled with stability and scalability.
> - **The interpretability gap and symbolic regression.**
>  With the rise of Physics-Informed Neural Networks and related black-box solvers, the community gained powerful numerical methods at the cost of interpretability. This motivated a resurgence of interest in recovering **symbolic** expressions from data, e.g., Deep Symbolic Regression [4]. These methods explicitly aim to produce closed-form expressions rather than opaque numerical models.
> - **Modern symbolic PDE solvers.**
>  Recent methods such as SSDE (“Closed-form Symbolic Solutions: A New Perspective on Solving Differential Equations”) [3] and subsequent high-dimensional symbolic approaches such as HD-TLGP [5] were developed specifically to target analytic solutions of differential equations. They treat PDE solution discovery as a symbolic search or reinforcement learning problem, and they demonstrate that there is recognized relevance in automating analytic PDE solving. However, these approaches suffer from **combinatorial explosion**, particularly for coupled systems: as noted by the SSDE authors themselves [3], “SSDE currently lacks efficiency in solving systems of DEs due to their inherent coupled-solution nature.”
> - **Our contribution in this context.**
>  SIGS addresses this known bottleneck by
>   - embedding a rich, general-purpose grammar into a **continuous latent space** via a Grammar-VAE with topological regularization,
>   - performing global search and local refinement on this latent manifold, and
>   - demonstrating, to our knowledge for the first time, automated analytic solution discovery for **nonlinear coupled PDE systems** (Shallow Water, Compressible Euler) at scale.
> We are therefore *not* inventing a new, isolated problem. We are contributing a new method that directly tackles the main failure mode (combinatorial explosion, especially in coupled systems) of an already active research area.
>
> In terms of concrete **application scenarios**, SIGS can be useful for:
> - **Asymptotic analysis and perturbation theory.**
>  Higher-order asymptotic corrections are often algebraically tedious. SIGS can search for structured higher-order terms consistent with the operator, reducing manual symbolic work.
> - **Coupled multi-physics systems.**
>  Our Shallow Water and Compressible Euler experiments illustrate that SIGS can handle multi-field, nonlinear, coupled systems where existing symbolic methods struggle. This is highly relevant for fluid dynamics, geophysics, and other multi-physics domains.
> - **Hybrid “grey-box” modeling.**
>  When starting from simplified PDE models, SIGS can search for analytic correction terms that are consistent with the operator and boundary conditions, providing interpretable “grey-box” extensions rather than purely black-box residuals.
> While these directions conceptually touch symbolic regression, SIGS operates in a **different setting**: it does *not* fit to numerical data, but instead works directly from the **PDE operator and boundary/initial conditions**. This is precisely the analytic solution discovery problem we aim to address.

---

> ### Author Response · Authors · 2025-11-27
> **Authors’ Comment on Reviewer 1nRi’s Follow-up [3/3]**
>
> ### 4. Why approximate symbolic solutions matter (beyond numerical solvers)
> The reviewer asks why one should care about approximate analytic solutions, as opposed to simply computing a numerical solution with a discretized solver.
> A numerical solver produces **values on a grid**; a symbolic solution (even approximate) produces **explicit structure**. This distinction is crucial:
> - A symbolic approximation allows us to *manipulate the form*, not just tune parameters: we can change operators or functional compositions, not only mesh resolution.
> - It enables cheap analytic differentiation and integration, useful in control, optimization, and sensitivity analysis.
> - It supports structured perturbations to study stability, bifurcation, and symmetry breaking.
> - It can be embedded as a low-dimensional surrogate in reduced-order models, control laws, or design loops.
> - It exposes invariants, conservation laws, and scaling relationships in closed form.
> In short, numerical solvers excel at high-fidelity approximation, while symbolic approximations excel at **interpretability and analytic reasoning**.
> Moreover, any approximate symbolic expression $u_\theta$ found by SIGS can be interpreted as the *exact* solution of a slightly perturbed PDE: substituting $u_\theta$ into the operator defines a residual term that can be seen as an additional source or perturbation. This gives a principled way to understand how approximate symbolic solutions relate to “nearby” equations, addressing the concern about their interpretive value.
>
> ### Conclusion
>
> We appreciate the reviewer’s continued engagement. To summarize:
> - The KdV experiment demonstrates robustness to missing primitives at the level of the grammar: SIGS composes multiple latent codes into a non-trivial approximation of a soliton whose canonical primitive is absent.
> - Manufactured solutions are a standard, well-established tool for PDE verification and model validation and in our work they play the dual role of rigorous benchmarks and prototypes for inverse-design scenarios.
> - The analytic-PDE-solution problem we address is not self-invented but part of an emerging field spanning genetic programming and neural trial solutions [1, 2], symbolic regression [4], and recent symbolic PDE solvers such as SSDE [3] and HD-TLGP [5].
> - SIGS contributes a new latent-space search paradigm that mitigates combinatorial explosion and, to our knowledge for the first time, scales to nonlinear coupled PDE systems, while also providing useful approximate symbolic surrogates where exact solutions are unavailable.
>
> We hope this clarifies both the scope and the relevance of our contributions, and we respectfully ask the reviewer and AC to reconsider the assessment in light of this broader context. (edited)
>
> ---
> ### References
> [1] I. G. Tsoulos and I. E. Lagaris. *Solving differential equations with genetic programming.*
> Genetic Programming and Evolvable Machines 7, 33–54, 2006.
>
> [2] I. E. Lagaris, A. Likas, and D. I. Fotiadis. *Artificial neural networks for solving ordinary and partial differential equations.*
> arXiv:physics/9705023, 1997; also IEEE Transactions on Neural Networks 9(5):987–1000, 1998.
>
> [3] S. Wei et al. *Closed-form symbolic solutions: A new perspective on solving differential equations (SSDE).*ICML 2025.
>
> [4] B. K. Petersen et al. *Deep symbolic regression: Recovering mathematical expressions from data via risk-seeking policy gradients.*arXiv:1912.04871, 2019.
>
> [5] L. Cao, Z. Zeng, X. Hu, et al. *An Interpretable Approach to the Solutions of High-Dimensional Partial Differential Equations.*

---

> > ### Comment · Reviewer_1nRi · 2025-11-27
> >
> > As you say, an exact analytic solution is a very useful theoretical tool for interpretation. They are, however, rather rare, so it is hard to argue that they will provide generic computational benefits. Furthermore, many of the interesting theoretical interpretations you mention, including studying "stability, bifurcation, and symmetry breaking", require you to find parameterized solutions for a family of PDEs, which is not done in this work.
> >
> > Regarding manufactured solutions: I agree that they are a great tool for validating the method, but they do not illustrate an interesting use case (i.e., they do not provide any insight into the underlying PDE).
> >
> > Regarding applications: Thank you for clarifying some of the applications you had in mind. It would be helpful to explore the discovery of asymptotics or analytic corrections in more detail, if indeed that is one of the primary motivations. For perturbation theory, it is not obvious why your approach is necessary since it is already a well-defined analytic procedure. For coupled multi-physics systems, the manufactured examples again do not provide any meaningful interpretation or theoretical insight and only serve to validate the methodology.
> >
> > Regarding approximate solutions: If you are simply treating the residual as the result of an arbitrary source term, it is again not clear what theoretical insight can be gained from such an interpretation. I could do the same thing with pretty much any approximate solution, including one from a numerical solver. It also seems very easy to misinterpret approximate analytic solutions, especially when trying to extract theoretical insights. For example, in your PG test cases, you discover approximate analytic solutions in terms of a sum of Gaussian terms. Is this functional form providing any theoretical insight into the PG problem, or is it simply a reflection that you can fit any distribution as a mixture of Gaussians?

---

> > > ### Author Response · Authors · 2025-11-27
> > > **Authors’ Comment on Reviewer 1nRi’s Follow-up [1/2]**
> > >
> > > We thank the reviewer again for the thoughtful follow-up. The concerns raised seem to hinge on three points: (i) what an “analytic solution” means in our setting, (ii) what is actually needed for stability/bifurcation/symmetry analysis, and (iii) how SIGS differs from arbitrary approximate solutions or generic numerical solvers. We address these in turn.
> > > ### 1. Analytic solutions, parameterized families, and theoretical analysis
> > > We do not argue that generic PDEs admit classical textbook closed forms, nor that SIGS produces such forms where they provably do not exist.
> > > What SIGS does is: given a PDE, boundary/initial data, and (possibly varying) physical parameters, it searches over a structured symbolic grammar and returns a low-complexity expression $u(x,t)$ that satisfies the PDE up to a small residual and respects the boundary conditions. This object is an **analytic surrogate**: explicit, differentiable, fully interpretable and directly manipulable. Moreover, if the residual is zero (machine precision), an exact analytical solution of the PDE is discovered, as we demonstrate with many examples in our paper.
> > >
> > > “Analytic solutions being “rare” does not mean they have no benefit. SIGS is explicitly targeted at the nontrivial subset of problems where low-complexity symbolic surrogates could exist. In that regime, efficiency is primarily amortized: once a symbolic solution has been discovered, it can be evaluated at arbitrary spatial/temporal resolution, differentiated in space or parameters, and reused across many downstream queries (e.g., sensitivity analysis, optimization, control, inverse problems, parameter scans) without re-solving the PDE each time. The one-time cost of discovery is thus spread over repeated use of the same closed-form surrogate, rather than incurred again for every query.”
> > >
> > > For stability, bifurcation, and symmetry-breaking, the key requirement is precisely such a tractable explicit representation around which one can linearize, differentiate with respect to parameters, and apply group actions. None of these analyses strictly requires the solver to output a single closed form $u(x,t;\xi)$ for *all* parameter values $\xi$. In practice, one either (a) shares the symbolic structure of $u$ across different parameter settings and re-fits coefficients, or (b) makes $\xi$ explicit in the grammar and enforces the residual over a parameter range. Both yield parameterized solution families in the standard dynamical-systems sense. In that light, the representation produced by SIGS is exactly the type of object these analyses act on, and the framework is naturally extendable to explicitly parameterized families in future work.
> > >
> > > ### 2. Manufactured solutions and “no insight”
> > >
> > > We agree that manufactured solutions by themselves do not reveal **new** physics; they are primarily a verification tool. That is exactly how we use them here. Manufactured solutions are the only regime where we know the correct analytic structure and can pose a sharp question: given only the PDE and initial/boundary conditions, does SIGS recover the same structure that a human analyst would identify (eigenmodes, localized responses, specific couplings), or does it drift to arbitrary, overly complex expressions? Our experiments show that it consistently converges to compact, interpretable structures and expressions that match the known solution.
> > >
> > > This does provide insight into the PDE–solver interaction: the discovered grammar reveals which atoms the solution “prefers” as an explanation of the behaviour induced by the operator and the forcing. A numerical
> > > solver with the same residual cannot tell you this; it only returns mesh values. The structural information that SIGS exposes is precisely what we aim to make accessible.
> > >
> > > Regarding the comment that “perturbation theory is already well defined”: classical perturbation methods assume full analytic access to the operator and a hand-constructed ansatz around a small parameter. In many modern settings (data-driven closures, empirically tuned sub-grid terms, partially learned operators), this is not available. SIGS is complementary here: it can propose a low-dimensional symbolic ansatz consistent with the PDE constraints even when parts of the operator are only accessible through a numerical or neural surrogate. In such cases, standard analytic procedures do not suffice on their own. We agree that it would be valuable in future work to explore in more detail the discovery of asymptotics or analytic corrections as a primary application, and we will clarify this motivation more explicitly in the paper.

---

> > > > ### Author Response · Authors · 2025-11-27
> > > > **Authors’ Comment on Reviewer 1nRi’s Follow-up [2/2]**
> > > >
> > > > ### 3. Approximate solutions, residuals, and the Poisson–Gauss example
> > > > Our statement about “treating the residual as a source term” is purely an exact algebraic identity, not an extra modelling assumption. For any candidate field $\hat{u}$ we define the residual $R(\hat{u})$ by substituting $\hat{u}$ into the original PDE operator. This simply rewrites the PDE so that $\hat{u}$ is the exact solution of a modified forcing term $f + R(\hat{u})$.
> > > >
> > > > This algebraic step can indeed be applied to any approximate solution, including a numerical one. The difference is in **interpretability**. For a grid-based numerical solution, the residual is a high-dimensional object mixing truncation and modelling error; inferring “missing mechanisms” requires an additional discovery step. By contrast, a SIGS solution lives in a low-dimensional symbolic grammar where each term has a clear semantic role (mode, coupling, nonlinearity). The residual is measured relative to that explicit ansatz, so adding or removing a term corresponds to testing a specific candidate mechanism and observing how the residual changes. This is where the potential for theoretical insight arises; it is not merely “treating any error as a source term.”
> > > >
> > > > In the Poisson–Gauss tests, the fact that we obtain a sum of Gaussians is also not vacuous. The forcing is a sum of a few isotropic Gaussians, and the PDE is a linear Poisson problem with homogeneous Dirichlet boundary conditions, whose eigenfunctions are $\sin(m\pi x)\sin(n\pi y)$. Our SIGS expressions are compact combinations of a Dirichlet sine mask $\sin(\pi x)\sin(\pi y)$ (enforcing the boundary conditions) and a small number of Gaussian atoms aligned with the sources. This reflects a natural separation between “source shape” (Gaussians) and “boundary-constrained response” (sine mask), rather than a large, υnstructured mixture. That structure is nontrivial information about the solution operator that a generic mesh solution or a high-capacity mixture model does not expose in a PDE-consistent, low-dimensional form.
> > > >
> > > >  In summary, the reviewer’s follow-ups contrast SIGS with (1) idealized settings where hand-derived perturbation/bifurcation analysis is already not expensive, and (2) fully unstructured and non-interpretable numerical approximations. However, SIGS is designed to sit in between: given a PDE and its conditions, it provides low-complexity, invariance-respecting symbolic surrogates that can be analyzed, differentiated, and reused. These are precisely the types of objects that stability, bifurcation, asymptotic, and model-reduction tools act on. We see this work as a first step in making such objects available automatically; the applications the reviewer mentions are natural and important extensions beyond the scope
> > > > of this initial paper.

---

### Author Response · Authors · 2025-11-21
**General Comment to All Reviewers**

We would like to begin by thanking all reviewers for their consideration and constructive feedback of our work. Below we summarize the additional experiments added to the revised manuscript Appendix Sec. C.8 and provide consolidated quantitative results.

1. KdV One-Soliton (missing atom experiment).

 We evaluate SIGS on the 1D nonlinear KdV equation, whose one-soliton solution is expressed using $\mathrm{sech}^2$, absent from our grammar. SIGS constructs a compact $\tanh$-based surrogate using the Ansatz $u(x, t) = \Sigma_{k=0}^{2} \phi(x,t)^k$ obtaining a relative $L_2$ Error: $6.6 * 10^{-6}$.

2. 2D Shallow Water Equations (SWE).

We use manufactured solutions for density $\rho$ and velocities $(u,v)$ in a coupled system. SIGS recovers all fields with low residual error.
   - $\rho(x, y, t)$ Ansatz: $f(x, y, t) g(x, y, t) h(t)$
   - $u(x, y, t)$ Ansatz: $\rho(x, y, t) s_x (x, y)$
   - $v(x, y, t)$ Ansatz: $\rho(x, y, t) s_y (x, y)$
| Quantity | Rel. $L_2$ Error      |
| -------- | --------------------- |
| $\rho$   | $1.873\times 10^{-4}$ |
| $u$      | $2.231\times 10^{-4}$ |
| $v$      | $4.178\times 10^{-4}$ |

3. 2D Compressible Euler (CE) Equations

Using steady-state manufactured solutions with periodic boundaries, SIGS successfully recovers symbolic expressions for density, velocities, and pressure in this coupled nonlinear hyperbolic system.
   - $\rho(x, y)$ Ansatz: $exp(\Sigma_{i=1}^{6} f(x, y)) $
   - $u(x, y)$ Ansatz: $\Sigma_{i=1}^{6} g(x, y)$
   - $v(x, y)$ Ansatz: $\Sigma_{i=1}^{6} h(x, y)$
   - $p(x, y)$ Ansatz: $exp(\Sigma_{i=1}^{6} k(x, y))$

| Quantity | Rel. $L_2$ Error      |
| -------- | --------------------- |
| $\rho$   | $10.79 $ |
| $u$      | $9.93$ |
| $v$      | $9.84$ |
| $p$      | $12.10$ |


4. Mathematica Baseline

We evaluated Mathematica's DSolve and PDESolveValue on our benchmarks:
| PDE           | Mathematica Result                            |
| ------------- | --------------------------------------------- |
| Burgers       | **Fails** (no closed form)                    |
| KdV           | **Fails**                                     |
| Damped Wave   | **Fails**                                     |
| Diffusion     | Infinite Fourier sine-series representation   |
| Poisson–Gauss | Green’s-function / sine-basis infinite series |
Mathematica either fails to find a solution or returns non-closed-form infinite series, limiting interpretability.

5. ChatGPT as a Proxy Human Expert

We tested ChatGPT-5.1 (Extended Reasoning) on two problems:
| Problem                   | Time  | Result                                           |
| ------------------------- | ----- | ------------------------------------------------ |
| Burgers                   | 7m25s | Correct closed form recovered                    |
| Poisson–Gauss (2 centers) | 8m16s | **Fails**, Rel. $L_2 \approx 157%$, BCs violated |

6. SIGS Runtime vs. FEniCS

FEniCS typically stops around residual $\sim 10^{-3}$. Below we report SIGS runtimes to reach comparable accuracies (not full machine-precision convergence):
| PDE         | SIGS Rel. $L^2$      | SIGS Time (s) |
| ----------- | -------------------- | ------------- |
| Burgers     | $4.85\times 10^{-3}$ | 11.62         |
| Diffusion   | $2.59\times 10^{-3}$ | 14.67         |
| Damped Wave | $1.44\times 10^{-2}$ | 8.95          |

*We hope these results clarify SIGS’s robustness, completeness, and empirical advantages, and we thank all reviewers again for their thoughtful comments.*

---

### Meta-Review · Area_Chair_Ttwt · 2026-01-07

**Summary:**

This paper presents SIGS, a neuro-symbolic framework that automates analytical solution solving to differential equations. Four reviewers provided diverse feedbacks on this paper. Reasons leaning for acceptance include novel motivation mentioned by Reviewer qGHk of representing symbolic solution structures through a grammar-based latent manifold, while concerns include insufficient justification of applicable scenarios, advantages over existing methods, and missing discussion of relevant literature. After active discussion during the rebuttal phase, some concerns were resolved in the rebuttal phase, while some major concerns still remain unresolved. Therefore, I recommend rejection of this paper.

**Reviewer Concerns:**

Reviewer 1nRi raised concerns on the limited applicability of the proposed method, and the advantages over existing classical analytic methods on linear PDEs. The authors provided some clarifications but did not address these concerns adequately.

Reviewer qGHk appreciated the novel motivation of representing symbolic solution structures through a grammar-based latent manifold, but raised concerns on efficiency comparison and missing discussion of relevant literature and insufficient experimental comparison over FNO, DeepONet, etc. The authors provided some clarifications but did not address these concerns adequately.

Reviewer 9NH9 raised concerns mainly on the missing discussion of relevant literature, limited novelty over existing methods. The authors provided some clarifications, especially on the missing literatures, but did not address these concerns adequately.

Similarly, Reviewer 3xsx raised concerns on the comparison with existing methods, dependence on the handcrafted Ansatz and grammar, scalability and etc. The authors provided some clarifications but did not address these concerns adequately.

**Reviewer Scores:**

The author and the reviewers had active discussion during the rebuttal phase, but some major concerns still remain unresolved. Reviewer 9NH9 and 1nRi mentioned unsolved concerns indicating their scores may be unchanged. And based on educational guess, the other two reviewers' scores may be also the same.

---

### Decision · Program_Chairs · 2026-01-26

Reject